# Spatial planning with long visual range benefits escape from visual predators in complex naturalistic environments

Ugurcan Mugan ⬢ [1,2] & Malcolm A. MacIver ⬢ [1,2,3,4]✉

It is uncontroversial that land animals have more elaborated cognitive abilities than their aquatic counterparts such as fish. Yet there is no apparent a-priori reason for this. A key cognitive faculty is planning. We show that in visually guided predator-prey interactions, planning provides a significant advantage, but only on land. During animal evolution, the water-to-land transition resulted in a massive increase in visual range. Simulations of behavior identify a specific type of terrestrial habitat, clustered open and closed areas (savanna-like), where the advantage of planning peaks. Our computational experiments demonstrate how this patchy terrestrial structure, in combination with enhanced visual range, can reveal and hide agents as a function of their movement and create a selective benefit for imagining, evaluating, and selecting among possible future scenarios—in short, for planning. The vertebrate invasion of land may have been an important step in their cognitive evolution.

[1] Center for Robotics and Biosystems, Northwestern University, Evanston, IL, USA. [2] Department of Biomedical Engineering, Northwestern University, Evanston, IL, USA. [3] Department of Mechanical Engineering, Northwestern University, Evanston, IL, USA. [4] Department of Neurobiology, Northwestern University, Evanston, IL, USA. ✉email: maciver@northwestern.edu

A crucial feature of the history of vertebrates is the massive increase in the range of visual perception that occurred around 380 million years ago when they started invading land[1]. In water, short visual ranges cause visually guided interactions to be urgent and reactive. After the vertebrate invasion of land, this limitation in reaction time is lifted with the increased transparency of the viewing medium and a consequent increase in visual range by a factor of 100[1] (Supplementary Fig. 1). Thus, whereas an aquatic animal has to make decisions on the order of a second with limited information, the same animal on land has at least 100 times longer and much more information.

This increase in time-to-act—while a necessary condition for deliberation[2]—may not have been a sufficient condition for the evolution of planning in dynamic scenarios. Importantly, within the greatly enhanced range of Devonian aerial vision was rich structure provided by vegetation[3] and terrestrial topography, resulting in environments with greater complexity compared to aquatic habitats[4] (Supplementary Fig. 2a–c). While the increase in time-to-act extended the available time for deliberation, the concomitant increase in spatial complexity may have been essential in increasing the number of potential future scenarios, which depend on the dynamics of the agents (e.g., predator/prey) involved (Supplementary Fig. 2d–f).

This study builds on research into the neural basis of decision making that suggests the existence of two competing, and largely parallel systems: habit- and plan-based action selection[5]. Under habit-based action selection, there is no explicit consideration of action outcomes. Instead, action choices are sculpted by prior experience. Research in rodents shows that for both appetitive and aversive stimuli, a shift in behavior from plan-based to habit-based action selection occurs in conjunction with a shift from the associative to the sensorimotor cortico-basal ganglia network[6]. A similar paradigm also exists for birds[7]. The conserved organizational structure of the basal ganglia from lamprey, jawless fish that preceded mammals by 560 million years, to mammals[8], suggests that this structure—and thus the habit-based system it supports—evolved very early on in vertebrate evolution.

In contrast, plan-based action selection occurs by extrapolating from actions to their possible outcomes, drawing on diverse mnemonic representations such as spatial maps and requires the hippocampus (or its functional homologs)[9,10]. In mammals, spatial planning has been related to the phenomenon of nonlocal spatial representations in hippocampal activity[11], sometimes interpreted as prediction or imagination. Two quintessential examples of this are sharp-wave ripple[12] associated replays/trajectory events[13], and vicarious trial and error[11]. Current theories suggest that flexible decision making that depends on such imagined simulations requires interactions between the hippocampus and prefrontal cortex (PFC)[11,14,15]. Despite the differences in brain architecture between birds and mammals[16], planning in birds also requires interactions between the avian homologs of the hippocampus and PFC[17]. While there has been significant research into neural systems involved with habit and planning in tasks with stationary rewards[18] (termed habitizable), the role that increased visual range and environmental complexity might have played in altering the relative advantage of these two decision making systems in dynamic tasks with high reward uncertainty (termed non-habitizable) has not yet been explored.

Here, we test the hypothesis that in non-habitizable scenarios, plan-based action selection is advantaged in proportion to visual range and environmental complexity. Using computational simulations, we model prey evading a predator while trying to reach a distant goal. We then systematically vary visual range and environmental parameters—such as aquatic versus terrestrial structure—to understand how these parameters may contribute to the appropriate decision making strategy. We find that independent of environmental parameters, planning advantage was proportional to visual range. In spatially simple environments (uncluttered or highly cluttered), evasion strategies were highly stereotyped, enabling habit to perform as well as planning. In spatially complex environments that featured clustered open and closed areas—not unlike savannas—the existence of multiple viable futures maximally advantaged planning and caused habit to fail.

We discuss the relevance of our findings for visually dominant mammals and birds that depend on prediction rather than detection while hunting, as well as planning in other non-habitizable scenarios. Our results suggest a possible connection between the emergence of vertebrate life on land and the evolution of planning in dynamic scenarios.

## Results

**Overview of habit- and plan-based action selection.** In both habit- and plan-based action selection, action choices are dependent on the current state of the animal—spatial location of the prey and the predator—which creates an association between outcome and chosen action. This enables the prey to predict long-term reward. Neurobiologically, we can assume that sensory information allows the prey to detect the location of the predator, while the hippocampus (or its functional homologs) and memory provides allocentric location, and connections to PFC[19] or its homolog in birds[20] provides a state-value estimate.

The habit solution for long-term reward prediction (model-free[5,18]) assigns a value to an action or state based on prior experience. Thus, the state-action values are divorced from their immediate outcomes, resulting in inflexible responses to changed circumstances[6]. Conversely, the planning solution (model-based[5,18]) relies on an action-outcome knowledge structure to generate action sequences by imagining future states and their expected outcomes. While this method is computationally expensive, it also creates flexible responses to changing circumstances that may not have been previously encountered, such as those caused by the movement of a mobile threat or opportunity.

To study the effects of these decision making paradigms on performance and behavioral complexity as a function of visual range and environmental complexity, predator–prey interactions within highly idealized pseudo-aquatic and pseudo-terrestrial scenarios were considered (Fig. 1a, b). For both of these computational experiments, the predator was designed as a reflex agent (aggressive prey pursuit with some randomness) with a belief distribution reflecting the likely locations of the prey when it was out of view. The prey was configured to have either habit-based action selection, or plan-based action selection with a preset number of states that it could forward simulate. Note that we were constrained to simulate planning in only the prey due to the high computational burden of simulating planning in more than one agent[21]; the choice of prey rather than predator was arbitrary, and we do not expect it to affect our primary findings. First, we simulated midwater aquatic conditions (Fig. 1a) where the prey's visual range was varied in a simple (open) environment. Structured aquatic environments such coral reefs are also considered in a subsequent analysis. Second, we simulated terrestrial conditions (Fig. 1b) by adding obstacles until a predetermined level of clutter density was reached (quantified by entropy (Eq. (2)), see Methods). Unlike the aquatic condition, visual range was limited only by the presence of occlusions, where the prey and the predator could not observe each other if an occlusion existed on the ray between them.

The prey's start location and goal location (Fig. 1a, b: Safety) was fixed, and the predator start location was randomly selected

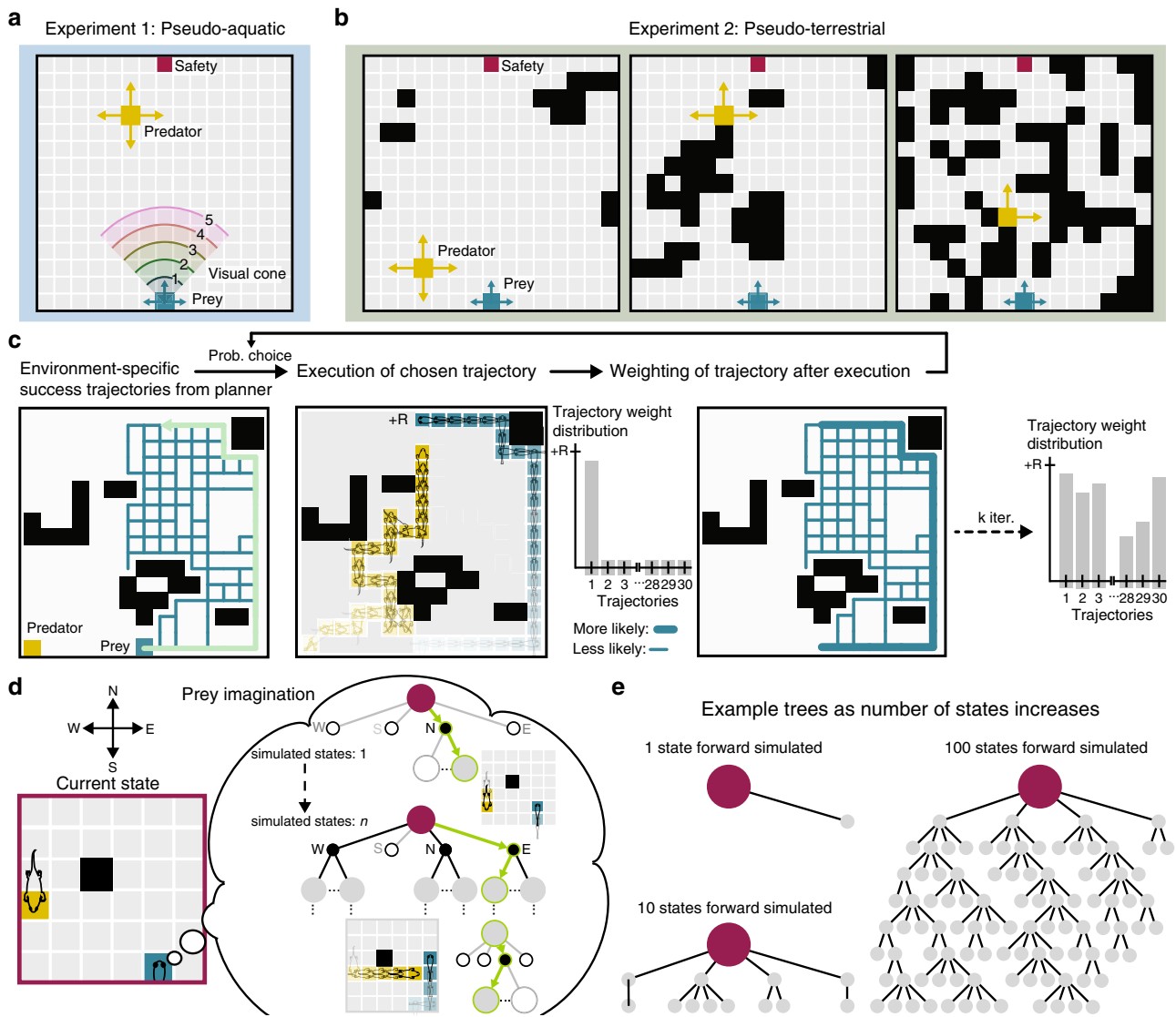

**Fig. 1 Environment models and schematic of habit- and plan-based action selection. a, b** Example 2-D environments used in the simulations; black squares represent obstacles. Examples of low, medium, and high clutter are shown for the pseudo-terrestrial condition. These environments can be experienced in the context of a predator-prey online game at https://maciverlab.github.io/plangame/. **c** Schematic of habit. A set of success paths (initially all weighted equally) are used in a loop in which a path is selected (green line with arrow) with probability proportional to its weight. After execution, the path is weighted by its total discounted reward, provided that it resulted in survival. Example weight distribution after *k* trials. **d** Schematic of planning. The prey imagines a tree of possibilities from the current state (dark red) by selecting virtual actions (green: next action and next state, white fill and black edge: unexplored possible actions, white fill and gray edge: unexplored possible next states, black fill: explored actions, gray fill: explored next states). Example virtual actions by the prey and the predator are shown on the smaller grid. **e** Example trees grown given a specified number of states being forward simulated.

for each trial. Both the predator and the prey had a complete cognitive map of the space (see Supplementary Note 1). Moreover, the prey had an accurate model of predator action selection.

**Algorithmic implementation of behavioral controllers.** Habit-based action selection exploited prior action sequences that resulted in survival for a given visual range and/or environment (Fig. 1c). Similar to prior approaches[5], we used successful trajectories generated by the planning algorithm at the maximum planning level (5000 states) to initialize the set of habit-based trajectories. In doing so, we ensure that we isolate the effect of behavior being driven by a previously learned sequence of actions,

in the absence of sensitivity to changes in predator location and in the absence of sensitivity to immediate outcomes. If instead we initialized a set of habits by a large number of training trials through model-free methods, differences in performance between habit and planing may stem from differences in initial knowledge. We note, however, that the limitation we set on the number of forward states simulated by the planner may result in suboptimal policies.

After initialization of the set of trajectories that seeded the habit-based action selector, a trajectory was chosen with probability proportional to its total discounted reward[22] (Eq. (1)), provided that it led to survival. While the probability of choosing an action sequence increased if it resulted in past survival, the action sequence itself was not changed and was executed until

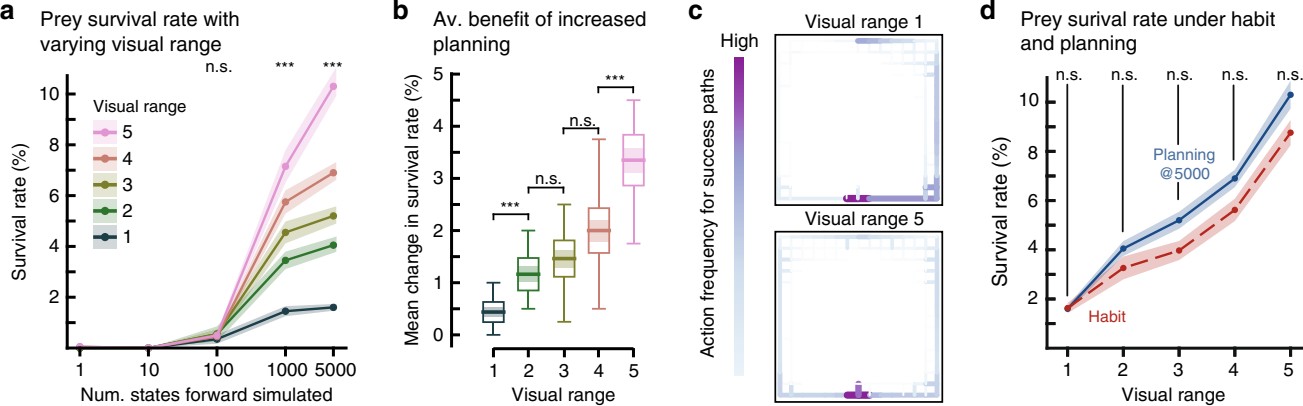

**Fig. 2 The utility of different behavioral controllers in open environments. a** Prey survival rate in pseudo-aquatic environments (Fig. 1a). The line plot shows the mean survival rate, and the surrounding fill indicates ±s.e.m across random initial predator locations ($n = 20$). Two-tailed Kruskal–Wallis (KW) test: $H_{100} = 2.0$, $p = 0.57$; $H_{1000} = 55.5$, $p < 10^{-10}$; $H_{5000} = 81.3$, $p < 10^{-16}$. **b** Mean change in survival rate across all the planning levels shown in (**a**) (see Methods). Horizontal line: mean; Shaded region: ±s.e.m; Box: 95% confidence interval of the mean; Vertical line: range of the data. $n = 20$ independent random initial predator locations. Mann–Whitney $U$ (MWU) tests (two-tailed) across visual ranges (with Bonferroni correction: $U_{1,2} = 54.0$, $U_{2,3} = 159.5$, $U_{3,4} = 127.5$, and $U_{4,5} = 63.0$ (\*\*\*$p < 0.0001$; n.s. is not significant $p > 0.025$) (KW over all visual ranges $H = 55.4$, $p < 10^{-10}$). **c** Heatmaps of all action sequences taken by the prey that resulted in prey survival at the maximum planning level (5000 states), with color density proportional to frequency. Color bar action frequencies range from 1 to 206, dependent on visual range (for all paths see Supplementary Fig. 3). Survival paths are aggregated across all tested predator locations ($n = 20$) and the total number of episodes per predator location ($n = 100$). **d** Mean ± s.e.m. across random initial predator locations ($n = 20$) of survival rate for prey that uses habit (red dashed line). The planning data (blue solid line) is another representation of the plot shown in (**a**) at maximum planning level (±s.e.m across random initial predator locations; $n = 20$). KW test (two-tailed) for each visual range (with Bonferroni correction): $H_1 = 0.11$, $p_1 = 0.74$; $H_2 = 3.53$, $p_2 = 0.30$; $H_3 = 4.26$, $p_3 = 0.20$; $H_4 = 3.42$, $p_4 = 0.32$; $H_5 = 4.04$, $p_5 = 0.22$. Source data are provided as a Source Data file.

termination (prey death or survival: reaching safety), which resulted in inflexible responses. In contrast, with plan-based action selection, within the imagination of the prey, each virtual action was evaluated based on the virtual action's possible outcomes. The prey thereby generated action sequences in imagination (Fig. 1d) that were dependent on the provided cognitive map and model of predator policy. This creates a tree structure (Fig. 1e) of virtual actions over the environment and their respective expected outcomes. This is somewhat similar to a chess player thinking through potential lines of play and counter-moves by an opponent. After each move by the prey and the predator, the prey re-planned, which resulted in flexible responses to changing predator location. Planning implemented in this study is based on Monte-Carlo tree search[23], a previously established efficient tree growth and search algorithm used within artificial intelligence approaches to games including AlphaZero[24]. Note that at 5000 states, assuming all four directions are available at each step, the planner can only evaluate six moves ahead, whereas the minimum number of moves to get to the goal is fourteen. Therefore the planner is likely finding policies that are locally optimal which may not be globally optimal.

**Performance in simple habitats while varying visual range**. In the pseudo-aquatic simulations of idealized predator-prey interactions (Fig. 1a), the increase in survival rate is proportional to visual range (Fig. 2a). The utility of planning, defined as the average change in survival rate across planning levels, is significantly higher for long visual ranges (Fig. 2b).

The two most important determinants of survival in open environments are: (1) distance at which the predator is detected—proportional to visual range—and (2) the number of times the predator is within view of the prey. Long visual range facilitates early predator detection, allowing the prey to react further away (Supplementary Movie 1). At these ranges, the prey's belief distribution (see Methods) is less diffuse than with short visual ranges (Supplementary Fig. 4), tailoring action choices to a smaller set of possible predator locations.

To analyze the behaviors that result from increased planning across visual ranges, we quantified the frequency of actions between linked cells, provided that the episode terminated with prey survival. The set of such action sequences are termed success paths. Interestingly, across all tested visual ranges, success paths are highly stereotypical and simple (Fig. 2c; Supplementary Movie 1). These emergent policies resemble a natural behavior to approach solid objects or boundaries—called thigmotaxis—commonly observed in terrestrial and aquatic animals in both laboratory[25,26] and naturalistic conditions[27,28]. These strategies most likely occur as a result of the prey trying to increase its distance from the predator while trying to get to the goal location. Notably, habit-based action selection that uses these success paths results in comparable performance to planning at 5000 states forward simulated (Fig. 2d; Supplementary Movie 2).

**Performance in environments while varying complexity**. Next we examined prey behavior in more cluttered environments. At midrange levels of environmental clutter (entropy 0.4–0.6), both the prey survival rate (Fig. 3a), and the utility of planning peaks (Fig. 3b). In environments that have either low (entropy 0.0–0.3) or high (entropy 0.7–0.9) levels of clutter, the environment and predator dynamics restricts both the survival rate (Fig. 3a) and the advantage gained from increased planning (Fig. 3b).

Performance and behavioral variability are affected by the spatial distribution of clutter. The spatial distribution of clutter—not captured by entropy—can be assessed using a quantitative measure termed visually occlusive spatial complexity (see Methods (Eqs. (3)–(4)); Fig. 3c), which is based on a network representation of cells connected by line-of-sight. The complexity of these networks, defined in terms of equivalence and diversity of the number of cells visible from a given cell, has two boundary conditions that are simple[29]: fully visible environments (open), and fully occluded environments. Between these edge cases, visually occlusive spatial complexity (hereafter spatial complexity)

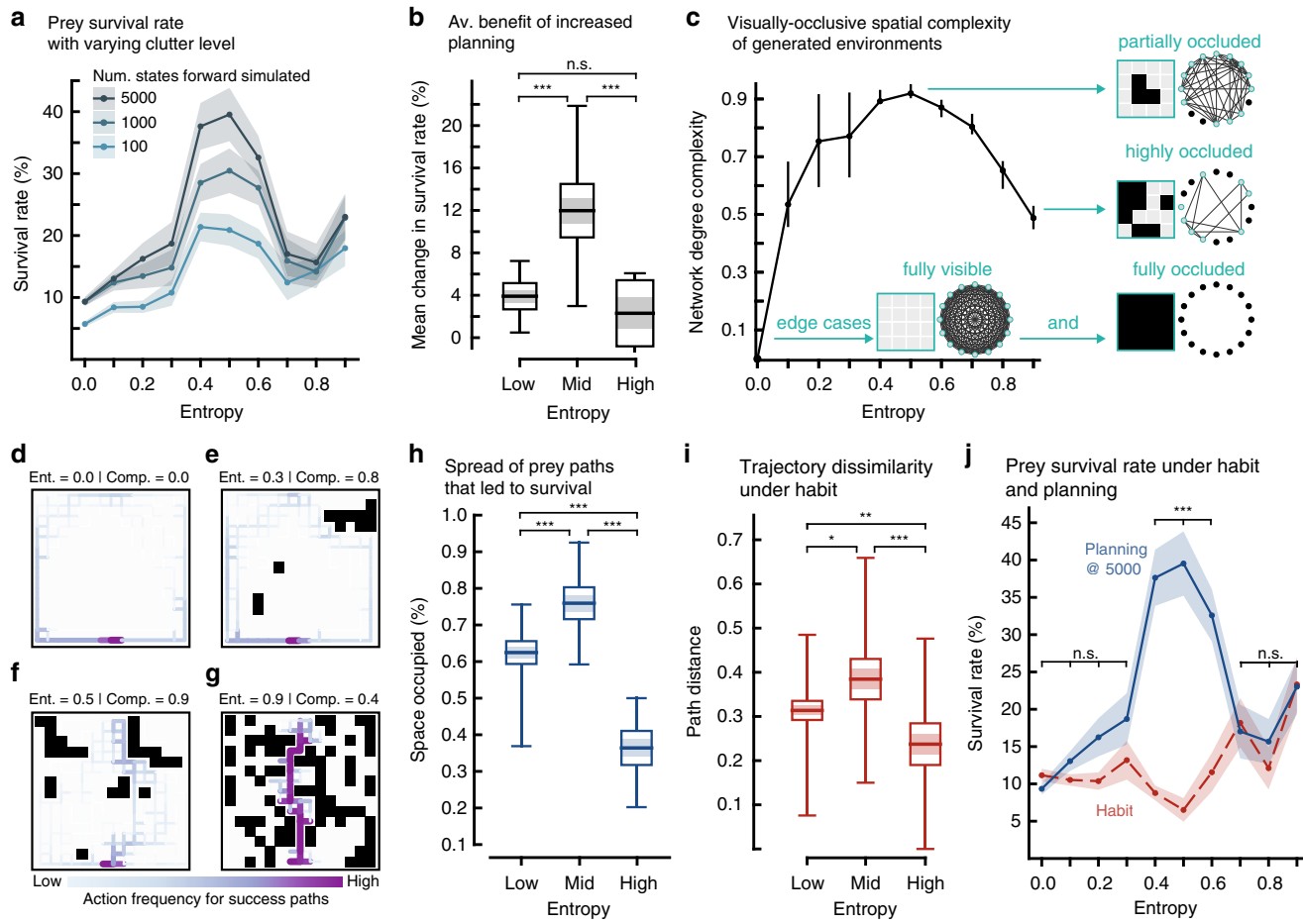

**Fig. 3 The utility of different behavioral controllers in habitats with varying complexity. a** Mean survival rate versus clutter density across random predator locations ($n = 5$), for a given planning level. The fill indicates ±s.e.m. across randomly generated environments ($n = 20$). **b** Mean change in survival rate across all planning levels shown in (**a**) calculated for Low: 0.0–0.3 entropy; Mid 0.4–0.6; High 0.7–0.9. Horizontal line: mean; Shaded region: ±s.e.m; Box: 95% confidence interval of the mean; Vertical line: range of the data. $n = 20$ for each entropy level; mean change in survival rate is averaged across tested initial predator locations ($n = 5$) and across entropy grouping. MWU tests (two-tailed) across entropy groupings (with Bonferroni correction): $U_{low,mid} = 39.0$, $U_{mid,high} = 51.0$, and $U_{low,high} = 142.0$, (***$p < 10^{-4}$; n.s. is not significant $p = 0.06$). (KW across entropy ranges $H = 24.7$, $p < 10^{-5}$). **c** Spatial complexity with respect to entropy. The line plot shows the mean complexity and the interquartile range ($n = 20$ generated environments). Insert shows example 4 × 4 environments and their corresponding visibility networks. **d–g** Heatmaps of all action sequences taken by the prey that resulted in prey survival at the maximum planning level (5000 states) in four out of the 200 environments examined, with color density proportional to frequency. Color bar action frequencies range from 1 to 68, dependent on entropy level. Survival paths are aggregated across all tested predator locations ($n = 5$) and total number of episodes ($n = 50$). For other examples see Supplementary Fig. 5. **h** Path spread denotes the percent of unique cells occupied by successful action sequences at the 5000 state planning level. Plot representation, and low, mid, and high entropy ranges as in (**b**) ($n_{low} = 76$, $n_{mid} = 58$, and $n_{high} = 46$ generated environments). MWU tests (two-tailed) across entropy groupings (with Bonferroni correction): $U_{low,mid} = 1109.0$, $U_{mid,high} = 246.0$, and $U_{low,high} = 595.0$ (***$p < 10^{-7}$). **i** Graph distance between action sequences that resulted in prey survival implemented by habit. Plot representation, and low, mid, and high entropy ranges as in (**b**) ($n_{low} = 70$, $n_{mid} = 41$, and $n_{high} = 32$ generated environments). MWU tests (two-tailed) across entropy groupings (with Bonferroni correction): $U_{low,mid} = 1069.0$, $U_{mid,high} = 322.0$, and $U_{low,high} = 717.0$ (*$p = 0.013$; **$p = 0.002$; ***$p = 0.0001$). **j** Mean survival rate for prey that uses habit (red dashed line). Representation as in (**a**). The planning data (blue solid line) as in (**a**) (±s.e.m. across randomly generated environments ($n = 20$), averaged across random initial predator locations ($n = 5$)). KW tests (two-tailed) for survival rate under habit- and plan-based action selection: low entropy (0.0–0.3) $p > 0.05$; mid entropy (0.4–0.6) $p < 10^{-4}$; high entropy (0.7–0.9) $p > 0.05$. Source data are provided as a Source Data file.

increases until midrange levels of clutter and then decreases for highly cluttered environments (Fig. 3c).

Similar to our analysis of behavior in pseudo-aquatic environments, we quantified behavioral variability in terms of action frequency and examined success paths for each environment. In spatially simple (low and high entropy) environments, success paths under planning are highly stereotyped (Fig. 3d, e, g, h; Supplementary Movie 3). Conversely, in spatially complex (midrange entropy) environments, the prey's survival strategy is variable (Fig. 3f, h). The spatial distribution of occlusions in these environments enables prey that plan to exhibit complex and

flexible behaviors that strategically deploy occlusions to escape from the predator, resembling hiding and natural diversionary tactics[30] (Supplementary Movie 4).

The suggested emergence of flexible behavior in spatially complex environments affects the success of habits. In low complexity environments, success paths under habits are highly stereotyped (Fig. 3i), resulting in their performing similar to planning at 5000 states forward simulated (Fig. 3j). In contrast, in high complexity environments, the high variability in success paths—the result of decisions specific to predator choices—results in significantly poorer performance under habits, as the prey

cannot re-valuate future actions based on changes in state (Fig. 3j; Supplementary Movie 2). When the prey visual range is restricted (1, 3, and 5 cells ahead) in environments that have the greatest path diversity (see Methods; Fig. 3h), performance is significantly degraded under plan-based action selection and is not significantly different from performance under habit-based action selection at long visual ranges (Supplementary Fig. 6a; two-tailed MWU test $U = 410.0$ $p = 0.28$). Notably, survival paths become stereotypical (Supplementary Fig. 6b, c), suggesting that complex behaviors such as hiding disappear (Supplementary Movie 1). Despite the emergence of complex behaviors at longer visual ranges in these same environments, the reduced visual range results in the predator frequently being out of view. Consequently, the prey often makes lethal errors as it cannot plan its actions based on the predator's precise location (Supplementary Fig. 7; Supplementary Movie 1). Thus, in highly complex environments, having long visual range—and thereby frequent and well-resolved updates on predator position—is necessary for the rapidly changing action values to reflect the actual adversary location, rather than reflecting an average across all the believed locations of the adversary as occurs in the shortened visual range condition.

These results suggest that emergent strategies for survival are dependent on environmental properties interacting with well-resolved dynamic contingencies. Non-habitizable complex environments may generate multiple viable futures, which might require planning to discover their diverse values.

**Spatial connectivity to arbitrate between planning and habit**. The stereotypy of prey success paths in simple environments allows the predator to follow a competing strategy that is similarly dependent on environmental properties. Complementary to our previous analysis, we quantified success paths for the predator by calculating the frequency of actions taken by the predator in episodes that resulted in prey capture (Fig. 4a, b, Supplementary Fig. 8). These resulting predator success paths resemble trajectories taken by primates in pursuit tasks[31] and seem to arise as a result of easy access to predicted prey locations.

Access and connectedness can be quantified using centrality measures on an equivalent graphical representation of environments (see Methods (Eq. (5))). One such measure is eigenvector centrality (eigencentrality (Eq. (6))), which in this case represents the weighted sum of direct connections (actions to and from a cell) and indirect connections of every length[32] (Fig. 4b). Graph theoretic measures in general have been used to describe neurobiological relational knowledge structures (e.g., hippocampal formation and the cognitive map)[33]. Such an approach allows for generalizations of emergent phenomena based on connectedness of an abstract representation. The emergent stereotypical prey policy in open environments is along poorly connected cells (correlation between cell eigencentrality and action frequency: Spearman $\rho_{mean} = -0.55$); while the emergent predator policy is distributed over highly connected cells to facilitate easy transitions between neighboring regions (Fig. 4b). Unlike open environments (Fig. 4b), which have a region of high eigencentrality that tapers away in all directions, spatially complex environments exhibit adjacent clusters of highly and poorly connected regions (correlation between spatial complexity and spatial clustering of eigencentrality (global Moran's I): Pearson $\rho = 0.61$, $p < 10^{-21}$) (Fig. 4d, e). In such environments stereotypical action sequences do not emerge.

We hypothesize that in environments with high eigencentrality clustering, planning becomes imperative during transitions from low to high eigencentrality and results in behavioral variability in regions of high eigencentrality (Fig. 4d–e). Preliminary support for this is found in the pattern of nonlocal hippocampal spatial representations that sweep in front of rodents at high-cost choice points[34,35] where there is a sharp change in eigencentrality (Fig. 4c). To examine this hypothesis, we implemented a hybrid controller that executed habit-based action selection in regions of low eigencentrality and switched to plan-based action selection when the prey was transitioning into and navigating a region of high eigencentrality.

In environments with low eigencentrality clustering (low complexity), behavioral control was rarely transferred over from habit to planning (Fig. 4f). Consistent with our previous findings (Fig. 3j), in these environments, all examined types of behavioral control strategies (habit, planning, and hybrid) performed similarly (Fig. 4g). In contrast, in environments that featured adjacent clusters of low and high eigencentrality (high complexity), our hybrid controller engaged planning more often (Fig. 4f; two-tailed MWU test $U = 361.0$, $p < 10^{-10}$) and resulted in survival rates that were not significantly different from those attained with full-time planning (Fig. 4g). The success of this hybrid strategy also agrees with prior research that suggests behavioral control should be transferred from habit to planning as uncertainty in state values increases[5]. Here, the prey's uncertainty increases in highly connected regions as a consequence of an increase in regional openness.

**Relating the generated environments to natural habitats**. The assumption of a detailed allocentric cognitive map (see Supplementary Note 1) and the limitation of actions to the 2-D plane has the consequence that for both agents, the 2-D randomly generated environments examined in this study (Fig. 1a, b) generate a 2-D perspective-independent representation of space through which these agents navigate. In extending these idealized 2-D environments out of the plane as a first step toward real environments, it is important to note that the planar motion restriction in our study implies that the 3-D height distribution of occlusions will not influence the planning process or the consequent action choice so long as the occlusions break the line of sight. Consequently, approaches to quantifying habitat spatial patterning based on a top-down orthographic projection, provided that the structures being projected down break the line of sight of the interacting animals, will provide a reasonable way to relate our 2-D environments to natural habitats for animals that are mostly confined to the ground or stay near to biogenic structure on the substrate of water bodies as do many fish.

Lacunarity[36], a measure commonly used by ecologists, quantifies the spatial heterogeneity of gaps that arise from, for example, spatially discontinuous biogenic structure such as grassland between wooded forests. It is typically based on a top-down projection of space. Lacunarity has been used to study the effects of habitat spatial patterning on ecological processes, such as the dispersion of predators and prey, and to characterize a wide range of aquatic and terrestrial habitats (Supplementary Table 1; Supplementary Methods 1). Based on this analysis, coastal aquatic environments such as those with small patches of seagrass have average lacunarities above 1.16 ($n = 5$), while flat seabeds or midwater aquatic conditions that have no structure approach infinite lacunarity. In contrast, highly cluttered aquatic environments such as those containing large patches of seagrass, salt marshes, and coral reefs result in average lacunarities between 0.12 and 0.49 ($n = 16$). Terrestrial environments, featuring both largely open environments such as herbaceous rangelands and highly cluttered environments such as dense forests, have average lacunarities between 0.23 and 1.35 ($n = 9$). We computed the lacunarity of the randomly generated environments we have examined (see Methods), and used these ranges to categorize

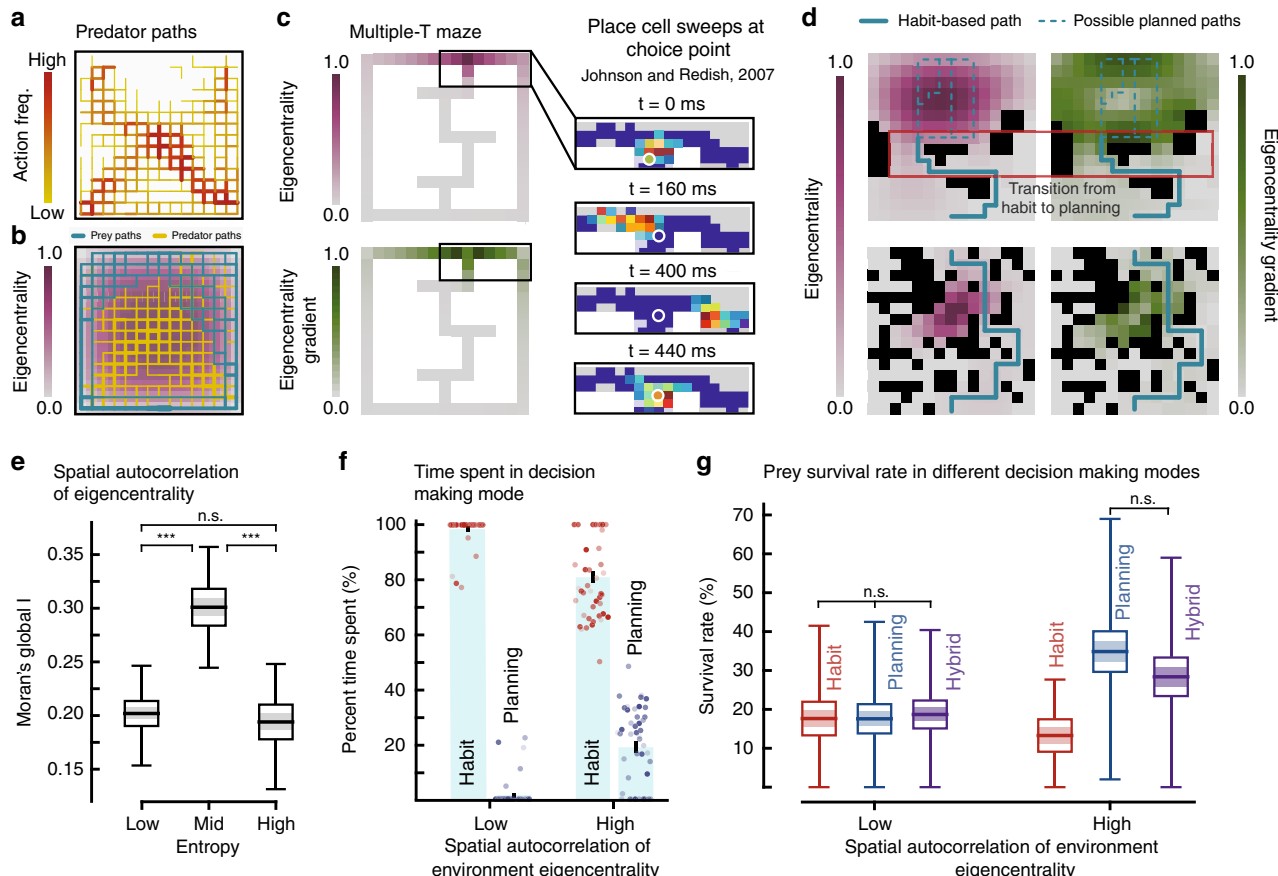

**Fig. 4 Using spatial connectivity to arbitrate between behavioral controllers. a** Heatmap of all predator paths that resulted in prey capture. Color bar: 1–87. **b** Eigencentrality of an open environment overlaid with representative prey success paths in teal, and prey capture (see (**a**)) in orange. Line thickness proportional to action frequency. **c** Multiple-T maze overlaid with eigencentrality and eigencentrality gradient. Color density proportional to the eigencentrality and eigencentrality gradient of each cell of the quantized maze. Johnson and Redish[34] showed that the neural representation of place (reconstruction on the right) moved ahead of the animal while it paused at the choice point. **d** Example environments (top row entropy = 0.5, bottom row entropy = 0.9) and their eigencentralities and eigencentrality gradients (see Supplementary Fig. 9 for prey success paths). Transition regions (red box) from low to high eigencentrality and behavioral control based on change in gradient and value of eigencentrality (see Supplementary Fig. 10 for other transition region examples). **e** Mean spatial autocorrelation (global Moran's I) of the environment eigencentrality. Horizontal line: mean; Shaded region: ±s.e.m; Box: 95% confidence interval of the mean; Vertical line: range of the data. For all entropy levels $n = 20$. MWU tests (two-tailed) across entropy groupings (with Bonferroni correction): $U_{low,mid} = 15.0$, $U_{mid,high} = 13.0$, and $U_{low,high} = 178.0$, (***$p < 10^{-6}$; n.s. is not significant $p = 0.28$). **f** Average percent time spent in decision making regime (habit vs planning) when environments are grouped based on their spatial autocorrelation of eigencentrality ($n_{low} = 50$: bottom 25%, $n_{high} = 46$: top 75%). The error bars indicate ±s.e.m of percent time spent. **g** Survival rate for a prey that uses planning (blue), habit (red), and hybrid control (purple) based on environment eigencentrality (see (**d**); see Methods). Environment grouping as in (**f**) ($n_{low} = 50$, and $n_{high} = 46$), and representation as in (**e**). Two-tailed KW test for survival rate in environments with low eigencentraliy clustering $H = 0.952$, $p = 0.62$. Two-tailed MWU test with Bonferroni correction for survival rate under hybrid and plan-based action selection in environments with high eigencentrality clustering $U = 820.5$, $p = 0.09$. Source data are provided as a Source Data file.

them into one of three major groups: coastal aquatic, terrestrial, and structured aquatic (Fig. 5a; Supplementary Table 1).

As we do not have the source data to compute the spatial complexity of the measured natural environments directly, we instead indirectly estimated this quantity by categorizing the generated environments into these three natural habitat domains via their lacunarity. We used the spatial complexity of each category as an approximation of the true spatial complexity of the corresponding natural habitat. Assuming gaps are transparent to vision, both coastal aquatic and highly structured aquatic environments like coral reefs have lower spatial complexity than terrestrial environments (Fig. 5b).

These results indicate that terrestrial environments advantage planning (Fig. 5c) in non-habitizable contexts and facilitate the emergence of complex behaviors. Notably, the subset of terrestrial

environments where planning outperforms habit (green circle, Fig. 5d)—due to the emergence of flexible behavior—feature clustered open and closed spaces not unlike savannas. In contrast, there is unlikely to be a significant advantage to planning in aquatic environments that are either largely open or closed (low and high entropy environments) where a habit-based approach is successful (Fig. 5d).

## Discussion

We show that in simulated aquatic environments similar to a flat seabed with small patches of seagrass or highly structured aquatic habitats such as salt marshes or coral reefs, the simplicity of the environment generates stereotypical strategies. This allows habit-based strategies to succeed in dynamic tasks. This conclusion is buttressed by ethological analyses of predator-prey dynamics in

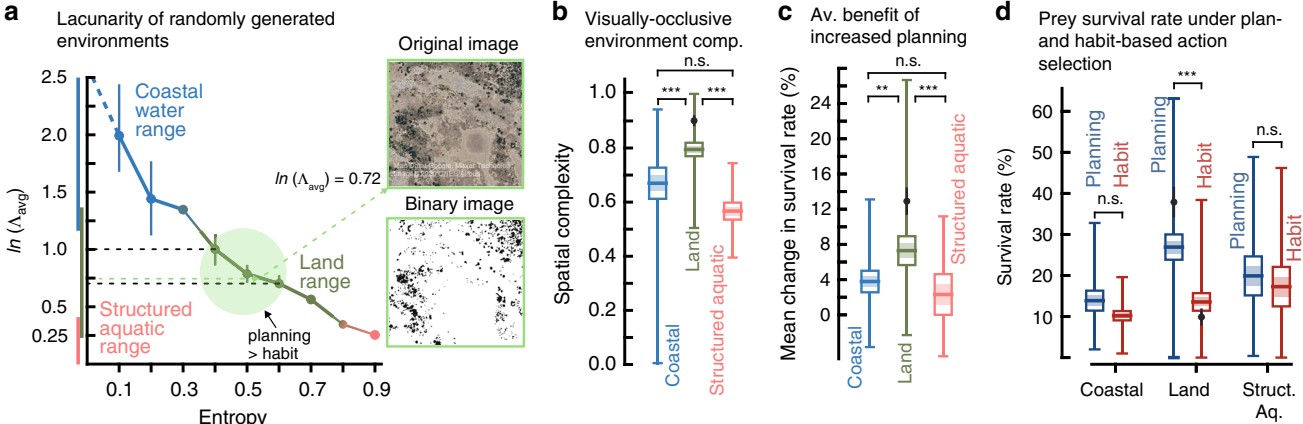

**Fig. 5 Comparison of generated environments to natural habitats. a** Distribution of lacunarities of generated environments (see Supplementary Methods 1). The line plot shows the mean natural log of average lacunarity (see Methods) and the interquartile range ($n = 20$ at each entropy level). Coastal: blue line $\geq 1.16$, $n_{coastal} = 70$; Land: green line 0.23–1.34, $n_{land} = 139$; Structured Aquatic: pink line 0.03–0.41, $n_{structured\ aquatic} = 42$ (Supplementary Table 1). The green circle highlights a zone of lacunarity where planning outstrips habit (based on Fig. 3j). Insert shows an example image from the Okavango Delta in Botswana ($\approx 800$ m $\times$ 800 m, from Google Earth), and its average lacunarity ($\ln(\Lambda_{avg})$). The Okavango is considered a modern analogue of the habitats that early hominins lived within after branching from chimpanzees[52]. For additional images and their corresponding lacunarity plots see Supplementary Fig. 11. **b** Spatial complexity (see Fig. 3c) of generated environments grouped by natural environment bands. Horizontal line: mean; Shaded region: $\pm$s.e.m; Box: 95% confidence interval of the mean; Vertical line: range of the data ($n_{coastal} = 70$, $n_{land} = 134$, $n_{structured\ aquatic} = 41$ generated environments). MWU tests (two-tailed) across environment groupings (with Bonferroni correction): $U_{coastal,land} = 2103.5$, $U_{land,\ structured\ aquatic} = 666.5$, and $U_{coastal,structured\ aquatic} = 1405.0$, (***$p < 10^{-10}$; n.s. is not significant $p = 0.43$). **c** The incremental benefit of planning (see Fig. 3b) of generated environments grouped by natural environment bands. Plot representation as in (**c**). ($n_{coastal} = 70$, $n_{land} = 139$, $n_{structured\ aquatic} = 42$ generated environments). MWU tests (two-tailed) across environment groupings (with Bonferroni correction): $U_{coastal,land} = 3631.0$, $U_{land,\ structured\ aquatic} = 2001.0$, and $U_{coastal,structured\ aquatic} = 1263.5$, (**$p = 0.001$; ***$p = 0.001$; n.s. is not significant $p = 0.08$). **d** Survival rate for a prey that uses planning (blue) and habit (red) of generated environments grouped by natural environment bands. Plot representation as in (**b**). ($n_{coastal} = 70$, $n_{land} = 139$, $n_{structured\ aquatic} = 42$ generated environments). KW tests (two-tailed) for survival rate under habit- and plan-based action selection: $H_{coastal} = 3.29$, $p = 0.07$; $H_{land} = 40.7$, $p < 10^{-9}$; $H_{structured\ aquatic} = 1.16$, $p = 0.3$. In (**b**), (**c**), and (**d**) black dot and vertical bar overlay indicates mean $\pm$ s.e.m. of the variable of interest in environments where planning outperforms habit (green shaded region in (**a**)). Source data are provided as a Source Data file.

open water or on open ground, which typically concern habitual and stereotyped responses. In aquatic contexts, stereotypical responses are mediated by a specialized neuron called the Mauthner cell that reduces the latency of an escape response[37]. Interestingly, this circuit disappears in the vertebrate line after amphibians[37], possibly related to the lowered importance of short range reactions once fully on land. The stereotypical strategy observed in both pseudo-aquatic and pseudo-terrestrial environments is following the boundaries of the space (Fig. 2c, Supplementary Movies 1 and 3). Similarly, in studies of live animals in both aquatic and terrestrial contexts, retreating from exposed areas to edges after detecting increased predation risk is well documented[28] and is used as an assay of anxiety in laboratory experiments[25,26]. Although the vast majority of the underwater predator-prey literature concerns reactive stereotyped behaviors, there are rare exceptions, such as cooperative hunting between moray eels and groupers[38] and triggerfish hunting behavior[39].

Our analysis is contingent on a large number of idealizations of visually guided predator-prey interactions, a nuanced and highly variable domain (see Supplementary Note 2 and 3). Nonetheless, in simple environments the complementarity of predator and prey strategy found in nature emerges out of the simulations, shown by the negative eigencentrality-taxis of prey and positive taxis of predator (Fig. 4b).

In complex environments, we observe complex behaviors that arise as a consequence of a need to deliberate over multiple viable futures (Supplementary Movie 4). Within ethology the use of occlusions to gain advantage during stalking and pursuit is well known in Carnivora[40,41], though more common in felids than canids[42]. Moreover, similar to mammalian hunters, birds of prey are able to predict their prey's likely paths of egress to

preemptively block and ambush the prey[43] using occlusions such as trees. This particular behavior is similar to luring strategies employed by our simulated prey. Given the strategic use of occlusions by the simulated prey in our environments, we would expect similar behaviors to emerge had we simulated planning in the predator. The response of prey to predator pursuit is relatively less well measured, often described as fleeing behavior toward areas of cover interspersed with concealment via freezing. While freezing was not an action our prey could take, its functional role in concealment[44] was observed in our simulations as back-and-forth oscillations behind a blocking occlusion (Supplementary Movie 4).

We show that spatially complex environments have clusters of low and high spatial connectedness. In these environments, arbitrating between habit- and plan-based action selection based on spatial connectivity results in no survival rate penalty compared to planning continuously. Vicarious trial and error, thought to reflect planning in rodents[11], similarly occurs at transitions into highly connected regions[34,35].

Prior research suggests that an increase in theta coherence between the hippocampus and PFC is needed to sort through options as the hippocampus imagines potential outcomes[11,14]. Interestingly, coherence between the hippocampus and PFC increases when rodents transition from a closed arm (poorly connected) to an anxiogenic open area (highly connected)[45]. This may portend a mechanism for switching decision making modes driven by anxiety as a proxy for spatial connectivity.

It should be noted that eigencentrality has broad applicability to patterns of connectivity outside of spatial contexts. Extended action sequences that require planning include tool making, non-spatial two-step decision tasks[46], and social decision making in

primate troops. As eigencentrality has been used to examine social grooming in primates[47], one possibility is that the circuitry for spatial planning evolved first and was later exapted for non-spatial uses.

Given the strong evidence concerning early expansions in brain size of mammals[48], potentially related to navigation via olfactory cues[49], it seems likely that planning first evolved in situations that are not as continuously dynamic as visually guided predator-prey interactions. Prior research has shown that there is a transition from plan-based to habit-based action selection after a novel set of contingencies arise but subsequently stabilize[5,11]—habitizable scenarios. An example of a habitizable use of planning would be an olfaction-dominant animal entering a new territory with relatively stable threats (e.g., the den of a predator) or opportunities (e.g., an insect nest) (Supplementary Fig. 2b, e). After updating the cognitive map, the planning system would be used initially for devising paths that avoid threats or result in faster access to the opportunities. After some time these action sequences would shift to habit-based control.

Finlay and co-workers have discovered an inverse correlation between the size of limbic (LI) (olfactory bulb, olfactory cortices, amygdala, hippocampus, and septum) and isocortical (IS) components of the telencephalon[50] that may be related to the importance of habitizable vs. non-habitizable planning in a given species. Jacobs[49] suggests that for animals whose foraging strategy involves detection rather than prediction (roughly similar to the habitizable case), the dependence on a cognitive map leads to an enlargement of LI components, and planning can be supported by an allocentric map calibrated by landmarks detected in the near field (e.g., olfactory cues or short range vision). Animals hunting more encephalized prey, which are predictable but difficult to capture, require planning and larger multisensory IS components[49].

Thus, one possibility is that after the rise of habitizable planning in ancestral endotherms, non-habitizable planning was selected for in patchy terrestrial environments using rapidly changing distal cues interrogated by vision. This may inform the emergence of visually dominant birds with reduced olfactory bulbs[51], visually dominant simians, and terrestrial carnivores which hunt with long range vision and olfaction[40–42]. Simians show a reduction in LI and an increase in IS[49,50]. The origin of bipedality in hominins, whose extant member excels in plan-based action selection, has been related to their exploitation of landscapes of high habitat diversity including open grasslands[52]. A modern analogue of these early hominin environments is the Okavango Delta[52], samples of which exhibit lacunarity favoring planning (Fig. 5a, Supplementary Fig. 11). In terrestrial carnivores, a more balanced increase in both IS and LI occurs as they use long range vision and olfaction during hunting[49,50].

We speculate that the reduction in LI, whose overall volume is dominated by the hippocampus, in high acuity mammals could be related to the simultaneous apprehension of distal landmark information, thereby allowing planning over the changing sensorium (rather than cognitive map only). This would be supported by recurrent isocortical connections particularly between the PFC and hippocampus. This contrasts with the picture emerging in teleosts, where an allocentric map[53] may be generated from a sequence of landmark encounters detected at short range, via a diencephalon to dorsolateral pallium (putative hippocampal homolog) circuit[54,55]. How such a map is used in teleosts is unclear, but it has been noted that there are multiple uses of a cognitive map outside of planning[56,57]. Given our results suggesting that planning has little to no advantage in water in the context of predatory interactions, non-planning uses of a cognitive map may help inform the presence of maps in fish[39].

Similarly, for reptiles there is evidence for cognitive maps but less evidence for planning[53,58]. In this case, however, as many are terrestrial and have good vision, this group may signal an additional constraint on planning that mammals and birds have overcome. In particular, planning requires exponential computation time with the number of steps into the future being considered, while habit requires constant computation time (see Supplementary Note 4). The ability to plan for stationary (habitizable) or dynamic rewards (non-habitizable) may therefore be correlated with absolute brain size (similar to what has been found for self-control[59]) and endothermy-related increases in the speed of neural computation[60] (see Supplementary Discussion 1).

We suggest that imaging systems—vision and echolocation—play a special role in spatial planning. Aquatic echolocation in whales and dolphins has significant range advantage over vision (see Supplementary Discussion 2). Unlike mechanosensation, audition, and olfaction, imaging systems are able to detect clutter, and other variables with high temporal and spatial resolution necessary for dynamic contexts. Such passive detection of objects may be important for the calibration of the cognitive map and may be why rodents require visual cues in maze tasks with learned olfactory cues[49]. Notably, nocturnal visual range typically exceeds diurnal aquatic range—Supplementary Fig. 1—but requires reduced resolution to gain sensitivity[61]. The inference that nocturnality equates to reduced importance of vision in early mammalian evolution, and extant nocturnal species, may therefore be suspect. While our contribution concerns dynamic planning in vertebrates, vision seems to also be important in the two invertebrates with the best evidence of planning, jumping spiders and cephalopods. These animals have visual acuity far beyond what is typical within this group[62] (see Supplementary Discussion 3).

Parker has suggested that the origin of the Cambrian explosion lies in the atmosphere or oceans of the period gaining higher transparency to sunlight, triggering the evolution of the first image-forming eye and sparking a predator–prey evolutionary arms race that gave rise to the Cambrian's profusion of animal forms[63]. A second great change in transparency occurred with the emergence of fish on to land, which gave rise to a sensorium large enough to fit multiple futures. Our idealized model of spatial planning during predator–prey interactions suggests that there may be a link between the enlarged visual sensorium and habitat complexity of terrestrial animals and the evolution of neural circuits for dynamic planning.

## Methods

**Simulation 1: Pseudo-aquatic.** A virtual prey and predator act in an empty $15 \times 15$ discretized environment (Fig. 1a). The prey either used habit-based or plan-based action selection with a predetermined number of states to forward simulate. We assumed that the prey and the predator had previously learned the environment (where the boundaries and occlusions are; more on the perfect map assumption in Supplementary Note 1). As the main acting agent, the prey was initialized with an environment and predator model, which allowed the prey to forward simulate the actions of the predator. The prey's aim was to reach the goal location, and the predator's aim was to reach the prey before the prey reached the goal location. However, the predator was not privy to the prey's aim and therefore did not explicitly associate any location within the environment as a goal for the prey. An episode terminated if the prey reached the goal location (survival), if the predator reached the prey location (death), or if the number of steps exceeded the maximum number of steps allowed for an episode (200). For these simulations the number of steps before termination was $19 \pm 6$ (mean ± std).

The predator was designed as a reflex agent that retained a memory of prey location. The predator selected actions based on the policy: aggressively pursue the prey with 75% probability, and act randomly with 25% probability. The predator was on average 1.5× faster than the prey (moved two cells with 50% probability), which fell within typical terrestrial predator speeds relative to prey of 1.2–2[64,65]. The predator observed the entire environment, and therefore knew the location of the prey at all time points (rationale for the predator having larger visual range: see Supplementary Note 2). During aggressive pursuit, the predator chose actions that minimized the Euclidean distance between itself and the prey (if there was more

than one action that met this criteria, then an action from this set was chosen at random). During random action selection, if the prey was within the reach of the predator, the predator chose actions that terminated the episode with prey capture. Otherwise, the predator chose a random action that kept it within the confines of the environment. The predator spawned at a random initial position exclusive of the $3 \times 3$ region surrounding the prey, and the goal location.

For an overview of all general parameters see Supplementary Table 2.

**Plan-based action selection.** We formulated planning as a partially observable Markov decision process (POMDP) consisting of the following variables[66]: a set of states $\mathcal{S}$ (prey and predator spatial location), a set of observations $\mathcal{O}$ (0 if the predator is not observed, cell number corresponding to predator location if the predator is observed), a set of actions $\mathcal{A}$ (cardinal directions: North, East, South, West), a list of action-observation pairs that constitutes the history $h$, a belief state $\mathcal{B}(s, h)$ specifying the probability distribution of the prey being in a state $s$ given history $h$, a reward function $\mathcal{R}(s)$ defined as the expected immediate reward for a given state $s$, and discount factor $\gamma = 0.95$[23] that attenuates distal rewards.

Here, the prey's aim was to estimate $\mathcal{V}^*(h)$ (expected total optimal future reward) by using its environmental model (further discussed below: Algorithm of plan-based action selection). Forward simulation of future states was implemented through construction of a tree that simulated possible actions until a termination condition was reached (Monte-Carlo tree search adapted for POMDPs (POMCP)[23]). Within this, the prey internally simulated its own actions, the reactionary actions of the predator, and the corresponding observations and rewards. These internal simulations are used to approximate the value function without explicitly calculating it.

One time step consisted of: (1) The prey choosing an action with a softmax decision maker[21] $a \in \mathcal{A}$ based on its estimation of $\mathcal{V}^*(h)$. If that action brought the prey to the goal then the episode terminated (survival); (2) The predator choosing an action $a \in \mathcal{A}$ based on its pursuit policy. If that action brought the predator to the prey position, then the episode terminated (death); (3) The prey receiving an observation $o \in \mathcal{O}$ and reward $r = \mathcal{R}(s)$ from the environment; (4) The prey adding the action it chose and the corresponding observation to its history $h$; (5) The prey updating its belief state $\mathcal{B}(s, h)$ based on current history $h_t$; (6) The prey planning (internally constructing a planning tree) until a fixed number states were forward simulated.

**Vision and partial observability.** The prey had a pre-specified visual cone that faced the direction of motion and extended outward 1–5 cells ahead (fixed at a single value per trial) (Fig. 1a). The prey always knew its own location within the environment, but only knew the predator location if the predator was inside of the prey's visual cone. If the predator was outside of the prey's visual cone, the prey sampled a predator location from its belief state ($\mathcal{B}(., h_t)$) proportionate to the belief state distribution (Supplementary Fig. 4); for example, if the predator is believed to be at cell (8, 12) with 90% probability, then 9 out of 10 draws from the distribution would be (8, 12). The number of samples (predator locations) the prey drew from its belief state was equal to the number of states the prey forward simulated[23]. For each of these samples, the prey constructed a planning tree (Fig. 1d–e).

At the start of an episode, the prey rotated its visual cone 360° to inspect the surrounding region. If the prey observed the predator during this initial sweep then it knew the predator's initial location, which conferred knowledge about the initial state. Otherwise, the prey's belief state was initialized to all possible predator locations outside of the prey's visual cone with equal probability. The rotation of the visual cone independent of the acting direction was conducted only in the initial time step ($t = 0$). At the start of plan-based action selection, $K$ particles were selected from this initial belief state[23]. Until the first observation, the prey's belief state was propagated based on the prey's model of the predator's movement; if the predator was then expected to be within the visual cone, but was not, then the set of belief states was correspondingly pruned (Supplementary Fig. 4). This process of propagation and pruning was repeated for all $K$ particles. In between two observations, the prey's belief state consisted of all the possible places the predator might have moved given the location of the predator at the time of observation and the prey's model of the predator's action-selection policy. In cases of particle deprivation at large $t$'s, particle reinvigoration was performed. Across all the states, $M$ new particles that would satisfy the $(a_t, o_t)$ pair were added to belief state (i.e. if the prey had not observed the predator after taking an action, $M$ other states that were unobservable from the prey's location and were not in the belief state were added to the belief state)[23]. As with prior practice[23], $K$ was set to be the number of states forward simulated, and $M$ was set to be 1 or the number of states forward simulated divided by 10, with a maximum of 50,000 attempts at adding new particles.

**Algorithm of plan-based action selection.** Planning was implemented through construction of a tree[23,66], which relied on a previously learned model of the environment (boundaries and predator model). After an observation $o \in \mathcal{O}$ was received and a state sampled from the belief state $\mathcal{B}(., h_t)$, the prey began planning from its current history $h_t$ to estimate the optimal value function $\mathcal{V}^*(h)$. Each node in the search tree, denoted by $T(h)$, had three elements associated with it: $\mathcal{B}(h)$

specifying a set of possible predator locations (which converges to a single state when the prey observed the predator), number of times a specific history $h$ has been visited ($N(h)$), and the expected value of an action and corresponding observation ($\mathcal{V}(h)$). $\mathcal{V}_{\text{init}}(h)$, and $N_{\text{init}}(h)$ are initialized to 0 for new nodes.

Planning tree construction and node value estimation in POMCP is divided into two stages: a tree-search policy that is on nodes with non-zero visit values (within-tree-search), and a rollout policy for nodes that have not previously been visited. After the evaluation of a state (predator and prey location), the node containing the first new history visited in the second stage is added to the search tree (Fig. 1d, e). The planner uses partially observable UCT (PO-UCT) during the first stage within-tree-search, which selects actions based on upper confidence bound (UCB1)[67]; and a uniform random rollout policy during the second stage. If a node has all of its children expanded (i.e., all the next states have values from a state $s_t$) the tree policy selects an action based on the in-tree policy, otherwise, the rollout policy is used to select actions. After the tree is expanded, the process is repeated from the root node. One iteration consists of: (1) The prey selecting a state from its belief state $\mathcal{B}(h)$, (2) Selection of child nodes from the root node based on the selected state using within-tree policy, (3) Expansion of a child node, (4) Rollout until termination condition, and (5) Backpropagation of values through the tree based on the rollout result. Therefore, the number of states the prey samples is equal to its number of states forward simulated (e.g., if the prey can forward simulate 100 states ahead, then it samples 100 predator locations from its belief state). As the search tree is constructed the set of sample states encountered during simulation for each node is stored. PO-UCT has been proven to converge to the optimal value function[23], which implies that as the number of states to be forward simulated increases, an action that is selected based on the search tree nears the optimal action to perform. At 5000 states forward simulated, however, it is unlikely we achieve global optimality.

After an action $a_t$ is selected with a softmax decision maker[19,21] and an observation $o_t$ is received from the environment, the planning agent's history is updated to reflect the new sample $\langle a_t, o_t \rangle$. The start node of the search tree and the associated belief state is updated to reflect the current history. The rest of the tree is pruned, since all other simulated histories are no longer representative of possible futures. Initial values and parameters are provided in Supplementary Table 3.

**Algorithm of habit-based action selection.** To model habit-based action selection (Fig. 1c), we implemented a variant of the PRQ-Learning algorithm[22]. For each visual range, a policy library $L = \{\Pi_1, \ldots \Pi_n\}$ was created based on the prey success paths—prey going from the initial position to the goal without being captured. If the prey was able to see the predator at the outset, then the policy library was initialized to be all success policies specific to that visual range and initial predator location. Otherwise, if the predator was outside of the prey's visual cone to start with, then the policy library was initialized to be all policies for that visual range across all simulated predator locations unseen from the prey's starting position. These success paths were taken from policies implemented by the prey using plan-based action selection at 5000 states forward simulated.

Following the PRQ-Learning algorithm, a policy $\Pi_k \in L$ was chosen by a softmax decision maker:

$$P(\Pi_k) = \frac{\exp(\tau W_k)}{\sum_{p=0}^{n} \exp(\tau W_p)} \quad (1)$$

where $W_k$ is the reuse gain of implementing the chosen policy, and $\tau$ is the temperature parameter. Initially all policies in the library were given zero weight. During policy implementation the prey did not deviate from the prescribed action sequence. After the implementation of the chosen policy $\Pi_k$, if the episode resulted in survival (prey reaching goal) the reuse gain ($W_k$) was weighted by the total discounted reward $R$, and the number of times the policy $\Pi_k$ had been chosen ($N_k \leftarrow N_k + 1$): $W_k \leftarrow \frac{W_k N_k + R}{N_k + 1}$, and the temperature parameter $\tau$ was updated to $\tau \leftarrow \tau + \Delta\tau$ by the decay parameter $\Delta\tau$. On the other hand, if the episode resulted in prey death (prey capture by the predator) $N_k$, $W_k$, and $\tau$ were not updated. Initial values and parameters are provided in Supplementary Table 4. The predator action-selection policy was the same as the one implemented in the planning task.

**Statistics and reproducibility.** A total of $n = 20$ random predator locations per visual range (1–5) and per number of states forward simulated (1, 10, 100, 1000, 5000) were used. Survival rate was calculated over 100 episodes for a given predator spawn location, prey visual range, and number of states the prey could forward simulate.

In Fig. 2b, the incremental benefit of planning is defined as the average difference in survival rate between the tested 1, 10, 100, 1000, 5000 states forward simulated (e.g., average of survival rate at 1000 minus survival rate at 100, survival rate at 100 minus survival rate at 10, etc.) for a given visual range. Due to a non-uniform increase from planning at 1000 states forward simulated to 5000 (difference is not 1 when converted to log ), a linear relationship was assumed, and the calculated difference was multiplied by 2.

Data analysis was done using Python 3.7.4. Statistical analysis was done using the 'numpy' (v1.17.2) and 'scipy' (v1.3.1) packages. Videos from raw episode files were created in Matlab (raw episode files and Matlab created videos available in

Source Data folder). All significance indicators follow: n.s. is not significant $p \geq$ 0.05; *$p < 0.05$; **$p < 0.01$; ***$p < 0.001$.

**Simulation 2: Pseudo-terrestrial.** A virtual prey and predator act in a $15 \times 15$ discretized environment featuring randomly added clutter with controlled density (Fig. 1b). A total of $n_{\text{env}} = 20$ random environments, with $n_{\text{entropy}} = 10$ levels of clutter—quantified by environmental entropy (see below Environment generation with randomized clutter)—were generated. The predator and prey model used for this experiment were the same as Simulation 1. The prey's plan-based action selection was based on the algorithm described in Simulation 1: Plan-based action selection.

The predator spawned at a random initial position exclusive of prey start location, goal position, and occlusions. A trial terminated if the prey reached the goal, the predator moved to the prey location, or if the episode reached the termination condition of 200 steps.

The sequences of computations that occurred in one time step were the same as Simulation 1.

**Vision and partial observability.** Unlike Simulation 1, the prey could see the entire environment except where blocked by occlusions (Fig. 1b). If an occlusion existed on the ray (Bresenham's line algorithm[68]) between the predator and the prey, the prey samples states from its belief state ($\mathcal{B}(., h_t)$) (see Algorithm of plan-based action selection). Initially, if the predator was not observed by the prey (behind an occlusion), the prey's belief state was all possible locations that were unobservable from the given prey location with equal probability. Until the first observation, the prey's belief state was propagated and then pruned based on the prey's model of the predator movement and locations that were hidden from the prey's position. In between observations, the prey's belief state constituted all the possible places the predator might have moved (based on the prey's model of predator movement) given the location of the predator at the time of observation and all the locations that were not visible from the prey's position.

The existence of occlusions impeded both the prey's and the predator's line of sight. Therefore, the predator knew the exact location of the prey if an occlusion was not present on the ray between the predator and the prey. The predator kept track (had memory) of the prey location while the prey was in view. When the prey was hidden, the predator propagated the prey's last known location randomly within the environment (exclusive of occlusions) to form a belief state. During aggressive pursuit, if the prey was within view, the predator used the actual prey location to choose an action that minimized the Euclidean distance. If the prey was hidden, the predator aimed to minimize its Euclidean distance to a single sampled state, selected proportionate to the predator's belief state distribution of possible prey locations. During random action selection, the predator chose a random action that did not move the predator to an occlusion or outside of the environment.

For Supplementary Fig. 6, similar to Simulation 1, the prey had a pre-specified visual cone that faced the direction of motion. However, for this experiment we limited visual range to 1, 3, or 5 cells ahead (fixed at a single value per trial). The prey only knew the predator's location if the predator was inside the prey's visual cone. Unlike Simulation 1, the prey's visual cone was cut by the presence of an occlusion, therefore the prey could not see locations that were blocked by an occlusion even if that location was inside the prey's visual cone. Similar to the full vision condition, for all locations inside the prey's visual cone, if an occlusion existed on the ray from that location to the prey's location, that location was deemed invisible (Supplementary Fig. 7). Initially, if the predator was not observed by the prey, the prey's belief of predator location was a uniform probability across all possible locations (other than occlusions and goal position) that were outside of the prey's visual field (visual cone minus invisible locations within the visual cone). Similar to the unlimited vision condition, and Simulation 1, the prey's belief state was pruned and propagated according to the prey's model of the predator when the predator was out of view.

**Algorithm of habit-based action selection.** The backbone of the algorithm (e.g., weighting and choice of policy from the policy library) was kept the same as Simulation 1. For each environment all the action sequences that led to prey survival during plan-based action selection were pooled together. If initially the prey was able to observe the predator, the policy library was pruned to reflect action sequences that were generated during plan-based action selection for that particular predator location. If the predator was not initially in view, the prey used the aggregate policy library after removal of all policies in which the predator was initially visible.

**Environment generation with randomized clutter.** The entropy of a general $m \times n$ discretized environment was calculated by treating the discretized environment as a binary matrix, where 1's represent occlusions, and 0's represent unoccupied

cells. The entropy of such an environment ($H(X)$) can be formally written as:

$$H(X) = -\frac{\sum_{i=1}^{m}\sum_{j=1}^{n}\mathbf{I}(x_{i,j}=0)}{mn}\log\left(\frac{\sum_{i=1}^{m}\sum_{j=1}^{n}\mathbf{I}(x_{i,j}=0)}{mn}\right)$$
$$-\frac{\sum_{i=1}^{m}\sum_{j=1}^{n}\mathbf{I}(x_{i,j}=1)}{mn}\log\left(\frac{\sum_{i=1}^{m}\sum_{j=1}^{n}\mathbf{I}(x_{i,j}=1)}{mn}\right) \qquad (2)$$

where $x_{i,j}$ refers to the value at row $i$ and column $j$.

In generating the occlusions for the environment, we assumed a random walk policy of random length that started at an unoccupied random position. The number of random walks performed could at least be 1 and at most be the number of occlusions for a given entropy, here denoted as $k$. The total number of random walk lengths $l$ must equal to $k$: $\sum_{1 \leq l \leq k} l = k$. This process was repeated if a path from the fixed prey position to the fixed goal position did not exist.

**Environment complexity analysis.** If an occlusion existed on the ray between the predator and the prey, both the prey and the predator were hidden from each other. By using the above principle we created a visibility network $G_v = (V_v, E_v)$ for all randomly generated environments. Vertices in this visibility network represented individual cells in the environment. An edge $e_{i,j}$ exists between two vertices $\{v_i, v_j\}$ if an occlusion does not exist on the line between the two vertices, which is based on the same visibility ray used in determining the prey's and the predator's current observation. This can formally be written as:

$$E_v = \left\{\left\{v_i, v_j\right\} | v_i \in V_v, v_j \in V_v, v_i \neq v_j \text{ and } l(v_i, v_j) \cap \Theta = \emptyset\right\} \qquad (3)$$

where $l(v_i, v_j)$ determines the vertices that fall on the line between $v_i$ and $v_j$, and $\Theta$ refers to the set of occlusions specific to the environment.

Each vertex $v_i$ has a degree $<v_i>$ that specifies the number of graph edges that touch $v_i$. With such a network transformation, an environment with $H(X) = 0$ is a complete graph with vertex degree $<v_i> = N - 1$, $\forall v_i \in V_v$. On the other hand, an environment that is only clutter is a disconnected graph with vertex degree $<v_i> = 0$, $\forall v_i \in V_v$. Therefore, the complexity of a graph passes through a maximum and goes down to zero for complete and disconnected graphs (Fig. 3c). An argument in support of this complexity definition arises from Shannon's information theory applied to random graphs[29]. Mathematically, the complexity of a network is defined as:

$$\alpha = \{<v_i> | v_i \in V_v\}$$
$$H(\alpha) = -\sum_{i=1}^{|\alpha|}\frac{\alpha_i}{|\alpha|}\log\left(\frac{\alpha_i}{|\alpha|}\right) \qquad (4)$$

**Eigenvector centrality of environments.** Environment quantization into a grid structure lends itself to a network representation based on how the system is connected together internally. If we again assume each cell is a vertex, in order to represent the environment dynamics we defined the edges in terms of actions. In such a network $G_w = (V_w, E_w)$, an edge $e_{i,j}$ between two vertices $v_i$, and $v_j$ exists if there is an action connecting the two vertices. Similar to before we can formally write this as:

$$E_w = \left\{\left\{v_i, v_j\right\} | v_i \in V_w, v_j \in V_w, v_i \neq v_j \text{ and } p(v_i) \xrightarrow{a} p(v_j) \forall a \in \mathcal{A}\right\} \qquad (5)$$

where $p(v)$ returns the cell for vertex $v$.

Eigenvector centrality (eigencentrality) depends both on the vertex degree $<v_i>$ and neighboring vertex centralities[32]. The centrality score $x$ of a vertex $v_i$ is defined as[32]:

$$x_{v_i} = \frac{1}{\lambda}\sum_{j=1}^{n}A_{v_i, v_j}x_{v_j} \qquad (6)$$

where $\lambda$ is the largest eigenvalue of the adjacency matrix $A_{v_i, v_j}$.

**Transitioning between habit and planning.** We combined habit-based action selection and planning to model the consequences on survival rate for prey that switched between habit- and plan-based action selection based on the eigencentrality of the environment (Fig. 4d). We grouped environments based on their spatial autocorrelation of eigencentrality (SAE) (Fig. 4e), in which environments with SAE below the 25th percentile were labeled as low, and environments with SAE above the 75th percentile were labeled as high. We then performed a habit/planning switching protocol within these environments.

The prey switched from habit- to plan-based action selection (with 5000 number of states forward simulated) when transitioning from a low eigencentrality region to a high eigencentrality region. During habit-based action selection, the prey used its knowledge about the next action to determine if the new location had a higher eigencentrality. Conversely, the prey switched from plan- to habit-based action selection when transitioning from a high eigencentrality region to a low eigencentrality region. During plan-based action selection, the prey compared the eigencentrality and gradient of eigencentrality at its current location to all other possible locations (details provided below). The transition regions were identified

based on the magnitude of the normalized eigencentrality ($X_E$) gradient:

$$|\nabla X_E| = \sqrt{\frac{\partial X_E}{\partial x}^2 + \frac{\partial X_E}{\partial y}^2} \qquad (7)$$

During habit-based action selection, given the prey's knowledge about the predator policy, the prey retained a belief state. The belief state was set to be the current state when the prey observed the predator. If the prey was not able to observe the predator, the belief state was randomly sampled and propagated based the prey's model of the predator. If and when the prey switched over to plan-based action selection at a given transition point, the prey's belief state generated during habit-based action selection was used to initialize the belief state used by the planner. During plan-based action selection, the belief state was set by the planner. If the prey switched to habit-based action selection, the belief state used by the planner was migrated over to the habit-based controller. This process was repeated in the case of multiple switches.

At the start of each episode the prey's first action was chosen by the habit-based controller. Given the nature of habit-based action selection, the prey's next position was set by the next action prescribed by the selected policy. During plan-based action selection, the prey's next position was a list, comprised of all allowable locations (e.g., not walls and obstacles). The prey's previous location ($x_{t-1}, y_{t-1}$), current location ($x_t, y_t$), and next location ($x_{t+1}, y_{t+1}$) were used to identify transition points. A transition was defined as:

$$|\nabla X_E(x_t, y_t)| > \max\{|\nabla X_E(x_{t-1}, y_{t-1})|, |\nabla X_E(x_{t+1}, y_{t+1})|\} \qquad (8)$$

The prey's action selection algorithm was switched from habit- to plan-based if the prey was at a transition point, and the current eigencentrality value was smaller than the next eigencentrality value (($X_E(x_{t+1}, y_{t+1}) - X_E(x_t, y_t)) > \epsilon$). Conversely, the prey's action selection algorithm was switched from plan- to habit-based if the prey was at a transition point, and the current eigencentrality value was greater than the maximum of the next possible eigencentrality values (($X_E(x_t, y_t) - \max_{a \in \mathcal{A}} X_E(x_{t+1}, y_{t+1})) > \epsilon$). We find that $\epsilon = 0.1$ works well.

**Environment lacunarity analysis.** Lacunarity (denoted by $\Lambda$) of the randomly generated environments were calculated by using the sliding box algorithm[36]. Simply, a box of size $r$ was moved over the entire image/environment, overlapping itself at each slide, thereby sampling each part of the image/environment multiple times.

An $r \times r$ box is placed at the starting corner of the environment, and the total number of occupied sites $S$ by the variable of interest is counted ($n(S, r)$; in our case occlusions). This value is often referred to as 'box mass'. The box is then moved $k$ columns/pixels to the right (notice that if $k = r$ we get fixed box counting), and the process of counting is repeated. The frequency distribution of box mass $S$ is converted to a probability distribution $Q(S, r) = \frac{n(S, r)}{N(r)}$, where $N(r)$ is the total number of sliding boxes of size $r$. For an environment of dimension $M_l \times M_w$ this value is then equal to: $N(r) = (M_l - (r + k - 1))(M_w - (r + k - 1))$.

Lacunarity $\Lambda(r)$ for a box size $r$ is defined as the ratio of the second moment of patches with mass $S$, $\mathbb{E}[S^2]$ to the first moment of the patches mass $S$, $\mathbb{E}[S]$ squared, i.e., $\Lambda(r) = \frac{\mathbb{E}[S^2]}{(\mathbb{E}[S])^2}$, where the mean number of occupied cells per box is $\mathbb{E}[S] = \sum SQ(S, r)$, and $\mathbb{E}[S^2] = \sum S^2 Q(S, r)$. More intuitively, since the variance of box masses is equal to

$$\begin{aligned} \mathrm{Var}(S) &= \mathbb{E}[(S - \mathbb{E}[S])^2] \\ &= \mathbb{E}[S^2] - 2\mathbb{E}[S]\mathbb{E}[S] + \mathbb{E}[S]^2 \qquad (9) \\ &= \mathbb{E}[S^2] - \mathbb{E}[S]^2 \end{aligned}$$

substituting $\mathrm{Var}(S) + \mathbb{E}[S]^2$ into the the lacunarity equation gives us: $\Lambda(r) = \frac{\mathrm{Var}(S)}{\mathbb{E}[S]^2(S)} + 1$, which implies that lacunarity correlates to the variance of box masses over the mean of box masses. Therefore, empty environments (entropy = 0.0) do not have a distribution of box masses ($\mathrm{Var}(S) = 0$, $\mathbb{E}[S] = 0$), which is therefore an asymptote. The reported values of $\Lambda(r)$ in Fig. 5a are the mean lacunarity values over the tested box sizes: $\Lambda_{\mathrm{avg}} = \frac{\sum_{g=1}^{N_r} \Lambda(r)}{N_r}$, where $N_r$ refers to the number of tested box sizes.

Each of the randomly generated environments were converted to black and white images of size $231 \times 231$. For these environments, the box sliding offset was set to $k = 1$ and box sizes were varied from $r = \{1, 24, 47, 70, 93, 116, 139, 185, 231\}$. These box sizes were chosen such that they would survey the space in increments of 10% (e.g., for box size $r = 24$ the total area being surveyed is ~10%, while for $r = 47$ the total area being surveyed is ~20% of the actual environment as the box is being glided across). Lacunarity analyses of the binary images were performed using the sliding box algorithm available in the FracLac plug-in[69] for ImageJ[70].

**Statistics and reproducibility.** A total of $n_{\mathrm{pred}} = 5$ random predator locations were used for Simulation 2 per random environment ($n_{\mathrm{env}} = 20$), per clutter level ($n_{\mathrm{entropy}} = 10$: 0.0–0.9 in steps of 0.1), and per number forward states the prey was allowed to plan over ($n_{\mathrm{sim}} = 5$:100, 1000, 5000). Survival rate was calculated over 50 episodes at a given predator spawn location, environment, clutter level, and number of states forward simulated. A total of 19 predator spawn locations were

removed from the 1,000 spawn locations present across all episodes due to the predator being stuck behind occlusions as a result of path planning using the Euclidean distance to the believed prey location. The consequent entrapment resulted in higher prey survival rates than would have occurred otherwise. In addition, the 1 and 10 number of states forward simulated were removed: With 1 state, the prey only evaluates one of the (at most) four possible cardinal directions, resulting in essentially random behavior; with 10 states, only two steps ahead are considered, making action choices largely random. This causes the prey to get stuck within relatively closed regions that require more than two steps ahead to obtain a path of egress. Therefore, the total number of trials for Simulation 2 was 147,150.

In high entropy environments (0.7–0.9) there are 18 predator spawn locations (out of the 150 total in these environments) that are on occlusions with an action that leads it out of the occlusion. There are also 12 predator spawn locations where a path from the predator spawn location to the prey does not exist. Counterbalancing this is a number of spawning locations in these packed environments which are adjacent to prey within relatively closed regions inevitably leading to death.

The simulations for Supplementary Fig. 6 were conducted in environments with success path occupancy (Fig. 3e, f) above the 75th percentile ($n = 30$). Similar to both Simulation 1 and Simulation 2, the policy of the predator was kept constant and the number of states forward simulated was (100, 1000, and 5000). Survival rate was calculated over 50 episodes for a given prey visual range, and number of states the prey could forward simulate (for a total of 22,500 trials).

In Fig. 3b the incremental benefit of planning is defined as the average difference in survival rate between the tested 100, 1000, and 5000 states forward simulated (e.g., the average of survival rate at 5000 minus survival rate at 1000) for a given environment. Due to a nonuniform increase from 1000 states to 5000 states (difference is not 1 when converted to log), a linear relationship was assumed, and the calculated difference was multiplied by 2.

In Fig. 4e, the spatial autocorrelation of the environment eigencentrality was calculated by using global Moran's I. Global Moran's I was calculated using the moransI function in R. Global Moran's I evaluates whether a set of given values and their locations are clustered, dispersed, or random. For this statistic, the null hypothesis is that the spatial distribution of feature values (eigencentrality score of a vertex) is random.

For habitat groupings in Fig. 5a–d, environments were grouped based on average lacunarities of typical habitats (Supplementary Table 1).

For all environment groupings, environments with entropies below the 25th percentile were categorized as low, and environments with entropies above the 75th percentile were categorized as high. Mid-level entropy was classified as entropies between low and high.

Data analysis was done using Python 3.7.4, R 3.6.1, and Mathematica 11.1. Network analysis for visually occlusive spatial complexity and eigencentrality were done in R and Mathematica with the 'igraph' package (R v1.2.4.2, Mathematica v0.3.116), and in Python with 'networkx' (v2.3). Statistical analysis was done using the 'numpy' (v1.17.2) and 'scipy' (v1.3.1) packages. Videos from raw episode files were created in Matlab (raw episode files and Matlab created videos available in Source Data folder). All significance indicators follow: n.s. is not significant $p \geq 0.05$; *$p < 0.05$; **$p < 0.01$; ***$p < 0.001$.

**Computing environment.** The computational resources for this work were provided by the Quest high performance computing facility at Northwestern University which is jointly supported by the Office of the Provost, the Office for Research, and Northwestern University Information Technology. The cluster is composed of 244 nodes of Intel Haswell E5-2680 processors with 128 GB memory/node, 184 nodes of Intel Xeon E5-2680 processors with 128 GB memory/node, 72 nodes of Intel Xeon Gold 6132 processors with 96 GB memory/node. Approximate runtimes: Simulation 1: 5,000 total compute hours (50 h on 100 Quest nodes); Simulation 2: 300,000 total compute hours (3000 h on 100 Quest nodes).

**Reporting summary.** Further information on research design is available in the Nature Research Reporting Summary linked to this article.

## Data availability
The source data underlying Figs. 2, 3, 4, and 5; Supplementary Figs. 3, 4, 5, 6, 7, 8, 9, 10, 11 and Supplementary Table 1 are provided as a Source Data file. Raw episode files used to generate the Supplementary Videos are provided in the Source Data folder. Source and binary satellite images from the Okavango Delta (Fig. 5a and Supplementary Fig. 11) are provided in the Source Data folder. Data to generate all the figures are available at https://github.com/MacIver-Lab/gridworld-decisionmaking/.

## Code availability
These results were generated using code written in Python, R, and Mathematica. Code is available at https://github.com/MacIver-Lab/gridworld-decisionmaking/. The accompanying browser-based game is hosted on https://maciverlab.github.io/plangame/, and the code is available at https://github.com/MacIver-Lab/plangame/.

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

## Acknowledgements

We thank C. Mobley (Sequoia Scientific, Inc.,) for sharing data and conducting Hydrolight simulations for the clear water condition of Supplementary Fig. 1. We thank D. Dombeck, J. Krakauer, M. Gallio, L. Schmitz, and H. Davoudi for comments on earlier drafts. We thank G. Espinosa and A. Lai for creating the accompanying browser-based game. This work was funded by NSF Brain Initiative ECCS-1835389.

## Author contributions

U.M. and M.A.M designed research; U.M. performed research and analyzed data; U.M. and M.A.M. wrote the paper.

## Competing Interests

The authors declare no competing interests.
