## [Peer Review File · Nature Communications]

Reviewers' Comments:

Reviewer #1:

Remarks to the Author:

The authors perform simulations to demonstrate that, under a certain set of assumptions, escape strategies that incorporate information regarding predator location during the escape behavior outperform feed-forward strategies in moderately complex environments. The authors argue that this result supports the hypothesis that access to more visual information (associated with the transition from seeing through water to seeing through air) favored an increase in cognitive ability in prey. The approach is novel and the hypothesis is interesting, but the predator and prey models appear to be informed by several distinct strategies, resulting in somewhat unrealistic simulations. It would be helpful to incorporate more specific biological data in the design of the predator-prey strategy models. Although the broader implications of the findings are interesting and impactful, it would be necessary to amend the modeling parameters to reflect our current understanding of predator-prey interactions for me to consider this paper as acceptable for publication in Nature Communications.

Major Comments:

Lines 53-54: The scenario of prey seeking a single refuge that is very distant, and having a predator placed between the prey and its refuge, is somewhat unrealistic. Prey that must seek the safety of a refuge rarely venture far from potential refuges, and generally tend to stay within the range of several refuges at a time (<https://doi.org/10.1007/BF00176714>, <https://doi.org/10.1007/BF00395696>). Previous research has shown that the flight initiation distance of these animals increases with distance to refuge, which demonstrates how important the distance to the nearest refuge is to these animals (<https://doi.org/10.1139/z89-033>). These prey are generally responding to sustained predation strategies, which explains the need to seek refuge rather than using other escape strategies. On the other hand, prey that venture far from potential refuges generally have high predator evasion ability and tend to be responding to predation strategies that are limited to a single strike in a specific location or a short duration of attack, so an evasive maneuver by the prey is effective at preventing capture (doi: 10.1007/s10164-014-0419-z, [https://doi.org/10.1016/0022-5193\(74\)90202-1](https://doi.org/10.1016/0022-5193(74)90202-1)). For example, in spiny mice the speed and flight initiation distance strongly predict the escape success, rather than direction of flight ([10.1007/s00265-007-0516-x](https://doi.org/10.1007/s00265-007-0516-x)). Thus, each possible location outside of the predator strike zone serves the equivalent purpose as a refuge. The diversity in escape strategies means that prey are rarely under selective pressure to simultaneously strategically evade sustained attacks while navigating a complex environment to reach a single distant refuge. The authors reference many different predation and escape tactics, but the constructed scenario does not account for the importance of matching a specific prey strategy to a specific predation strategy.

A stronger model to test this hypothesis would be based on matched data from real predators and the specific prey they hunt. Perhaps the authors can consider a pair of animals that interact under multiple conditions. For example, fox and rabbit locomotion has been well characterized, they inhabit ecosystems with varying complexity, and they are active both during the day and at night. The support for the overarching hypothesis of land animals being smarter relies on two observations: A) water environments have low network complexity and B) water environments have low visibility. If the authors consider a single land-based predator-prey system that can vary in both network complexity and visibility, then the same overarching hypothesis can still be tested. Designing the models to reflect pre-matched predator-prey strategies ensures that the simulations aren't testing the effectiveness of a prey strategy in response to a mismatched predation strategy. (Note, building models based on a semi-aquatic prey, such as frogs, would not be as beneficial because their escape strategies differ drastically in water and on land.)

Line 699: Does the predator know the location of safety and that the prey are trying to travel there? If so, why should the predator pursue the prey instead of hovering around the safety? In other words, why would the predator act as a reflex agent if given perfect knowledge of the environment?

Line 707: The speed capability of the model of predator behavior is based on pursuit predators, but an important strategy for evading pursuit predators is the turning gambit, which depends on the relative maneuverability of prey and predator ([https://doi.org/10.1016/0022-5193\(74\)90202-1](https://doi.org/10.1016/0022-5193(74)90202-1)). This is especially important in birds, but also in terrestrial predator-prey interactions with cheetah and impala (<https://www.nature.com/articles/nature25479>). Therefore, in simulations specific to pursuit-style predation, the relative maneuverability and accelerative capacity of the predator and prey are important to consider.

Modeling the habit-based strategy: Is it correct that the habit-based strategy is built by running 5000 simulations of the plan-based strategy? Then the "prey" has perfect knowledge of the result of each trajectory, which determines the weight with which each is randomly selected during the actual experiment? My concern is that this does not accurately reflect either how instinct evolves nor how learning happens in the lifetime of an individual.

Discussion:

There are several references to predator-prey interactions here, but most are quite different from the constructed scenario and therefore are limited in their benefit to the overall narrative of the paper. For example, felid use of occlusions during predation is likely the consequence of biomechanical constraint - unlike canids, felids are not capable of sustaining high-speed pursuits and are instead specialized for ambush predation, especially in trees. If the authors choose to base their simulations on a specific example of predator-prey interaction as suggested above, describing exactly how each would help streamline the discussion. This would be desirable as the manuscript is currently approximately 600 words over the limit by my counting.

The current simulations for informing habit-based models actually provide a mechanism for the emergence of thigmotaxis in low-visibility environments. This would be interesting to discuss further.

Figure 5: B) It would be helpful to label this panel "Environment network complexity." At what scale is network complexity calculated for the environments in figure 5? The most relevant measure of environmental complexity seems like it should be with respect to the prey animal's body size, rather than being measured overall at different scales. Does the fractal dimension metric have an effect on prey strategy simulations?

Minor comments:

Figure 4: Eigencentality and eigencentality gradient are interesting factors to consider in environmental complexity. The visual depiction of these concepts is helpful. The test for this shift in eigencentality triggering planning strategies is very interesting. C) The variety of colors in the choice point are not explained and not necessarily helpful.

Lines 8-10: Has this statement been tested? What about cephalopods? These are mentioned later in the discussion as aquatic species with high cognitive abilities. Team hunting has been observed in fish, and could be considered evidence of advanced cognitive abilities (<https://www.jstor.org/stable/1444679>, <https://doi.org/10.1098/rsbl.2014.0281>). Perhaps a less strong statement would be more appropriate here.

Lines 29-37: There are a few places throughout the paper in which references are made to neuroanatomy without sufficient context to add to the reader's understanding at that specific point in

the paper. The extra information in lines 66-70 regarding the roles of the hippocampus and prefrontal cortex would be useful here, and equivalent context regarding the lateral striatum would be beneficial.

Line 51: According to Jacobs (2012), "cognitive evo-devo" is "an enterprise to identify the primitives of cognition hand-in-hand with the primitives of the nervous system." This term should be explained when used, as it is not a commonly understood concept (only mentioned in 7 papers on google scholar). The use of this term may not be entirely appropriate in this manuscript, as these prey escape simulations are not providing additional insight into the function of the nervous system itself

Line 112: If it is correct that both reaching the refuge and continuing to evade the predator for the duration of the simulation are considered successful, then it would be more accurate to indicate "survival at the end of the simulation" as the termination state.

Line 116: It would be helpful to indicate here whether the simulations are turn-based or whether the agents can act simultaneously.

Line 155: "both the effective size... and the number of possible escape routes... decrease (no s)..."

Line 93/711: Should the visual conditions apply to both prey and predator?

Videos: For the videos showing representative death and survival trials, it would be helpful to indicate whether these are habit-based or planning-based strategies.

Reviewer #2:

Remarks to the Author:

The authors present a simulation study of a prey individual interacting with a predator over a range of environmental conditions that include structural complexity and differences in visual range. The stated goal is to relate the increase in visual range vertebrates experienced when moving from aquatic to terrestrial environments to the evolution of cognitive planning and evaluation of future actions and their rewards. The authors argue, using simulation results from simple and complex environments with short and long visual ranges, that planning is advantageous only when visual range is long and the environment is complex. They present this as evidence that the ability to plan in vertebrates may have evolved only after the evolution of terrestrial life histories due to the greater visual range possible on land. I am not an expert in brain evolution or higher-level cognitive processing. However, I can comment on the overall logic of the paper, the simulation results, and the inferences about the costs and benefits of motor planning.

Overall, I found the topic interesting and the analysis creative. However, the results felt over-sold in ways that make it difficult to see the real contribution of the simulations. My suggestion is that the authors significantly dial back the breadth of their hypotheses and the conclusions they draw from their analyses to focus more narrowly on the more proximate hypotheses they can explore with the simulations they have done. I believe there are many interesting ones, such as the advantage of mixed planning/habit based strategies, action selection at choice points and how it depends on environmental structure (e.g. Fig. 4 results), tradeoffs between computation time and performance in planning tasks, etc. In my opinion, the results would feel much stronger if these more specific hypotheses were given the space they deserve. Below I have described what I see as some of the main challenges with the paper as it is currently presented. I want to emphasize that the paper explores lots of interesting ideas and that the authors have made significant efforts to parameterize many aspects of the simulation environment, and to formulate a sensible model for planning versus

habit-based action selection. However, the hypotheses laid out in the abstract and introduction and the conclusions drawn in the discussion are just too grand and sweeping. I don't think the simulation results presented in the paper are enough to answer these ambitious questions.

- The overall premise of the paper is that a large visual range and a complex environment is required in order for motor planning to be beneficial, and that the evolution of the capability for planning may therefore have evolved only after vertebrates became terrestrial. This argument makes sense on the surface, but it takes some very thorny assumptions for granted. For example, is it really true that visual range is greater on land than in the water? On average, I'm sure it is, but there are so many exceptions that one immediately wonders whether this line of logic is too good to be true. Clear tropical waters, come to mind. Clear polar waters also come to mind. Low light terrestrial environments such as swamp and forest understory also seem to defy the general trend. The authors address this to some extent with their coral reef calculations, but there too, I am not sure the data are extensive enough to justify the conclusions (see my comment below). Another key element of the premise is that terrestrial environments are more complex than marine environments but again, this feels overly simplified. In addition to coral reefs, kelp forests, seagrass beds, salt marshes, rocky reefs, sea mounts, and lots of other marine habitats abound with structural complexity. Again, these may be exceptional locations but there seem to be so many of them that one cannot help but feel that sweeping statements about visual range and structural complexity on land versus in water are overly simplified. This might not be an issue if these general patterns were not so central to the authors' hypotheses about the evolution of planning — but they are.

- A second limitation that relates more directly to the simulations is the use of a discretized arena-like environment and the assumption that the animal has access to its absolute position in the environment via a cognitive map or the equivalent. Again, simplifications are obviously necessary to make the simulations tractable, but one wonders to what extent the setup of the arena and the cognitive map assumption drives results. An example is in figure 2c, where the authors point out that the action sequences of successful paths skirt the boundary of the simulated arena. This is presented as a strength of the simulations by referencing studies that have observed similar behaviors of animals placed in arenas (lines 136-138); but this seems to ignore the fact that the authors are trying to make statements about natural behavior in open environments — not the behavior of lab animals in arenas. A second inconsistency is the 2D treatment of three-dimensional environments like reefs. The fractal measurements reported by the authors for coral reefs are essentially measures of the structural complexity of the reef surface, but it is not clear how this relates to the way animals actually use the reef, which does not require them to travel the entire surface of a coral bommie, for example, to get from one point to another. Again, these seem like details but given that one of the main goals of the paper is to compare performance in aquatic and terrestrial environments, it seems essential that the salient details of these environments are captured in the simulations.

-The visual range argument — i.e. the idea that increased visual range is important for increasing the value of planning — is another key component to the overall premise of the paper. However, it is difficult to determine from the intro, results, and discussion why visual range and planning are related, particularly because the prey is assumed to have a full cognitive map of the environment. After digging through the methods (lines 723-726) and results more carefully, it seems that the advantage comes from having more time between detection of the predator and the capture of the prey by the predator. This makes sense, but it also makes me wonder how important a more explicit accounting of time would be (i.e. habit-based and plan-based actions may require very different amounts of time to implement). In any case, I think the overall argument for the importance of visual range would be clearer if the way visual range is being used in the simulations were made more explicit in the abstract, introduction, and/or discussion.

-The association between vision and planning also seems like a strong assumption. The authors address whether other sensory modalities could serve the purpose that vision serves in their simulations but I did not find this discussion fully convincing. The idea of Infotaxis (Vergassola et al. 2007 Nature and many follow-up papers) may be particularly relevant here. Infotaxis is a plan-based strategy that was originally formulated in the context of olfactory search behavior alongside a cognitive map. Later papers by Masson 2013 PNAS, Calhoun et al. 2014, eLife, Alvarez-Salvado et al. 2018 eLife focused on how the strategy could be implemented quickly and by a more biophysically constrained model of neural encoding/processing. I bring this up because this whole framework is very similar to the planning scheme the authors develop, but it does not require vision at all. I think it would be worth addressing this body of work in justifying the link between vision and the evolution of planning.

Minor comments

-The habitat complexity analysis in lines 276-279 is difficult to follow. Earlier in the paragraph, the authors present fractal dimension estimates and an entropy hierarchy that makes it appear that coral reef habitats are the most structurally complex, but then Mann-Whitney U test results and a summary stating that terrestrial environments are the most complex. I understand that different measures of complexity are being used here but it is challenging to see why one should be used rather than another. Could the justification of this be made clearer?

-The statement about fractal dimension and contrast attenuation on lines 263-261 is difficult to follow. The simulations assume the animal has a full cognitive map of the environment, regardless of its visual range. To me, this would imply that it should have the ability to use all of the structure in the environment, regardless of visual range. So why is the fractal dimension being related to the visual range? Reading through the Methods, I don't see a full explanation of this.

Reviewer #3:

Remarks to the Author:

This manuscript is a modelling and simulation study with the goal of understanding the circumstances that could benefit planning behaviour in animals. The key idea behind this manuscript is that there had to be a combination of factors occurring at the same time, to benefit planning over simpler habit based solutions. Firstly, animals had to evolve longer range vision, only possible in terrestrial environments. Secondly, there had to be environments that are complex enough to benefit planning, as too empty or too crowded environments only require simple behaviours (in the first you just go straight, in the second you just go the only available paths, i.e. don't have room to take alternative paths after detection of a predator). While making an evolutionary argument, all ideas would theoretically hold for predicting when a specific agent should engage in planning behaviours. In fact, some of the modelling addresses this question of when to plan directly, arguing for switches between habit and planning systems at points with high eigenvector centrality (essentially points where several specific paths are possible).

Overall, the paper is very interesting and has likely many implications for future research in the biology and evolution of behaviour as well as neuroscience. I had however some comments about how the habit model is implemented and some broader points about the interpretation of the results and future implications.

Major comments:

1) My biggest concerns were about the habit system compared to the planning system. Firstly, I was a bit confused by the fact that the habit system selects from the successful survival paths that were set

out by the planning system (specifically draws from a library of such paths). This simply sounds a bit implausible for a model trying to explain survival for simpler organisms that don't plan. The alternative, i.e. to learn from experience directly, is also strange because prey survival can't be used like reward in other paradigms because an animal cannot have any experiences of being eaten. How then can this system learn from negative feedback, i.e. by walking into a predator? I think this is a relatively fundamental problem because while they argue planning isn't adding any value in too simple environments, that is only true given that the habit system draws from a library generated by planning. It is not too surprising then that if there are only few paths available, planning and habit don't differ because they mostly draw from the same thing. The differences then only occur when different paths could be taken or paths need to be switched, which is where eigenvector centrality comes in. I suspect that it mostly acts as a surrogate measure for decision points or success path divergences. In other words, it allows switching of path based on planning and new information when there the possibility of actually divergent success paths. Otherwise the habit system anyway had a library of the success path from planning, which means habit and planning should be almost the same anyway. However, my intuition of the modelling might be wrong, so I am happy to be corrected. To summarize my two major concerns are: 1) is the way the habit system is created plausible. 2) doesn't the way the habit system is set up by definition make habit and planning very similar in many situations except at places with more than one possible success path or when dynamic updating is necessary? To be fair this is almost what the authors want to say, i.e. that planning gives selective advantages in only specific environments and then only at specific points except that the habit system's plausibility is crucial for this. If it draws from information it should not have, the argument isn't as persuasive. Additionally, if the only thing that the planning system allows is for decision points to be used that should be clearer. However, I am not sure whether this is true, so I would appreciate more clarity on what specifically drives the planning success at those specific points vs what the habit system is doing.

2) Neuroscientifically there is more work on planning than hippocampal prefrontal interactions. However, the authors are correct to point it out for particularly spatial planning abilities but should appreciate that there are many other prefrontal components likely crucial for high level planning. Overall, while the manuscript does not contain any neuroscientific results directly, it does, as the authors point out, have some implications for neuroscientists as it makes specific predictions about when an agent should plan. It gives, for example, an evolutionary argument for why there are sweeps during decision points in electrophysiological recordings in rats. However, this ends up simply restating what the original recording study already said, i.e. that when there are decisions to be made, there is planning and potentially simulation of future states. That those moments should happen when paths diverge is, I think, obvious to us and the rat. Maybe the authors know of any unique predictions their models make that go beyond "when there are potential decision points or diverging paths the agent engages in decision making and potentially planning", because any such predictions would be interesting and useful to experimentalists.

Potentially more exciting and worthy of more emphasis are the implications for comparative neuroscience (e.g. work by Mars et al). Equipped with a model it is potentially possible not just to give a generic argument for the existence of any planning but more specific predictions about the degrees of planning that should exist in different niches/environments within the same species and between species. In fact, it would be very useful if the authors could map this out a bit more. Is it possible to run simulations of different animal species within different environments (e.g. deserts might also be equally lacking complexity) and different movement speeds and visual ranges and get out specific predictions about their relatively level of spatial planning? If so, comparative neuroscience could use those values and predict differences in brain anatomy and functional circuits. Thinking about other predictions or implications of the model, would another prediction from the model be that there should be more planning in species with more decision diversity, i.e. larger repertoires or ways to flee? To put it another way, having more possible routes makes planning variable. Does this also mean

everything that allows for more diversity in potential responses should equally make planning more valuable? For their specific prey model this would boil down to if animals have different type of avoidance behaviours, rather than visual range combined with ideally cluttered environments creating multiple success paths?

3) The paper really only addresses spatial planning. However, there are many other aspects to planning as well as evolutionary pressures to develop it. For example, terrestrial environments allow for complex extended action sequences and tool building, which could also lead to different kind of planning.

Equally, planning can happen on very different timescales such as planning and food storage. I do completely accept that all those can't be implemented in the model, but they should be discussed when talking about planning, evolution and terrestrial life.

Minor Comments, questions and suggestions:

-Maybe it is worth mentioning whether there modelling has any implications for human planning and when planning should occur? E.g. when and whether people should have flexibly planning horizons and engage with planning only in specific environments?

-From figure 1 it is unclear whether trees can re-join i.e. if you are at same position at same step the tree branches should converge again.

-It is unclear from Figure 1 whether the cone is around the whole prey agent. If not, the state is not completely defined by position but also orientation, meaning direction of travel is an important consideration for strategy. However, the methods mentions sweeps and it wasn't completely clear to me whether those imply that the prey animals can sample all around themselves between moves?

-I was wondering whether the visual analysis was the right level of abstraction. It only deals with visual complexity, while it should be about spatial and navigational complexity on the level of the map the simulation uses. I think the conclusion is probably approximately correct, either way, but maybe the authors should acknowledge the problem of equating visual complexity with spatial complexity.

- I was also curious what the eigenvector centrality correlation with number of likely successpaths is for a given location?

There is a typo in the following:

"734 the prey observed the predator during this sweep than it knew the predator initial location, which"

Then not than and predators not predator

Reviewer #4:

Remarks to the Author:

The paper "The shift from life in water to life on land.." is an extremely exciting one, a major step forward in understanding brain-behavior evolution from an integrated computational and ecological perspective. It combines both empirical and modeling work in a novel way – while the main

information presented derives from modeling of predator-prey interactions in a variety of environments, an extraordinary amount of integration of basic empirical information, from a wide range of fields, is presented a prerequisite to this integration. These subjects presented include the comparative organization of map formation and habit formation in the brain, optical analysis of animal/environment systems, particularly resolution properties, fractal analysis of various environmental topographies, energetics of respiration and locomotion in land versus water, kinetics of animal pursuit and escape, and more.

This paper raises a great number of questions, the raising of which should not be taken as criticism, as it is a major strength of this paper that it opens new research fields. A good paper should not be penalized for doing so. I will list a number of them, assuming the authors can decide what kind of question each is: weaknesses that might be addressed, misunderstandings arising from problems in exposition, desirable questions that they intended to evoke; future studies, and so on.

There is one feature of exposition that I think might be improved – the author's narrative structure is story-like, raising the immediate information about brain structure and environment relevant to their modeling work, and proceeding through their modeling experiments in a logical sequence. As they proceed, however, the reader encounters one major evolutionary question after another with a proposed answer in part or whole, as well as applications of this work for interpreting laboratory experiments. (as an aside, the one large question they do mention in the abstract, the basis of the limited ability of animals to plan, is hardly addressed at all). I think some of this might be made explicit in the abstract or introduction or the significance of these experiments could be missed. For example, what is the likely reason for the increase in cognitive ability from water to land? How is homeothermy related? To what extent do sensory specializations themselves drive the computational structure of decision making? Is there a predator/prey arms race, and on what features? How does memory capacity inform decision-making? How might we understand common laboratory measures like open field "anxiety", freezing and so forth from this ecological context?

This reader would also have appreciated somewhat better delineation of when the authors are choosing a particular stance in research areas that are controversial, when they're simply citing established work (like the fractal geometry of the environment?) and when they're making their best guess in areas that haven't been much studied. Overall, a bit more meta-comments both at the intention and the data aspects of the project.

Specific questions:

32-35 Is a direct parallel between lateral striatum and hippocampus/ pfc in birds and mammals with the indirect and direct pathways in lamprey striatum intended? Possibly better to say that there is an overall homology of basic reinforcement of motor decisions. Might you say why the focus of the paper is delineating these two components of the forebrain, and not other brain divisions?

36 "nonlocal spatial representations" ? Maybe just say prediction or "imagination"?

43-44 It's my understanding that many researchers would locate a lot of the action-sorting in the basal ganglia proper, back to thalamus, to PFC/motor commands, contextualized by hippocampus. I think the interpretation offered in the paper is reasonable, but this might be a place where selection of an interpretation might be signaled.

66-68 – The distinction between "prey's location" in the egocentric (e.g. left 30 degrees) and not hippocampal, versus allocentric (e.g.the meadow I hunted yesterday) should be made somewhere.

74-80 Struggling a bit here to distinguish different classes of prediction. Overall, though, this is an

unusually clear exposition of a very confusing literature.

87-89 – in one of these animals at a time - I thought you meant just the prey for a while

99 – again, ego-allocentric. You'd been talking about the predator tracking the prey with uncertainty, and you didn't mark the change about which spatial context you were referring to.

Circa 112 I'm having some trouble visualizing how this scales up from a single, simple escape response to a sequence of actions, but the next section of the paper made the kinds of outlines much more clear. You might telegraph the reader to restrain their anxiety.

160 Satisfying analysis for clutter and behavioral variability

229 also very nice. I quite liked your supplementary Figure 3 for visualizing the strategies – are you out of allowed figures?

254 A lot more information about how you acquired the images for the fractal analysis would be nice, both here and in the methods. You don't even mention the terrestrial – perhaps it's because this is very well known?

311-340 Discussion of strategies. I'm wondering whether the assumption that the layout of the map is completely available to the animals makes a whole lot of sense, especially for occluding "clutter". Is that perhaps part of what pushes behavior out of the planning approach?

343-358 Really clear, interesting

359 – on mammals --To me, the elephant in the room here (obstructing the egress) is that most early mammals are nocturnal. You should really discuss this some.

370 LI-IS inverse correlation. There's no evidence that developmental pattern is a "constraint" – minimally, it's just the mechanism by which a selected pattern is put in place. Neural thigmotaxis – best overall strategy.

394 – Nice!

Methods: I am not able to evaluate the mathematical detail, but I have no reason to distrust it.

Reviewer -- Barbara Finlay, Cornell University

Response to Reviewer 1

Comment 1.1. The authors perform simulations to demonstrate that, under a certain set of assumptions, escape strategies that incorporate information regarding predator location during the escape behavior outperform feed-forward strategies in moderately complex environments. The authors argue that this result supports the hypothesis that access to more visual information (associated with the transition from seeing through water to seeing through air) favored an increase in cognitive ability in prey. The approach is novel and the hypothesis is interesting, but the predator and prey models appear to be informed by several distinct strategies, resulting in somewhat unrealistic simulations. It would be helpful to incorporate more specific biological data in the design of the predator-prey strategy models. Although the broader implications of the findings are interesting and impactful, it would be necessary to amend the modeling parameters to reflect our current understanding of predator-prey interactions for me to consider this paper as acceptable for publication in Nature Communications.

Response 1.1. As this is a summary of the gist of several specific points Reviewer 1 makes in the course of their review for better alignment between modeled and measured predator-prey ecology, we will address this point in our subsequent responses.

Comment 1.2 Line 53-54: The scenario of prey seeking a single refuge that is very distant, and having a predator placed between the prey and its refuge, is somewhat unrealistic. Prey that must seek the safety of a refuge rarely venture far from potential refuges, and generally tend to stay within the range of several refuges at a time (<https://doi.org/10.1007/BF00176714>, <https://doi.org/10.1007/BF00395696>). Previous research has shown that the flight initiation distance of these animals increases with distance to refuge, which demonstrates how important the distance to the nearest refuge is to these animals (<https://doi.org/10.1139/z89-033>). These prey are generally responding to sustained predation strategies, which explains the need to seek refuge rather than using other escape strategies. On the other hand, prey that venture far from potential refuges generally have high predator evasion ability and tend to be responding to predation strategies that are limited to a single strike in a specific location or a short duration of attack, so an evasive maneuver by the prey is effective at preventing capture (doi: 10.1007/s10164-014-0419-z, [https://doi.org/10.1016/0022-5193\(74\)90202-1](https://doi.org/10.1016/0022-5193(74)90202-1)). For example, in spiny mice the speed and flight initiation distance strongly predict the escape success, rather than direction of flight. Thus, each possible location outside of the predator strike zone serves the equivalent purpose as a refuge. The diversity in escape strategies means that prey are rarely under selective pressure to simultaneously strategically evade

sustained attacks while navigating a complex environment to reach a single distant refuge. The authors reference many different predation and escape tactics, but the constructed scenario does not account for the importance of matching a specific prey strategy to a specific predation strategy. A stronger model to test this hypothesis would be based on matched data from real predators and the specific prey they hunt. Perhaps the authors can consider a pair of animals that interact under multiple conditions. For example, fox and rabbit locomotion has been well characterized, they inhabit ecosystems with varying complexity, and they are active both during the day and at night. The support for the overarching hypothesis of land animals being smarter relies on two observations: A) water environments have low network complexity and B) water environments have low visibility. If the authors consider a single land-based predator-prey system that can vary in both network complexity and visibility, then the same overarching hypothesis can still be tested. Designing the models to reflect pre-matched predator-prey strategies ensures that the simulations aren't testing the effectiveness of a prey strategy in response to a mismatched predation strategy.

Response 1.2. A minor clarification: Reviewer 1 states above that “having a predator placed between the prey and its refuge, is somewhat unrealistic.” The predator’s starting location is random (exclusive of cells occupied by obstacles, the prey, or the refuge) as noted at page 5, line 116. Given the start location of the prey is at the bottom of the gridworld, it is, however, not uncommon for the predator to be between prey and the refuge, although depending on the clutter level it is often the case that the predator and prey do not initially see one another.

To the larger points that Reviewer 1 is making: There is an extraordinary amount of idealization occurring in our model. No animal works in a 2D world, on a checkerboard environment, with square occlusions, making discrete steps with turns restricted to multiples of ninety degrees, and given (in the case of the very time consuming process of planning) as much time as needed to select a plan before acting, with the predator held in suspended animation while that is done. Such abstractions are, of course, an essential aspect of any study with a theoretical or computational emphasis. For example, in the Howland paper from 1974 cited by Reviewer 1, there are also many idealizations, such as using a radius of curvature—not applicable to any real animal trajectory as their motion never conforms exactly to an arc of a circle—and assumptions of perfect friction, among many others. Our use of the word “refuge” carried many implications for Reviewer 1, some of which we had not intended. For the purpose of our study, the refuge was simply a point the prey had to arrive at without death. Thus, this location is better described as a “goal point”—it could be an intermediate point along a path away from the predator; it could be a refuge such as a burrow; it could be an exit from a region of space that cannot

be taken by the predator (for example, a narrow crevice that only the prey can get through to another open area). It is not obvious whether all these meanings are included in what ecologists mean by “refuge”—in case not, to avoid confusion between our use of the word and how ethologists think of the concept, we have modified the language of the study to only use “goal” rather than refuge.

Reviewer 1 asserts that “prey are rarely under selective pressure to simultaneously strategically evade sustained attacks while navigating a complex environment to reach a single distant refuge.” If, as a category, animals do not strategize while under predatory stress, then that does raise a problem for our study. However, the status of this claim is unclear. Certainly there is a significant amount of animal behavior in the context of predatory stress that has been interpreted as strategic, such as the broken wing display of certain ground nesting birds. If R1 could supply some references on this assertion we would be grateful.

Relabeling the refuge as a goal within our study clearly does not address the thrust of Reviewer 1’s concerns, which is that there is considerable empirical evidence regarding such goal pursuit in the context of predation across different species, and the dynamics of these movements vary greatly according to things like distance from the goal, predator evasion ability, whether predator strikes are single attacks or sustained pursuit, among other factors Reviewer 1 describes. The solution that Reviewer 1 favors is to reformulate major portions of the study such that these details are incorporated into the model. Since it would not be feasible to do this for all well documented predator-prey dynamics, the recommendation is to take one well characterized case and model this in detail. However, it is not apparent to us that any predator-prey dynamic is sufficiently well studied to completely and accurately codify into a model for use here, including rabbit-fox. Kinematic data is not sufficient—a control strategy (how does the predator move given any move of the prey; how does the prey move) needs to be devised/fitted to data. This is a major enterprise that would make the paper considerably more complex, and raise a large number of questions about the sensitivity of any of our conclusions to variations in what is likely to be dozens of model parameters—some of which are likely to be poorly supported by measurements. Here, we make the minimal assumption that the predator chases, and the prey either uses habit (essentially they do what has worked “in the past”) or plans (sample a significant fraction of all possible futures and pick the best). Finally, as also noticed by Reviewer 1, using such a pair of terrestrial animals would strain or break the connections to the aquatic condition which are really crucial to the whole thrust of the study.

As Reviewer 1 will appreciate, the question for any theoretical or modeling study is whether the theory or model provides insight into the target process. Measures of

that utility include predictions and examples of confirmation from the literature. We believe that the study provides a sufficient number of such instances. For example, Reviewer 1 states “The authors reference many different predation and escape tactics, but the constructed scenario does not account for the importance of matching a specific prey strategy to a specific predation strategy.” Reviewer 1 is correct that the constructed scenario does not prescribe such matching: but it is a testament to the strength of the modeling work, and its relevance to ethology, that strategy complementarity nonetheless emerges organically in the results. This is shown by the complementarity between thigmotaxis in the prey in low clutter spaces (Fig. 2c, Fig. 3d1), corresponding to a positive taxis toward areas of low eigencentality (Fig. 4a), and the predator’s strategy of moving into open areas, corresponding to a positive taxis toward areas of high eigencentality (Fig. 4b).

To help address Reviewer 1’s core concern that our idealizations may vitiate any relevance of our results to real instances of visually-guided behavior, such as visually-guided predator-prey interactions, we have added qualifiers in multiple locations (e.g., page 3, line 98, page 14, line 312).

Comment 1.3 Lines 699: Does the predator know the location of safety and that the prey are trying to travel there? If so, why should the predator pursue the prey instead of hovering around the safety? In other words, why would the predator act as a reflex agent if given perfect knowledge of the environment?

Response 1.3. The predator knows the location of all the obstacles (otherwise it would run into them) and—so long as the prey is in an unoccluded position—the prey. However, it does not know the location of the goal point, as shown at page 28, line 708, for precisely the reason the Reviewer suggests. The reason the predator acts as a reflex agent despite all this knowledge (other than goal location) comes down to computational limitations: We cannot afford the compute time needed to equip the predator with planning ability. Technically, as mentioned in our study, it is understood that running multiple partially-observable Markov decision processes (what the prey uses to plan) is typically not feasible [1]. We consequently equip the predator with the simple pursuit strategy described in the paper. There is no inconsistency in the predator having perfect knowledge of the location of all obstacles and boundaries yet still being a reflex agent.

Comment 1.4. Lines 707: The speed capability of the model of predator behavior is based on pursuit predators, but an important strategy for evading pursuit predators is the turning gambit, which depends on the relative maneuverability of prey and predator ([https://doi.org/10.1016/0022-5193\(74\)90202-1](https://doi.org/10.1016/0022-5193(74)90202-1)). This is especially important in birds, but also in terrestrial predator-prey interactions with cheetah and impala (<https://www.nature.com/articles/nature25479>). Therefore, in simulations specific to

pursuit-style predation, the relative maneuverability and accelerative capacity of the predator and prey are important to consider.

Response 1.4. Similar to Comment 1.1, Reviewer 1 is requesting us to specify a particular type of predator and prey with measure kinematic parameters, such as turning capability or acceleration, as these parameters are important in any given interaction. We have been deliberately conservative in our investing the predator and prey of our study with kinematic parameters: essentially, there is speed of the predator, and speed of the prey. They have to move one square at a time (the predator either moves one square or two squares to maintain a higher speed on average) but this can be in any direction. As predators are almost always larger than prey, and locomotory speed is proportional to body length, having the predator move faster than the prey seems a reasonable and minimal requirement. While it would be interesting to move beyond that minimalistic specification, as stated above, we are unsure how we could do so unless we pick a well characterized dyad, such as rabbit-fox or impala-cheetah, raising the problems discussed above.

Comment 1.5. Modeling the habit-based strategy: Is it correct that the habit-based strategy is built by running 5000 simulations of the plan-based strategy? Then the “prey” has perfect knowledge of the result of each trajectory, which determines the weight with which each is randomly selected during the actual experiment? My concern is that this does not accurately reflect either how instinct evolves nor how learning happens in the lifetime of an individual.

Response 1.5. Regarding the turning gambit, please see our response above. Regarding the habit strategy: the method we use to generate habits is the planning algorithm POMCP [2], allowing examination of 5000 states ahead of the current time. This algorithm has been formally shown, given sufficient samples, to converge to the optimal habit (See Theorem 1 of [2]). Further, as we see that the survival rate improvement is slowing as we plan using between 1 and 5000 states ahead, we have some evidence that convergence to the optimal policy is occurring. Using success paths generated by the planning algorithm as our policy set for habit is likely far better than what can possibly occur with selection of habits over phylogenetic time or learning over ontogenetic time. However, it is advantageous to our claims to equip the prey with habit policies that are the best possible policies; if we use something more realistic, then we run into questions of whether we might be underestimating the learning rates of our model prey.

Once the habit set is initialized, the probability of any given policy being selected at first is “flat”—equal probability. During simulations, one “success trajectory” is picked from that set according to a probability whose value is determined by the mean reward attained by that trajectory in past use. Initially, since none of the trajectories

in the set have been used, the probability of selecting any one of them is, again, flat and equal to the reciprocal of the number of habits in the set. Over time, each habit has a mean total discounted reward value that progressively approaches its “true” value, and as the selection probability is proportional to its mean discounted reward, the chosen path more accurately represents the selection of the best habit. Further details are in the Methods section at page 31, line 810. We have highlighted and clarified the rationale for using the planning algorithm to initialize the habit set of the prey at page 5, line 123.

While these details may be slightly helpful to Reviewer 1, the core of their concern is whether our model of habit approximates habits as they are learned in actual animals. As indicated at page 2, line 35 and elsewhere in our study, critical to this work is building upon a pre-existing literature on modeling habit and planning in animals, as well as a literature on reinforcement learning. Nonetheless, there are many reasons to believe that the current “dual system” view (either use precomputed action sequences, e.g. habit, or select from forward simulation of several candidates, e.g. planning) is overly simplistic and this binary will be refined into a continuous spectrum where habit and planning is blended together in some as yet not understood way (see [3]—where this transcendence is proposed by Daw, the computational neuroscientist most responsible for the prevalence of the dual system approach in more theoretical/computational accounts). Our current expectation is that a more refined future understanding will involve continuous and fluid arbitration of decision making mode along this spectrum, using something like the gradient of eigencentality arbitration mechanism, but with many more inputs than what we have modeled. For example, it may be the conjunction of our current scheme plus the sensed presence of a potential reward or threat above a certain magnitude. Ultimately, refinement of the dual system binary to something more continuous would increase the efficiency of planning, a metabolically costly operation, but is not likely to substantively change our key claims.

Comment 1.6. Discussion: There are several references to predator-prey interactions here, but most are quite different from the constructed scenario and therefore are limited in their benefit to the overall narrative of the paper. For example, felid use of occlusions during predation is likely the consequence of biomechanical constraint - unlike canids, felids are not capable of sustaining high-speed pursuits and are instead specialized for ambush predation, especially in trees. If the authors choose to base their simulations on a specific example of predator-prey interaction as suggested above, describing exactly how each would help streamline the discussion. This would be desirable as the manuscript is currently approximately 600 words over the limit by my counting.

Response 1.6. We have evaluated and further qualified our use of predator-prey interactions from the literature in light of these remarks by Reviewer 1. Our central aim was to discuss the role of habitat complexity in such interactions, recognizing of course that the interactions in real life are far richer with many more factors to consider than is the case in our highly idealized set-up.

Comment 1.7. The current simulations for informing habit-based models actually provide a mechanism for the emergence of thigmotaxis in low-visibility environments. This would be interesting to discuss further.

Response 1.7. We have elaborated on this finding further at page 6, line 150 and page 9, line 200. Our original work showed that thigmotaxis arises in low entropy environments, in both habit- and plan-based action control, in both low and high visibility conditions. In order to address another reviewer's concern, for the revision we also investigated what happens with low visibility in medium entropy (complex habitat) conditions with plan-based control, and found that thigmotaxis also emerges in this context (Supplementary Fig. 6).

Comment 1.8. Figure 5: B) It would be helpful to label this panel "Environment network complexity." At what scale is network complexity calculated for the environments in figure 5? The most relevant measure of environmental complexity seems like it should be with respect to the prey animal's body size, rather than being measured overall at different scales. Does the fractal dimension metric have an effect on prey strategy simulations?

Response 1.8. This panel has been relabeled accordingly. Regarding the scale for network complexity calculation: before we address this, we note that to respond to a concern of Reviewer 2, we have shifted to a new habitat measure to translate between our entropy-based results and results found in the ecology literature. This measure is called "lacunarity" and is described at page 37, line 985 in the main, and at page 7, line 183 in the Supplement. With that in mind, the question is now whether the spatial scale for the lacunarity measure used ought to conform to the prey's size.

Environment network complexity, also called vision-based spatial complexity. The complexity of the environments are calculated with respect to the prey body size, since one cell is one prey body size. These networks are calculated by looking at the distribution of the number of other cells that are **visible** from a given cell. Therefore, this analysis indicates the spatial distribution of occlusions that are 1) with respect to the body size (e.g. an occlusion that spans two adjacent cells would be 2 prey body lengths); and 2) with respect to the assumed dominance of vision in the behavior. Consequently, a visibility network analysis, also referred to as vision-based spatial

complexity, better encapsulates aspects of (visually-guided) navigational complexity, since it is with respect to the prey's body size. We believe that this is the measure that more directly affects prey strategy. However, given that using the vision-based spatial complexity is, to the best of our knowledge, an original contribution of our work, we cannot compare the values for our generated environments to prior literature for real life habitats. Therefore, our aim in using fractal methods (previously fractal dimension, currently lacunarity) was to have a method to relate the geometry of our synthetic environments to real world environments. Once we categorize generated environments according to the three lacunarity ranges of real environments (coastal aquatic, terrestrial, and structured aquatic), we then infer the vision-based spatial complexity of real world environments based on the values for the lacunarity-matched generated environments grouped within the same category. This has been spelled out more clearly in the new text where lacunarity is introduced.

Environment lacunarity. Fractal analyses are one of the simplest methods to describe an object or environment across a range of scales. Therefore, our fractal dimension analysis, or the lacunarity analysis that has replaced it in our revised submission, were not conducted at a single scale but spanned multiple different scales that encompass body size. Intuitively, lacunarity (Λ) quantifies the pattern and spatial scaling of gap sizes in landscapes (in this case gaps that arise as a result of obstacle distribution in a 2D environment) at a particular scale [4, 5]. In our analysis we used the mean lacunarity across different spatial scales, which encompasses prey body size. Fractal dimension and lacunarity are tools to analyze pattern or process across a wide range of scales. Therefore, these analyses provide the best null model against which to judge the real behavior of multi-scale natural patterns. This is similar to spatial randomness being the null model against a type of spatial patterning at a single scale.

Patterns with high lacunarity are coarsely textured and have gaps that are dissimilar in both size and distribution. Despite the measure encompassing multiple different scales, the idea that at all tested scales the patterning is coarsely textured implies large gaps that possibly cause regions with high eigencentality that taper away in all directions (low entropy). On the other end of the spectrum, patterns with low lacunarity are finely textured with homogeneous gap sizes at similar intervals. Such environments would have only a very small region of high eigencentality, not unlike what we observe for high entropy cases. Lacunarities in between these two edge cases (largely mid-entropy environments) would feature environments that have adjacent regions of low and high eigencentality (although, not exclusively such environments). It seems unlikely that lacunarity plays a direct role in prey strategy given that it does not necessarily quantify navigational complexity due to what the Reviewer has rightly pointed to. We believe that the metric that influences prey strategy

and behavioral complexity is vision-based spatial complexity as mentioned above.

Even though lacunarity may not directly play a role in prey strategy we want to address the problem of what the boundaries of different environments based on lacunarity would be at more relevant scales. In order to address this problem we limited our maximum box size to 100 m. The papers that we have cited for lacunarity values of different environments had study areas that were smaller than 100 m × 100 m. However, in cases where the study areas were larger than that, our maximum box size was set to a percentage of the study area length that was approximately 100 m. When we averaged lacunarities over these new, limited set of box sizes, the boundaries for ‘coastal aquatic’, ‘land’, and ‘structured aquatic’ changed. The ‘coastal aquatic’ region changed from >1.13 to >1.16, while the ‘land’ region changed from lacunarities in the range 0.24–1.16 to 0.24–1.34, and the ‘structured aquatic’ region changed from <0.49 to <0.56. **Despite these shifts in the boundaries between different environments, our overall statistics regarding environment network complexity and the difference in performance between habit- and plan-based action selection did not change.** Environments within the ‘coastal aquatic’ and ‘structured aquatic’ are similar in vision-based spatial complexity (MWU with Bonferroni correction $P = 0.22$), and are significantly simpler than ‘land’ environments (MWU with Bonferroni correction Coastal-Land: $P < 10^{-8}$; Structured Aquatic-Land: $P < 10^{-8}$). Notably, the performance of habit- and plan-based action selection in aquatic environments (coastal and structured aquatic domains) is not statistically different (MWU Coastal: $P = 0.052$; Complex Aquatic: $P = 0.056$). Therefore, our results are largely robust to changes in scale.

Comment 1.9. Figure 4: Eigencentality and eigencentality gradient are interesting factors to consider in environmental complexity. The visual depiction of these concepts is helpful. The test for this shift in eigencentality triggering planning strategies is very interesting. C) The variety of colors in the choice point are not explained and not necessarily helpful.

Response 1.9. We are glad Reviewer 1 found this part of our analysis interesting. Regarding the choice point coloring, this concern also arose for Reviewer 4 and we have made modifications to this panel. We would appreciate feedback on whether this makes the coloring of the choice point more helpful to Review 1.

Comment 1.10. Lines 8-10: Has this statement been tested? What about cephalopods? These are mentioned later in the discussion as aquatic species with high cognitive abilities. Team hunting has been observed in fish, and could be considered evidence of advanced cognitive abilities (<https://www.jstor.org/stable/1444679>, <https://doi.org/10.1098/rsbl.2014.0281>). Perhaps a less strong statement would be more appropriate here.

Response 1.10. We stand by our assertion that it is uncontroversial to assert that land animals have more elaborated cognitive abilities than aquatic animals. Within key paragraphs of the Discussion, our study references multiple studies that survey the very limited evidence of cognition or planning in fish. For example, we cite [6], which states “studies on spatial cognition capabilities are very scarce or even completely lacking in some anamniote groups.” In our own laboratory’s experience within teleost behavioral research (20 years), and those of many others we have had informal discussions with, it is not for lack of trying. The lack of studies is a result of the absence of result when more advanced paradigms are attempted. We also reference the few examples Reviewer 1 mentions, such as group hunting in fish. Even though singular exceptions such as group hunting in moray eels and grouper and cephalopods exist, as a group it is clear that land animals, particularly the mammals and birds we focus on in the study, have more advanced abilities in this domain. We have added depth to our discussion of the possible basis of complex cognition in cephalopods, in the Supplementary Text.

Comment 1.11. Lines 29-37: There are a few places throughout the paper in which references are made to neuroanatomy without sufficient context to add to the reader’s understanding at that specific point in the paper. The extra information in lines 66-70 regarding the roles of the hippocampus and prefrontal cortex would be useful here, and equivalent context regarding the lateral striatum would be beneficial.

Response 1.11. We have provided more context to the neuroanatomy references around page 2, line 38.

Comment 1.12. Line 51: According to Jacobs (2012), “cognitive evo-devo” is “an enterprise to identify the primitives of cognition hand-in-hand with the primitives of the nervous system.” This term should be explained when used, as it is not a commonly understood concept (only mentioned in 7 papers on Google scholar). The use of this term may not be entirely appropriate in this manuscript, as these prey escape simulations are not providing additional insight into the function of the nervous system itself

Response 1.12. We have removed references to cognitive evo-devo, although we do believe its use was appropriate since the simulations provide insight into the function of the nervous system itself, its evolution, and cognition, as the term was meant to designate. Here is a subset of some of the relevant contact points:

1. The proposed mechanism for switching between habit-based decision making and plan-based decision making, with anxiety (with increased theta coherence) as a proxy for eigencentality, is a clear neuroscience-relevant prediction.
2. The lacunarity analysis points to a specific subregion of terrestrial habitats

where planning gets maximal advantage in spatial contexts. This forms a testable prediction for comparative neurobiologists that planning ability in dynamic contexts will be proportional to the degree that the ancestral environment of the given group or species inhabited landscapes within that lacunarity window. The neurobiological correlates include consistency with the anticorrelation between IS and LI fractions, with IS larger in the animals that engage in dynamic planning.

3. We make a prediction that non-habitizable planning, lacunarity, and the corresponding need for open areas to be visually transparent may make sense of the data regarding how early hominins differentiated from stem hominids by exploiting landscapes that had open areas interspersed with more dense areas [7]—and that bipedality might have been necessary to achieve the visual transparency over these open areas as needed for our results to hold. This has implications for human cognitive neuroscience and studies of early hominin evolution.
4. Non-habitizable planning is predicted to rely on recurrent loops between PFC and hippocampus for planning over the sensorium rather than cognitive map. Parts of this are already indicated in the literature, but using a dynamic (non-habitizable) visual task will much more clearly corroborate this idea, and may help direct experiments on how an ancestral aquatic map system was elaborated into the tetrapod system and finally into one that could do planning over the contents of sensory experience.

Comment 1.13. Line 112: If it is correct that both reaching the refuge and continuing to evade the predator for the duration of the simulation are considered successful, then it would be more accurate to indicate “survival at the end of the simulation” as the termination state.

Response 1.13. We have made this clarification.

Comment 1.14. Line 116: It would be helpful to indicate here whether the simulations are turn-based or whether the agents can act simultaneously.

Response 1.14. We have made this clarification at page 6, line 139. Additionally this information is explicitly outlined in the methods at page 30, line 769.

Comment 1.15. Line 155: “both the effective size... and the number of possible escape routes... decrease (no s)...”

Response 1.16. This has been corrected.

Comment 1.17. Line 93/711: Should the visual conditions apply to both prey and

predator?

Response 1.17. The simulations in which the prey and predator’s visual range differ are 1) for the empty world simulations (pseudo-aquatic), and 2) newly added simulations to address concerns of Reviewer 2, conducted in a set of mid-entropy environments (pseudo-terrestrial condition) where the prey’s visual range was explicitly controlled (1, 3, or 5 cells ahead) (Supplementary Fig. 6). In the new mid-entropy simulations, prey and predator could not observe one another if there was an occlusion on the line between them, as with all other pseudo-terrestrial simulations. Below we will explain why we have chosen to provide the prey with a different visual range from that of the predator in a subset of our simulations.

The predator policy in most time steps is aggressive pursuit towards the prey’s location. In our first version of pseudo-aquatic simulations (not shown in the paper), the predator had the same visual range and visual cone in the direction of travel as the prey. When the prey was not in view, the predator created a belief distribution of all the possible prey locations that were not inside its visual cone (as is done for the prey with respect to the predator). However, unlike the prey, which samples as many predator locations as the number of states forward simulated (1–5000) from its belief distribution of predator locations to reason about the value of taking an action, due to the predator being a reactive agent, the predator only sampled one prey location from its belief distribution, and chose an action to minimize its distance to that sampled prey location. Moreover, because the predator does not have access to a model for prey action selection (as the prey has for the predator) nor access to the goal location, the pruning and propagation of the predator’s belief about the prey location was based on the assumption that the prey acts randomly. Given that we do not provide the predator with the location of the goal, or the overall aim of the prey, we could not use a perhaps more natural prey model (e.g. the prey is choosing actions that minimizes its distance to the goal).

In these simulations, the way in which the predator chose actions and maintained its belief distribution caused most episodes to terminate without any observation (i.e. neither the prey nor the predator ever observed each other), since the predator’s actions were effectively random. Even if we had used a more reasonable prey action selection model for the propagation of the predator’s belief distribution (as outlined above), we expect that the predator and the prey would only observe each other near the end, when the prey was near to the goal position. Therefore, unless the predator was positioned near the prey at the start of the simulation, it would not have enough time steps (actions left) to reach the prey before the prey reached the goal position. The only way to enable the predator to make good decisions, as well as maintain a well supported belief distribution about the prey’s location, is to

simulate both the prey and the predator as planners. This would allow the predator to choose actions based on more than one sample of the prey location, and model the prey action selection as an independent planner. While this is clearly ideal, it is not computationally tractable given the way planning is implemented (as recognized in the literature [1]): Both the prey and the predator would have to plan an action at each imagined step for each other. But given the number of action choices, the state-space of the simulated environment, and how temporally extended the task is, such an implementation is not feasible (addressed above in Response 1.3 and at page 4, line 104).

In sum, we have made the predator's visual range larger than the prey's because 1) it is not feasible to simulate both agents as planners; and 2) in the absence of that framework, which allows for more effective decision making in the absence of visual information due to more intelligent use of the belief distribution of prey locations, most of the trials ended with no observation events between the predator and prey. Extending the predator's visual range avoids this problem; it provides some of the benefits of simulating the predator as a planner without the computational intractability of so doing. Finally, as we know that for a good number of predator-prey dyads, the predator is larger, that eye size generally scales with body size (data cited in our prior work [8]), and that visual range scales with eye size (also cited in prior study [8]), a predator-prey visual range differential also has some ecological justification.

How does the asymmetrical visual range assumption impact our results? First, note that the predator having larger visual range makes it **more challenging** for the prey to survive. As such, it is a harsher test of the efficacy of either habit- or plan-based action selection in the prey. Our expectation is that as we reduce the predator's visual range, survival rate would increase, but our key findings—a) thigmotaxis and b) there is no advantage of planning in low visibility open environments—would not be affected. We have added these points to the Supplementary Text under Model Limitations.

Comment 1.18. Videos: For the videos showing representative death and survival trials, it would be helpful to indicate whether these are habit-based or planning-based strategies.

Response 1.18. We have changed the captions for the movies as well as the title cards to indicating whether they are habit- or plan-based strategies. We have also added an additional Supplementary Movie 2 that shows example survival and death trials for both pseudo-aquatic and pseudo-terrestrial simulations.

Response to Reviewer 2

Comment 2.1 The authors present a simulation study of a prey individual interacting with a predator over a range of environmental conditions that include structural complexity and differences in visual range. The stated goal is to relate the increase in visual range vertebrates experienced when moving from aquatic to terrestrial environments to the evolution of cognitive planning and evaluation of future actions and their rewards. The authors argue, using simulation results from simple and complex environments with short and long visual ranges, that planning is advantageous only when visual range is long and the environment is complex. They present this as evidence that the ability to plan in vertebrates may have evolved only after the evolution of terrestrial life histories due to the greater visual range possible on land. I am not an expert in brain evolution or higher-level cognitive processing. However, I can comment on the overall logic of the paper, the simulation results, and the inferences about the costs and benefits of motor planning.

Overall, I found the topic interesting and the analysis creative. However, the results felt over-sold in ways that make it difficult to see the real contribution of the simulations. My suggestion is that the authors significantly dial back the breadth of their hypotheses and the conclusions they draw from their analyses to focus more narrowly on the more proximate hypotheses they can explore with the simulations they have done. I believe there are many interesting ones, such as the advantage of mixed planning/habit based strategies, action selection at choice points and how it depends on environmental structure (e.g. Fig. 4 results), tradeoffs between computation time and performance in planning tasks, etc. In my opinion, the results would feel much stronger if these more specific hypotheses were given the space they deserve. Below I have described what I see as some of the main challenges with the paper as it is currently presented. I want to emphasize that the paper explores lots of interesting ideas and that the authors have made significant efforts to parameterize many aspects of the simulation environment, and to formulate a sensible model for planning versus habit-based action selection. However, the hypotheses laid out in the abstract and introduction and the conclusions drawn in the discussion are just too grand and sweeping. I don't think the simulation results presented in the paper are enough to answer these ambitious questions.

Response 2.1. We would very much like to avoid the impression of the results being over-sold. We believe that Reviewer 2 is stating this because they took issue with several core points in our original submission, as we describe—and we hope, correct—in the revisions we detail below. We have also written a section on model limitations in the Supplementary Text. Finally, we have made it more clear through-

out the study that our contribution regards dynamic planning—continuously changing rewards—which we have termed 'non-habitizable' planning. As we state in the Discussion, it is likely that planning over static rewards, or 'habitizable' planning, evolved first. We would appreciate the reviewer's feedback as to whether these adjustments help to avoid this negative impression.

Comment 2.2. The overall premise of the paper is that a large visual range and a complex environment is required in order for motor planning to be beneficial, and that the evolution of the capability for planning may therefore have evolved only after vertebrates became terrestrial. This argument makes sense on the surface, but it takes some very thorny assumptions for granted. For example, is it really true that visual range is greater on land than in the water? On average, I'm sure it is, but there are so many exceptions that one immediately wonders whether this line of logic is too good to be true. Clear tropical waters, come to mind. Clear polar waters also come to mind. Low light terrestrial environments such as swamp and forest understory also seem to defy the general trend. The authors address this to some extent with their coral reef calculations, but there too, I am not sure the data are extensive enough to justify the conclusions (see my comment below). Another key element of the premise is that terrestrial environments are more complex than marine environments but again, this feels overly simplified. In addition to coral reefs, kelp forests, seagrass beds, salt marshes, rocky reefs, sea mounts, and lots of other marine habitats abound with structural complexity. Again, these may be exceptional locations but there seem to be so many of them that one cannot help but feel that sweeping statements about visual range and structural complexity on land versus in water are overly simplified. This might not be an issue if these general patterns were not so central to the authors' hypotheses about the evolution of planning — but they are.

Response 2.2. Reviewer 2 expresses doubt about the validity of two essential claims made in the work: first, based on our prior work, that visual range on land is substantially better than visual range in water, and second, based on previously published landscape ecology data presented in the prior submission, that certain terrestrial habitats are substantially more complex than aquatic habitats. Unless these points are more convincing to the Reviewer in both regards, we have clearly failed. Broadly speaking, we can divide each into: A) what is the evidence for the assertion; B) if not convincing, then the claims need to be removed or weakened; if convincing, is the evidence communicated well in the current submission, and if not, what are the corresponding revisions to do so?

Visual range. Regarding visual range in water, we have previously shown [8] that

Visual range data for a 30 cm black disk versus eye diameter. These data are computed using equations and models presented in our earlier study [8], through bright daylight (noon, no clouds), moonlight, starlight, clear water, and coastal water. For reference, typical human eye diameter is 24 mm. The water clarity used for the clear water range estimate (in full sun, at a depth of 10 m) is based on a very clear sample taken in the Bahamas, comparable to the clarity of swimming pool water. The water clarity used for the coastal water estimate (also full sun) is based on water typical of freshwater bodies and saltwater near coastlines (Baseline model in our prior work [8]). These values are an upper bound on possible visual range for a black disk (as detailed previously [8]). The fundamental reason for the increase in visual range in air is the level of light attenuation in water compared to air. Light, at its most penetrating (blueish) wavelengths, decreases to $1/e$ or 37% of its original value at distances of about 20 m (the attenuation length) in perfectly clear water—far clearer than swimming pool water and only approximated at the deepest reaches of oceans [8]. The attenuation length for typical inland waters (where terrestrial vertebrates emerged) or coastal zones is much smaller, resulting in the shorter ranges shown. In comparison, air has an attenuation length of 10,000–100,000 m [8]. For less than full sun, or for more naturalistic contrast ratios than a black disk provides, range decreases rapidly (Supplementary Fig. 7 of the earlier study [8]). This results in aquatic visual ranges on the order of one body length [8, 9] for coastal and freshwater biomes where much of aquatic life is concentrated. Given body sizes in the range of meters, and typical locomotion speeds on the order of a body length per second, these range estimates show that visually-guided decision making often has to occur on the order of a second underwater. Above water, the same eye (after corneal shape changes to correct for the different index of refraction) provides around 100 times more time for these decisions to be made.

aquatic visual ranges are greatly diminished over those through air. The core of the argument, and a visualization of the result, is summarized below and for the convenience of readers it is also summarized in Supplementary Fig. 1.

Given these data we hope Reviewer 2 will agree that the difference between vision on land and in water is quite large, and well-founded rather than assumed (sensitivity analyses can be found in the Supplement to [8], p.11). The data are so stark that—as we have added to the Discussion—nocturnal terrestrial vision outperforms diurnal aquatic vision in terms of range in most cases. Clearly, this critical point was not sufficiently detailed in the original submission. To rectify this, we now make some of these same points in the opening paragraph of the study, in addition to citing this new summary of prior work in the Supplement at page 6, line 151.

Habitat complexity. Regarding “Another key element of the premise is that terrestrial environments are more complex than marine environments”:

Reviewer 2 is concerned that this is too broad a generalization and that the data are too sparse to make such a significant assertion. One oversight of our original submission was to assert several times, including in the Abstract, that terrestrial habitats **as a whole** are more complex and advantage planning. The new lacunarity analysis described below allows us to be more precise about the specific sub-type of terrestrial habitats—roughly speaking, savanna-like—that advantage planning. We have been careful to specify this throughout the revision including in the revised Abstract.

This point notwithstanding, we hope that Reviewer 2 would agree that with respect to the many pelagic fish in the ocean, living away from sources of structural complexity such as those they list, the complexity of the environment is low—essentially consisting of prey and predator that such fish happen upon, and conspecifics that they may be schooling with or attempting to mate. As noted in the literature, for these organisms, habitat is ephemeral.

Moving on to the case of demersal vertebrates—those living on or near the floor of the ocean or freshwater bodies: A major source of terrestrial complexity is plant life, which current evidence suggests was quite significant even by the time vertebrates invaded land [10]. Most major plant evolutionary innovations, such as vascular anatomy, occurred on land, and only more recently in aquatic plants (arising in land plants 345 million years before aquatic plants [11]). One reason is obvious, and redounds to the previously mentioned massive attenuation of light with distance. Because of this, there is far less light, and correspondingly less energy, for the generation of biogenic structure. While light at the surface of Earth is over 1,000 Watts per square meter in full sun, just 10 meters down it is 22% of this value.

After 200 m there is insufficient light to drive photosynthesis. This decrease in energy has obvious consequences for the generation of biogenic structure and thus for habitat complexity. A 2014 review of differences between seascape and landscape mosaic summarized that “In marine areas, relatively homogeneous environments are widespread, and inorganic substances cover the bottom in many cases” [12].

In addition to this light-deprivation-induced lack of biogenic structure, it should be noted that whatever structure is present at the substrate of aquatic systems, the coupling between this structure and the animals that live in water is optional given that aquatic animals are at near neutral buoyancy, while for terrestrial animals gravity makes the coupling to structure obligatory for all but flying birds.

These logical considerations aside—and in view of the fact that in shallow water there *is* sufficient energy for biogenic structure—Reviewer 2 raises the question of how strong our claim is given all the examples of aquatic structure they mention. As a brief background to our habitat analysis, given length limitations for the submission, we had not discussed the very real challenge of quantifying the complexity of natural environments. Thus, we used one measure—fractal dimension—without notice that it is not without its problems. Nonetheless, the literature suggests it is a solid choice [13, 14], despite weaknesses inherent in any single measure of what is undoubtedly a multidimensional phenomena.

However, given how sparse our data support for fractal dimension was, particularly in the aquatic context, we have decided to shift from fractal dimension to lacunarity. Lacunarity has been well-characterized for all the habitats of concern in our study, across multiple studies for each habitat type [15]—our new data table and figure incorporates measurements from 29 habitats, including seagrass beds, salt marshes, and other aquatic habitats that Reviewer 2 mentions. Lacunarity analysis is used to characterize the pattern and spatial scaling of gap sizes in landscapes [4, 5]. Intuitively, it is a measure of ‘gapiness’ or ‘hole-iness’ of geometric structure [16]. It therefore has some kinship to the eigenvector centrality measure used throughout our study. We have provided more description of lacunarity, including how it is computed, within the revised main text (page 11, line 260), as well as a section within the Supplementary Text (page 7, line 183).

Besides its far better data support, another reason we have decided to shift to using lacunarity is that it is a curve showing values across box (or voxel) sizes, rather than a scalar as in the case of fractal dimension. Therefore, we have moved away from a method that assumes a linear relationship between box count and box size. This means that it gives a multi-scale analysis of habitat geometry. As it turns out, in lacunarity plots, a fractal corresponds to straight line (showing the same level of self-similarity across scales) whereas most habitats, including our 2D gridworlds, are not

perfectly fractal [5]. Lacunarity analysis can be applied to data of any dimensionality. In the case of 2D images, it can be applied to both binary and grayscale image data (where grayscale values are interpreted to give an estimate of height of structures in plan-view images). It provides a compact representation of fractal, multifractal, and nonfractal patterns. While averaging across lacunarity across box sizes—as we have done presenting data in terms of mean lacunarity—flattens some of these details, it is still able to capture some important aspects of the curve.

With these points on lacunarity in mind, Reviewer 2's concern resolves to whether the aquatic habitats of demersal vertebrates are distinctly different from the terrestrial habitats of terrestrial vertebrates, in terms of their lacunarity. The literature confirms that this is the case, as we had previously found for the less powerful approach of using fractal dimension. **Of key importance, the region of mean lacunarity where we find planning attains advantage over habit (corresponding to entropy values of 0.4–0.6) is uniquely terrestrial—not overlapped by mean lacunarity values of aquatic systems.** Below we show a table of the values provided by the literature (included in the resubmission as new Supplementary Table 1):

Environment	Value ($\ln(\Lambda_{avg})$)	Reference
Coastal Aquatic Environment Lacunarity Values ($\ln(\Lambda_{avg})$)		
Fine and coarse sand flat seabed	asymptotic - unary image	Data from[17] Synthetic aperture sonar (SAS) image near La Spezia Italy, and SAS image Riga, Latvia
Seagrass with small patches covering <7% and ~13% of sampled area	1.16–1.26	Data from[18] Geographical information system (GIS) coverage samples from Owen Anchorage and Cockburn Sound region south of Fremantle, Western Australia. Additional reference[19]
Structured Aquatic Environment Lacunarity Values ($\ln(\Lambda_{avg})$)		
Seagrass with small patches covering ~37% of sampled area	0.18	
Seagrass with medium patches covering 16%–25% of sampled area	0.40–0.45	
Seagrass with medium patches covering 31%–37% of sampled area	0.25–0.27	Data from[18] Geographical information system (GIS) coverage samples from Owen Anchorage and Cockburn Sound region south of Fremantle, Western Australia. Additional reference[19]
Seagrass with large patches covering 17%–21% of sampled area	0.42–0.46	
Seagrass with large patches covering 32%–45% of sampled area	0.12–0.13	

Continued on next page

Table 1 – continued from previous page

Environment	Value ($\ln(\Lambda_{avg})$)	Reference
Salt marsh spatial distribution of plant species	0.03–0.49	Data from[20] Skallingen salt marsh in southwestern Jutland, Denmark
Coral reefs in the Upper Palaeozoic Era	0.41	Data from[21] Norwegian Barents Sea
Modern day coral reefs	0.23–0.41	Data from[21] Alacranes Reef and data from[22] Hawaii and the Carribean
Terrestrial Environment Lacunarity Values ($\ln(\Lambda_{avg})$)		
Herbaceous rangeland (ecosystems dominated by grasses and forbs[23])	1.34	
Shrub rangeland (ecosystems dominated by woody vegetation[23])	1.19	Data from[24] central coast region of California; land use land cover data developed by the US Geological Survey
Mixed rangeland (ecosystems where more than one-third of the land is a mixture of herbaceous and shrub or bush[23])	0.26	
Typical grazing land (shrub rangeland)	1.2	Data from[25] semi-arid Mediterranean region of Israel
Acre with groves	1.35	
Meadow	0.63	Data from[26] normalized digital surface model (NDSM) from German low-range mountain area in Baden-Wurttemberg
Meadow with fruit trees	0.56	
Orchard	0.47	
Terra firme forest	0.23	Data from[27] rainforest at Caxiuana, Para, Brazil. Additional references[28, 29]

These data are summarized in new Figure 5a, below, showing the lack of overlap of mean lacunarity between aquatic and terrestrial systems in the zone where planning is advantaged. Along the x-axis we have the entropy of the randomly generated environments we used in the study, and along the y-axis we have the mean lacunarity of natural habitats.

Lacunarity of coastal aquatic, terrestrial, and complex aquatic habitats. Distribution of lacunarity of generated environments. The line plot shows the mean natural log of average lacunarity (see Methods) and the interquartile range ($n = 20$ at each entropy level). Coastal: blue line ≥ 1.16 [17, 18, 19], $n_{\text{coastal}} = 110$; Land: green line 0.23—1.34 [26, 24, 25, 27], $n_{\text{land}} = 274$; Complex Aquatic: pink line 0.03—0.41 [21, 22, 18, 20], $n_{\text{complex aquatic}} = 88$.

In sum, in terms of mean lacunarity, demersal aquatic vertebrates inhabit a largely different habitat from terrestrial vertebrates, and it is only within a subregion of terrestrial habitats—not overlapped by aquatic habitats of demersal vertebrates—where planning is advantaged over habit.

To convey this new measure and in general strengthen the habitat complexity analysis we have made the following changes: Figure 5 and its caption; page 37, line 985; Supplementary Table 1; results section starting from page 11, line 260.

Comment 2.3. A second limitation that relates more directly to the simulations is the use of a discretized arena-like environment and the assumption that the animal has access to its absolute position in the environment via a cognitive map or the equivalent. Again, simplifications are obviously necessary to make the simulations

tractable, but one wonders to what extent the setup of the arena and the cognitive map assumption drives results. An example is in Figure 2c, where the authors point out that the action sequences of successful paths skirt the boundary of the simulated arena. This is presented as a strength of the simulations by referencing studies that have observed similar behaviors of animals placed in arenas (lines 136-138); but this seems to ignore the fact that the authors are trying to make statements about natural behavior in open environments—not the behavior of lab animals in arenas.

Response 2.3. As the reviewer points out, there is a high level of idealization throughout our submission. No animal lives in a 15 x 15 checkerboard 2D world, taking discrete steps, and—in the prey’s case—being given essentially as much time as it needs to plan regardless of the planning depth while the rest of the world is held in suspension. As with all modeling/theoretical work, we are making a large number of such assumptions. The concern of the reviewer is that some of these assumptions are tantamount to circular reasoning, in that we are choosing a setup that necessarily leads to our results. Of course, this being a model, the results of the simulation have to be as present in the setup as the proof of any theorem in formal math has to be present in the axioms. Thus we cannot easily defend our case, other than to argue why choices within our setup are good choices to make and do not seem to obviously correspond to our results, or to argue that varying the assumption does not cause a change in our results. We will do this first for the cognitive map concern, and second for the concerns about the arena.

Perfect map assumption. In the case of our assumption that the predator and prey have a perfect cognitive map (in the predator’s case, excluding the location of the safety point that the prey is trying to reach), our modeling choice is driven by a desire to surgically isolate the relative properties of planning versus habit. First, as the assumption that the predator has a perfect cognitive map (minus the prey’s goal) is the most detrimental to the prey’s survival, we think this is a good assumption to keep, particularly as we are not able to simultaneously implement planning in the predator due to computational limitations. In the context of the current body of literature on model-based decision making, let’s consider the following possibilities for the prey: 1) no cognitive model (no cognitive map), but the prey learns the model; 2) inaccurate cognitive model (map), updated with learning; 3) inaccurate cognitive model (map), not updated with learning; 4) perfect cognitive model (map), not updated with learning.

For 1) our study becomes much more difficult (in fact, we are not sure how we would proceed), since a large number of empirically poorly-justified modeling choices on learning rates and number of exposures to a given random environment would need

to be made so that the planning algorithm has a model over which to plan. For 2) largely the same considerations apply, but now the planning algorithm can generate action decisions; however, those action decisions will be based on a value function that corresponds to this inaccurate cognitive model. The decisions may not be optimal. It is unclear, in this case, in what way to add noise or error to the cognitive model. Putting that problem aside, if the decisions fail to be optimal, in the context of this scenario (irreversibility since death occurs upon capture), there may not be much to learn; but inasmuch as learning can occur, the issues raised about the complexity of our study and poorly justified parameters enter in. For 3) decisions may again not be optimal; but now the failures of the cognitive model—again, the parameterization of which is uncertain—are not corrected over time. For 4) the attraction of this choice for our study is avoidance of uncertain parameterization of learning the model, and avoidance of uncertain parameterization of having an inaccurate model. We believe this enables us to surgically isolate the impact of planning in the context of a generous assumption of learning having previously been completed, and completed with perfect fidelity to the underlying structure (not counting the predator), where that structure is necessarily static (since learning cannot occur to accommodate a dynamic environment, and a dynamic environment without learning would quickly lead to an inaccurate model—case 3).

Arena. The study required a host of modeling choices, such as the 15 x 15 domain size among others: we provided a basis for those choices in Table 1 of the Supplement of our original submission, and in a new section of Supplementary Text on Model Limitations. Here we focus on the Reviewer’s concern about results accruing from arena setup.

In regard to thigmotaxis, while it is often referred to as “wall following” within the neuroscience literature (appropriate given their typical experimental apparatuses), the way ethologists define it is as a positive taxis toward solid objects or confined spaces. Thus, if a rodent was dropped in an open outdoor space near a boulder, running for the boulder would be thigmotaxis. This is therefore a natural behavior, one originally described in naturalistic contexts [30]. We have made this clearer in the text at page 7, line 164. In our study, we decided to use the eigencentrality metric specifically to abstract away from the particularities of our arena. An example of this is provided by what—as Reviewer 2 noted—we think is a strength of the work, which is that thigmotaxis emerges as a behavior. In the context of the eigencentrality analysis, thigmotaxis is described as negative eigencentrality-taxis.

Reviewer 1 raised a different concern that is relevant here, and that is the position of the safety point being on the opposite side of our environment. The start location of the prey is always (7, 0), while the safety is at (7, 14). If the safety were located

at (7, 5), for example, it is unlikely we would observe thigmotaxis. However, the reason we are using 15 x 15 environments is because larger environments are not computationally feasible given our resources, and we wanted to use as large an environment as possible so that the space being planned over during plan-based action control is as large as feasible; with the goal being closer, the environment is effectively reduced in size. (The infeasibility is due to the number of states that need to be traversed. By going from 15 x 15 to 20 x 20, the number of states increases from approximately 10^{36} to 10^{52} . With planning 5,000 states ahead in a 15 x 15 arena, a single trial in one environment requires roughly 2 hours of compute time, and the pseudo-terrestrial experiment alone consists of 250,000 trials.)

A further point to note is that prior to our simulations and eigencentality analysis, it was not apparent to us that thigmotaxis would emerge—on the contrary, we were surprised to observe it. We don't see any obvious linkage between our arena setup and this result.

Comment 2.4. A second inconsistency is the 2D treatment of three-dimensional environments like reefs. The fractal measurements reported by the authors for coral reefs are essentially measures of the structural complexity of the reef surface, but it is not clear how this relates to the way animals actually use the reef, which does not require them to travel the entire surface of a coral bommie, for example, to get from one point to another. Again, these seem like details but given that one of the main goals of the paper is to compare performance in aquatic and terrestrial environments, it seems essential that the salient details of these environments are captured in the simulations.

Response 2.4. To our knowledge there is no quantification of how structural complexity relates to behavioral complexity—clearly, the present work may help to address this gap at the very coarse level of granularity of aquatic versus terrestrial habitats. What ecologists have computed with these measures are factors such as biodiversity—number of species [13] and how the distribution of interstitial space might impact predator-prey relationships [31]. We hope that the additional efforts to strengthen the habitat complexity analysis described above will allay Reviewer 2's concerns in this area.

Comment 2.5. The visual range argument—i.e. the idea that increased visual range is important for increasing the value of planning—is another key component to the overall premise of the paper. However, it is difficult to determine from the intro, results, and discussion why visual range and planning are related, particularly because the prey is assumed to have a full cognitive map of the environment. After digging through the methods (lines 723-726) and results more carefully, it seems that the

advantage comes from having more time between detection of the predator and the capture of the prey by the predator. This makes sense, but it also makes me wonder how important a more explicit accounting of time would be (i.e. habit-based and plan-based actions may require very different amounts of time to implement). In any case, I think the overall argument for the importance of visual range would be clearer if the way visual range is being used in the simulations were made more explicit in the abstract, introduction, and/or discussion.

Response 2.5. We thank Reviewer 2 for these helpful suggestions. To address them, we have added a new Supplementary figure, included below for your and their convenience, that draws a closer connection between visual range and the utility of planning. This is for the pseudo-terrestrial condition, where we have selected the top 75th percentile environments in terms of success path diversity (a total of 30 environments). In those environments, we have re-run the simulations of planning, but with the visual range of the prey limited to 1, 3, or 5 cells ahead (in a cone, similar to Fig. 1a). This figure shows that when visual range is reduced, thigmotaxis again re-emerges, and survival rate decreases to a level similar to that obtained with habit-based action selection (two-tailed One-way ANOVA $P = 0.11$).

Survival rate and prey success paths with limited visual range. (a) Mean percent change in survival rate (computed as [survival rate with limited visual range minus survival rate with unlimited visual range] divided by [survival rate with unlimited visual range]) versus visual range in environments that had prey success path spreads above the 75th percentile for prey with ‘unlimited’ visual range ($n = 30$) (see Methods page 33, line 885). The fill indicates \pm s.e.m. of percent change in survival rate across selected environments. (b) For three different environments (rows), we illustrate how success path diversity varies with visual range (columns). Heatmaps are of all action sequences taken by the prey that resulted in prey survival at the maximum planning level (5000 states forward simulated), with color density proportional to frequency. Success paths for full visual range are given for the same predator start location. Color bar action frequencies range from 1–62, dependent on environment entropy and visual range. (c) Path spread is defined as the percent of cells occupied by action sequences that resulted in prey survival, at the maximum planning level (examples in b). High path spread indicates that the success paths are highly dissimilar. The horizontal line corresponds to the mean, the shaded region corresponds to the s.e.m., and the box corresponds to the 95% confidence interval of the mean. The lines extending from the boxes show the range of the data. (two-tailed MWU test with Bonferroni correction visual range 1-visual range 3: $P = 0.42$; two-tailed MWU test with Bonferroni correction visual range 3-visual range 5: $P = 0.006$; two-tailed MWU test with Bonferroni correction visual range 5-visual range max: $P = 0.016$; two-tailed One-way ANOVA across $P < 10^{-5}$)

We have amplified the relationship between visual range and planning in the following locations: We added a new methods section page 33, line 885 and modified the main text at locations page 6, line 150 and page 9, line 200.

However, even with this, note that we are allowing the prey to complete planning

at each move, regardless of how much (actual) time this takes. Clearly, planning is a time consuming process for brains and algorithms alike, due to the computation scaling exponentially with the number of steps ahead being planned for. In our highly abstracted framework, an increase in visual range results in increased time for the prey to escape behind a barrier or take another evasive move but is independent of the time needed for the calculations of planning. Other investigators have explicitly examined time [32], but this is currently outside the scope of our work.

Comment 2.6. The association between vision and planning also seems like a strong assumption. The authors address whether other sensory modalities could serve the purpose that vision serves in their simulations but I did not find this discussion fully convincing. The idea of Infotaxis may be particularly relevant here. Infotaxis is a plan-based strategy that was originally formulated in the context of olfactory search behavior alongside a cognitive map. Later papers by Masson 2013 PNAS, Calhoun et al. 2014, eLife, Alvarez-Salvado et al. 2018 eLife focused on how the strategy could be implemented quickly and by a more biophysically constrained model of neural encoding/processing. I bring this up because this whole framework is very similar to the planning scheme the authors develop, but it does not require vision at all. I think it would be worth addressing this body of work in justifying the link between vision and the evolution of planning.

Response 2.6. The new Supplementary figure for the last point above may help convince Reviewer 2 that the association we make between vision and planning is sound: Planning with reduced visual range collapses the advantage gained by planning to something similar to habit-based action selection. The basic logic of our argument is as follows. **First**, for short visual ranges, the prey's belief distribution for the predator's (unseen) location becomes very diffuse over time (Supplementary Fig. 4, and the new Supplementary Fig. 7). In the context of a diffuse belief, planning occurs with such a high level of uncertainty that any particular action selection is likely to be wrong given the actual location of the predator. **Second**, other sensory modalities, due to their lower spatial resolution, will typically result in a belief distribution more diffuse than the distribution resulting from a visual update—at least for targets at the distances (equivalently, sufficient times before interception) over which we expect planning to be possible. The time demands are related to the computational burden of planning as mentioned above (see also [32]). In practical terms, for spatial planning, one measure of this burden is the difference in speed between an animal through space and the speed of movement along an imagined trajectory. Across multiple studies in rodents, pre-play and the related phenomena of re-play occurs with a speed that is approximately 20 times that of real-time speed. Thus to imagine two trajectories, the animal needs a time equal to the sum of the length of

the two imagined paths, divided by the speed of the animal, divided by twenty.

Long range sensing with high acuity and rapid updates is nearly exclusively the province of visual systems (though acuity spans a wide range [33]). This is widely acknowledged in the literature [34]. In addition, while other sensory modalities require that the objects of perception also be generators of the energy of transduction, in the case of vision, objects of perception need only be reflectors of the energy of transduction [34, 35]. For example, objects sensed through olfaction need to emit odors (whether or not they are attached to their generator, such as an animal); objects sensed through audition need to emit sound (not including echolocation). Under daylight—and, as shown in our new figure above, even under nighttime conditions such as used by flying fruit bats with their rod-dominated retinas [36]—objects sensed through vision do not need to emit light, but only reflect it.

We do not claim that planning can only occur with vision (as we are clear at page 15, line 352); rather, the linkage between planning and vision is clearest in the case of “non-habitizable” planning. This is planning over a space which is so dynamic that it changes each planning cycle—typically this would be planning with respect to a moving predator or prey, but it could include dynamic conditions that do not arise from animal activity, such as crossing a river by way of stepping on floating objects moving along its surface, planning a safe path of egress from a rapidly moving forest or bush fire, or (as mentioned in the Supplement in the section on planning in invertebrates) certain kinds of sexual competitions. Olfactory cues can be an effective basis for what is initially plan-based action selection that then transitions to the habit system—habitizable planning. To help clarify the habitizable vs. non-habitizable distinction, we are updating our Supplementary Fig. 2 to show the habitizable scenario.

To ease Reviewer 2’s concerns, we have strengthened the discussion of the relationship between planning and vision at several places within the Discussion, such as at page 16, line 400, in addition to describing the implications of the new Supplementary Figure within the Results, and some additional material within the “Sensory ecology of planning” section of the Supplementary Text.

While these arguments make the connection between vision and planning—at least for spatial planning—more secure, the point regarding infotaxis is well taken. A way that infotaxis may be applied to this domain would be to perform entropy minimization over the likelihood of survival. It is possible that in environments where we do not see behavioral variability (low and high entropy environments), despite the task featuring a dynamic threat that would conventionally necessitate the relearning of the value function at each time step (essentially, to plan), there is in practice very little change in the value function. Therefore in some sense, while our task is dynamic it may not be ‘fully dynamic’ in all environments (which is further evidenced by the

success of blind habit in low and high entropy environments). Such an approach would possibly simplify planning in these environments.

While infotaxis chooses actions by minimizing the entropy of a belief distribution representing, say, target location, the planning algorithm presented here optimizes a policy using Bellman backups of a terminal reward. A theorem [2] shows that with sufficient samples, the method converges to the optimal policy. As a result, although these algorithms are distinct, their results could potentially agree if the chosen reward were entropy of a belief. To use infotaxis-like methods here, one would need to use entropy minimization on the belief distribution representing likelihood of survival. However, the complexity of infotaxis (the need to grid the space and compute entropy for all possible actions) makes it impractical to use directly on survival beliefs. Alternatively, bio-inspired algorithms, such as those referenced by Reviewer 2 [37, 38, 39] that approximate infotaxis in a physiologically plausible manner, might be sufficient to show that the rough framework provided by infotaxis can be used for the problems pursued here. More broadly, both infotaxis and the algorithms used here are methods of the more general problem of optimal planning under uncertainty, so one should expect that there will ultimately be many algorithms that lead to the same optimal or near-optimal solutions.

These methods can be used in low or high entropy environments where we expect a stable distribution of survival probabilities. However, in mid-entropy environments that feature adjacent clusters of high and low eigenvector centralities, a possible reason for why we see behavioral variability and the need to plan in regions of high eigenvector centrality is that there is a significant amount of change in the value function over an animal's likely sensorimotor delay time or planning cycle duration. This makes the task fully dynamic, and necessitates the re-learning of the value function via planning at each step. Unless short update horizons are used, and robustness to multi-peaked distributions is available (as in the approach we have been developing, ergodic information harvesting [40, 41]), infotaxis-like methods may become infeasible.

We have expanded on these points in the Supplementary Text starting at page 6, line 151, as we currently do not see a way to incorporate them in the main text given that we are already over our word limit.

Comment 2.7. The habitat complexity analysis in lines 276-279 is difficult to follow. Earlier in the paragraph, the authors present fractal dimension estimates and an entropy hierarchy that makes it appear that coral reef habitats are the most structurally complex, but then Mann-Whitney U test results and a summary stating that terrestrial environments are the most complex. I understand that different measures of complexity are being used here but it is challenging to see why one should be

used rather than another. Could the justification of this be made clearer?

Response 2.7. As detailed above, we have shifted from the fractal dimension analysis to using lacunarity. In addition we have clarified our use of what is initially labeled network complexity, and subsequently called vision-based spatial complexity, at these locations in the manuscript: page 7, line 174 and page 13, line 285.

Comment 2.8. The statement about fractal dimension and contrast attenuation on lines 263-261 is difficult to follow. The simulations assume the animal has a full cognitive map of the environment, regardless of its visual range. To me, this would imply that it should have the ability to use all of the structure in the environment, regardless of visual range. So why is the fractal dimension being related to the visual range? Reading through the Methods, I don't see a full explanation of this.

Response 2.8. The reviewer is correct that the existence of a perfect cognitive map implies that any structure can be used independent of visual range and perceived visual complexity. We have moved away from that argument with our lacunarity analysis. In calculating the lacunarity values of different types of environments found from the literature, and calculating the lacunarity of our generated environments, we use the entire environment from a perspective-independent top-down view. We have expanded on this in the Supplementary Text, and removed all mentions of perceived environmental complexity.

Response to Reviewer 3

Comment 3.1 This manuscript is a modelling and simulation study with the goal of understanding the circumstances that could benefit planning behaviour in animals. The key idea behind this manuscript is that there had to be a combination of factors occurring at the same time, to benefit planning over simpler habit based solutions. Firstly, animals had to evolve longer range vision, only possible in terrestrial environments. Secondly, there had to be environments that are complex enough to benefit planning, as too empty or too crowded environments only require simple behaviours (in the first you just go straight, in the second you just go the only available paths, i.e. don't have room to take alternative paths after detection of a predator). While making an evolutionary argument, all ideas would theoretically hold for predicting when a specific agent should engage in planning behaviours. In fact, some of the modelling addresses this question of when to plan directly, arguing for switches between habit and planning systems at points with high eigenvector centrality (essentially points where several specific paths are possible). Overall, the paper is very interesting and has likely many implications for future research in the biology and evolution of behaviour as well as neuroscience. I had however some comments about how the habit model is implemented and some broader points about the interpretation of the results and future implications.

My biggest concerns were about the habit system compared to the planning system. Firstly, I was a bit confused by the fact that the habit system selects from the successful survival paths that were set out by the planning system (specifically draws from a library of such paths). This simply sounds a bit implausible for a model trying to explain survival for simpler organisms that don't plan. The alternative, i.e. to learn from experience directly, is also strange because prey survival can't be used like reward in other paradigms because an animal cannot have any experiences of being eaten. How then can this system learn from negative feedback, i.e. by walking into a predator? I think this is a relatively fundamental problem because while they argue planning isn't adding any value in too simple environments, that is only true given that the habit system draws from a library generated by planning. It is not too surprising then that if there are only few paths available, planning and habit don't differ because they mostly draw from the same thing. The differences then only occur when different paths could be taken or paths need to be switched, which is where eigenvector centrality comes in. I suspect that it mostly acts as a surrogate measure for decision points or success path divergences. In other words, it allows switching of path based on planning and new information when there the possibility of actually divergent success paths. Otherwise the habit system anyway had a library of the success path from planning, which means habit and planning should be almost the same anyway. However, my intuition of the modeling might be wrong,

so I am happy to be corrected. To summarize my two major concerns are: 1) is the way the habit system is created plausible. 2) doesn't the way the habit system is set up by definition make habit and planning very similar in many situation except at places with more than one possible success path or when dynamic updating is necessary? To be fair this is almost what the authors want to say, i.e. that planning gives selective advantages in only specific environments and then only at specific points except that the habit system plausibility is crucial for this. If it draws from information it should not have, the argument isn't as persuasive. Additionally, if the only thing that the planning system allows is for decision points to be used that should be clearer. However, I am not sure whether this is true, so I would appreciate more clarity on what specifically drives the planning success at those specific points vs what the habit system is doing.

Response 3.1. We understand Reviewer 3's concern with using the planning algorithm to generate policies for the habit-based action selector. We are now providing more extensive justification at line page 5, line 123 in the body, and page 31, line 814 in the methods. The method we use to generate habit-based actions, using the same partially-observable Monte-Carlo planning algorithm [2] that is also used for plan-based action selection, is one which has been formally shown to converge to the optimal policy (See Theorem 1 of [2]). In addition, the increase in performance (survival rate) as the number of forward states simulated increases is clearly declining as we approach 5000, providing some evidence that convergence is occurring (for select trials not shown where we went out to 10000, we saw very little difference from 5000). Relative to biological learning of a habit over ontogeny, or selection for a policy over phylogeny, this seems likely to be an upper performance bound on what a given species or animal could achieve. However, it is advantageous to our claims to equip our prey with habit policies that approximate the best possible. For example, if we use something more realistic, then we run into questions of whether we might be underestimating the learning rates of our model prey, or the number of exposures to a given environment. In contrast, by using an approach that gives us better performance than would be typically expected with habit, then instances where planning does better will only be amplified in the context of a more realistic approach to implementing habit-based action selection.

While it is true that, even in the current implementation, habit draws from a library of paths generated by planning, it was surprising to us that in a fully dynamic task habit would perform just as well as full time planning. Despite the stereotypy seen in path spread, the paths that are generated by planning are direct reactions to predator movements. For example, for the case of a prey with a visual cone, under plan-based action selection if the prey was to observe the predator it would turn to the

opposite direction in the next time step, and in most cases go to the opposite wall from where the predator was observed. Under habit-based action selection, even if the prey was to observe the predator, it would keep taking actions along the trajectory prescribed by the planning algorithm (as initialized within the habit-based policy set) and potentially not react, causing it to run into the predator (Supplementary Movie 2, visual range 3 & 5). While direct paths that were toward the predator could be negatively weighted (essentially pruned out) of the library that habit draws from, the current implementation keeps the weight of such paths the same as unexplored paths. Even so, we see no difference in performance between habit- and plan-based action selection. In order to explain this finding, and potentially abstract away from a purely spatial domain, we conducted the eigenvector centrality analysis. Before this analysis it was not clear to us that in a fully dynamic task, an agent could simply implement a prescribed action sequence, and perform similar to a system that could react to a mobile and changing reward. This work potentially points to an interesting line of inquiry that could more directly link graph connectivity measures to the change in value function to determine in what types of dynamic tasks one would expect good performance from pure habit-based control, or how planning could be simplified by considering environmental connectivity. We have added a new section to Supplementary Text detailing potential methods and simplifications to plan-based control.

Given Reviewer 3's concerns, however, we have completely re-calculated all habit-based action selection trials, such that now the prey only learns from successful trials: any trials in which the predator is able to capture the prey are discarded from the set of success paths that are used to seed the PRQL habit-based action selection algorithm. The methods are updated to reflect this new approach (page 31, line 829) and the corresponding figures (Fig. 2d, Fig. 3f–g, Fig. 4g, and Fig. 5d) are updated.

The next portion of R3's comment concerns how eigencentality cues path diversity—this is correct; we think it is a strength of our contribution to show data supporting this in the context of the predator-prey task space, and how this diversity varies with environment type.

However, regarding “doesn't the way the habit system is set up by definition make habit and planning very similar in many situation except at places with more than one possible success path or when dynamic updating is necessary?” . . . “If it draws from information it should not have, the argument isn't as persuasive.” We are unsure what R3 is suggesting here. We would appreciate further clarity on what information the prey (or predator) should not have. We think it is an important result that for a dynamic, shifting threat, habit performs equally well as planning in certain types of

environments. With our eigencentality analysis we have attempted to explicate the underlying basis for this result.

Comment 3.2 Neuroscientifically there is more work on planning than hippocampal prefrontal interactions. However, the authors are correct to point it out for particularly spatial planning abilities but should appreciate that there are many other prefrontal components likely crucial for high level planning. Overall, while the manuscript does not contain any neuroscientific results directly, it does, as the authors point out, have some implications for neuroscientists as it makes specific predictions about when an agent should plan. It gives, for example, an evolutionary argument for why there are sweeps during decision points in electrophysiological recordings in rats. However, this ends up simply restating what the original recording study already said, i.e. that when there are decisions to be made, there is planning and potentially simulation of future states. That those moments should happen when paths diverge is, I think, obvious to us and the rat. Maybe the authors know of any unique predictions their models make that go beyond “when there are potential decision points or diverging paths the agent engages in decision making and potentially planning”, because any such predictions would be interesting and useful to experimentalists.

Response 3.2. The example of sweeps from the data on rodent decision making was to show a case where our quantitative method for predicting when planning should commence can be informative. We are uncertain why Reviewer 3 believes that predicting when these sweeps should occur is a restatement of what the original study found—as that study only described the phenomena, rather than provide a predictive model for it. The intuition that sweeps should occur when paths diverge seems correct; we hope that there is value in building on that intuition to predict the onset of planning based on eigenvector centrality, an easily computed network metric with wide application beyond spatial contexts.

Note that many studies on sweeps have been done with mazes where there are explicit decision points. Our contribution is more general, showing where decision regions should occur in arbitrary spaces.

Comment 3.3 Potentially more exciting and worthy of more emphasis are the implications for comparative neuroscience (e.g. work by Mars et al). Equipped with a model it is potentially possible not just to give a generic argument for the existence of any planning but more specific predictions about the degrees of planning that should exist in different niches/environments within the same species and between species. In fact, it would be very useful if the authors could map this out a bit more. Is it possible to run simulations of different animal species within different environments (e.g. deserts might also be equally lacking complexity) and different movements speeds

and visual ranges and get out specific predictions about their relatively level of spatial planning? If so, comparative neuroscience could use those values and predict differences in brain anatomy and functional circuits. Thinking about other predictions or implications of the model, would another prediction from the model be that there should be more planning in species with more decision diversity, i.e. larger repertoires or ways to flee? To put it another way, having more possible routes makes planning variable. Does this also mean everything that allows for more diversity in potential responses should equally make planning more valuable? For their specific prey model this would boil down to if animals have different type of avoidance behaviours, rather than visual range combined with ideally cluttered environments creating multiple success paths?

Response 3.3. Reviewer 3 has made some very interesting and helpful suggestions here. Due to concerns from Reviewer 2, we have considerably revised our method for relating gridworlds to actual habitats, using a more robust multiscale measure of habitat structure called lacunarity. Lacunarity has a large literature across terrestrial and aquatic habitats. Our results show that the narrow window in which we find planning leads to higher survival rates over habit, entropy levels 0.4–0.6, corresponds to a zone of mean lacunarity that typifies a subset of terrestrial habitats. Roughly speaking, this zone of mean lacunarity corresponds to habitats with clusters of open and closed space somewhat akin to a savanna. (One of the many theories for the rise of bipedalism in hominins argues that it is related to climate change in Africa causing a shift from dense forest to savanna, which we briefly allude to now). If we are correct, and this level of mean lacunarity does correspond to higher levels of planning benefit, then there is, we hope, some significant and worthwhile studies to be done in relating how animals perform in the habit-to-planning spectrum to their sensory ecology (given the connection to long range high resolution sensing now more emphasized in the study in Supplementary Fig. 6 and Fig. 1); the lacunarity of their ancestral environment; and their neuroanatomy (such as relative LI / IS relationship). We delve into the LI/IS comparative neurobiology issue in slightly more detail within the Discussion.

We largely agree that it is the diversity of potential responses that makes planning over them useful, whether those be spatial or not, but an additional factor has to include that these responses are contingent on other factors that shift over time (if stable, then habit is sufficient). Otherwise, learning can be handled by the less flexible habit system. Consequently, we have been careful in the Revision to indicate more precisely that our contribution concerns planning in dynamic scenarios—what we term “non-habitizable” regimes—where rewards are continuously shifting.

Finally, since ectotherms seem to have diminished planning capacity, while in certain cases possibly having habitats of the right lacunarity and having sufficient visual range, the study suggests that endothermy may be an important final element for overcoming the onerous computational demands of the non-habitizable variant of planning. This sets the stage for a number of interesting comparative analyses. We would like to delve into outlining what some of those might be, but as we are significantly over our word limit, at the present time we will stay with the points we have made above while expanding somewhat in the Supplementary Text.

Comment 3.4 The paper really only addresses spatial planning. However, there are many other aspects to planning as well as evolutionary pressures to develop it. For example, terrestrial environments allow for complex extended action sequences and tool building, which could also lead to different kind of planning. Equally, planning can happen on very different timescales such as planning and food storage. I do completely accept that all those can't be implemented in the model, but they should be discussed when talking about planning, evolution and terrestrial life.

Response 3.4. Reviewer 3 is correct that the general action-outcome representation and its efficient traversal is the heart of all planning, be it spatial or non-spatial. Just as spatial cognitive maps are speculated to have preceded non-spatial maps, we speculate that non-spatial planning is an exaptation of circuitry originally needed for successfully navigating predator-prey relationships on land. We have added some remarks to the study along those lines at page 14, line 342.

Comment 3.5 Modeling the habit-based strategy: Is it correct that the habit-based strategy is built by running 5000 simulations of the plan-based strategy? Then the “prey” has perfect knowledge of the result of each trajectory, which determines the weight with which each is randomly selected during the actual experiment? My concern is that this does not accurately reflect either how instinct evolves nor how learning happens in the lifetime of an individual.

Response 3.5. This concern is addressed in our responses to comment 3.1 above.

Comment 3.6 From figure 1 it is unclear whether trees can re-join i.e. if you are at same position at same step the tree branches should converge again.

Response 3.6. It is unclear to us how—in a tree structure—the resulting next states should re-join. For example, if the agent is at coordinate (10, 1) there are 4 possible actions that would move it to a different location from its current position. Independent of the predator location, for any action taken by the prey in imagination, the

state therefore would be different. Moreover, given that we have partial observability, each node in the search tree is associated with a belief state $B(h)$, number of times a specific history has been visited $N(h)$, and the expected value of an (action, observation) pair. Given that each node has a belief state associated with it, the likelihood that two nodes at the same time step would have the same belief state is very small.

If the Reviewer is instead asking about the building of the tree with sampling from the belief state, then it is true that there is re-use. After each expansion (an addition of a node), the prey samples a state from its belief state. The within-tree search for all of these are the same. For example, at the first iteration, the prey samples from its belief state and expands one child node based on that sampled state. At the next iteration, the prey samples from its belief state again, but uses the expanded node to perform a within-tree search. After termination is reached at the phase of out-of tree search (rollouts), the prey returns to the root node to sample a new state, and uses the already expanded tree to search through its nodes and expand each node's belief state as it is going through the tree. In order to make this point clearer we have added details to our Methods section at page 30, line 792.

Comment 3.7 Maybe it is worth mentioning whether there modeling has any implications for human planning and when planning should occur? E.g. when and whether people should have flexibly planning horizons and engage with planning only in specific environments?

Response 3.7. We have extended our Discussion points to more explicitly address humans. Our model suggests that the environments in which we see the greatest benefit in planning are similar to savannas. One hypothesis for bipedalism (addressed at page 15, line 372) suggests that hominins gained bipedalism in conjunction with a shift in African climate that lead to a transition from dense forest to patchy forest with grassland—a possible environmental structure within the range of lacunarities that maximally advantage planning. The “savanna hypothesis” holds that bipedalism was advantaged as a result of structural environmental change that forced apes that lived nearly exclusively in trees to spend more time on the ground. Through bipedalism hominins were now able to see over grassland that separated domains of taller structure (such transparency of open regions to vision is a requirement for our lacunarity analysis to hold, and may not apply therefore to many mammals and reptiles whose visual systems are not high enough to clear such grassland). Therefore, the increased visual range from bringing the eyes higher off the ground, in addition to patchy biogenic structure could have further advantaged planning in hominins, potentially selecting for the highly developed planning capa-

bilities manifest in the one extant member of the hominins. It is therefore very interesting to think about how this ancestral environment could have affected human planning horizons. Given that we only compare performance under high planning levels (5000 states forward simulated) and habit, our data is not sufficient to explain how to flexibly control planning horizons, and the link between planning horizons and environment.

While our work has been within the domain of spatial navigation, the mechanism of arbitrating between habit- and plan-based action selection is dependent on eigencentrality (and its gradient), and thus need not be spatial. In dynamic tasks that include transitioning in the graph, our results would suggest that planning should be engaged in highly connected regions. A way in which a transition from habit- to plan-based action selection could be triggered is through anxiety or other psychological correlates of high connectivity (prior research suggests that anxiety along with theta coherence increases as rodents transition from a poorly connected (arm) to a highly connected region (open area)). We have expanded on more general uses of eigencentrality in the discussion at page 14, line 342.

Comment 3.8 It is unclear from Figure 1 whether the cone is around the whole prey agent. If not, the state is not completely defined by position but also orientation, meaning direction of travel is an important consideration for strategy. However, the methods mentions sweeps and it wasn't completely clear to me whether those imply that the prey animals can sample all around themselves between moves?

Response 3.8. The cone is not around the whole prey agent, but faces the direction of travel. For example, if an agent with visual range 1 is at location (8, 1) and picks action North to (8, 2), the cone would face toward the North direction spanning coordinates $\{(7, 3), (8, 3), (9, 3), (7, 2), (9, 2), (8, 1)\}$ (visual cone, plus the agents immediate surroundings). However, in the beginning of an episode, before the prey begins planning, the prey rotates its visual cone around itself (see Methods, page 29, line 742). This approach avoids bias from a certain initial body orientation.

If the predator is within this bubble-like sensorium, the prey's belief state is initialized with that particular predator location, otherwise the prey's belief state is all other predator locations outside of its sensorium with equal probability. This process is only conducted at time step 0. We believe the direction of travel would be part of the state if the cone facing outward was decoupled from the direction of travel (i.e. if the prey agent had a neck). In that case the prey would need to account for its action and the direction of its cone, which would mean that the observations that the prey would get would be a combinatorial problem of cardinal action directions and cardinal viewing directions. However, when these two are coupled the observations

that the prey receives are a result of the direction that it moves in. The possible observations for each direction and the consequences of both observing and not observing are accounted for within the prey's belief state.

During its planning process the prey samples from its belief state. For each sample the prey chooses an action and receives an observation based on its environmental and predator model. This expected belief state based on the prey's tested samples essentially propagates the belief state. This belief state is pruned to all the positions the predator could be in, after the prey takes a real action and receives a real observation. For example, if initially the prey did not observe the predator and the real observation the prey got from taking one action was again no observation, then, at least for long visual ranges, the initial predator location was probably not immediately outside of the prey's initial sensorium. The way in which we prune and propagate belief state, along with the coupling of actions and direction of visual cone, allows the prey's decisions to be only dependent on its location and the predator location. This point is further buttressed when success trajectories under full visual range are considered. The similarity in success trajectories when the prey has full visual range and a visual cone facing the direction of motion suggests that the two variables that are needed for the prey to make a 'good' decision are the prey's own location and the predator location.

Comment 3.9 I was wondering whether the visual analysis was the right level of abstraction. It only deals with visual complexity, while it should be about spatial and navigational complexity on the level of the map the simulation uses. I think the conclusion is probably approximately correct, either way, but maybe the authors should acknowledge the problem of equating visual complexity with spatial complexity.

Response 3.9. The reviewer is correct in that our environmental complexity analysis is based on vision, and therefore in more strict terms quantifies visual complexity. However, it is important to note that this analysis also provides a quantification for the spatial distribution of environmental clutter. While we acknowledge that we do not strictly quantify navigational complexity, to the best of our knowledge there is no metric that would address both navigational complexity and visual complexity, which we believe are both important in advantaging planning. Therefore, we have quantified—in loose terms, the variance of cell visibilities—to understand both the spatial distribution of occlusions (affecting navigational complexity) as well as the visual complexity of the environment (via network complexity). It is also important to note that while the actual navigational complexity and visual complexity for cognitive map-based simulations may only approximate each other, we also wanted to be more broad to encompass the high isocortical fraction case discussed at the end of

the contribution: in novel environments for which a high fidelity cognitive map does not yet exist, it may be visual complexity that is (at least initially) more important during plan-based action selection.

One possible way to think about navigational complexity is to think about how many different paths exist between either two cells or between the agent start and goal cell. However, in all of our environments, whether it be low or high entropy, there are an infinite number of possible paths between the start cell and goal cell, unless perhaps certain actions are limited (e.g. opposite actions that would take the agent back to the previous cell). In mid-entropy environments some of the most interesting behaviors that we see involve back and forth motions (e.g. hiding), and thus restricting actions may miss aspects of navigational complexity under a dynamic threat.

A better approach to understand navigational complexity under a dynamic threat would possibly be to look at the evolution of the value function over time due to the presence of a dynamic threat. Our intuition is that this is roughly approximated by the eigenvector centrality metric. For example, in low entropy environments, a more fundamental reason as to why we see very little to no behavioral variability in the action choices of agents is possibly that despite the task featuring a dynamic threat that would conventionally necessitate the relearning of the value function at each time step (essentially, to plan), there is in practice very little change in the value function. However, in mid-entropy environments that feature adjacent clusters of high and low eigenvector centralities, a possible reason for why we see behavioral variability and the need to plan in regions of high eigenvector centrality is because there is a significant amount of change in the value function, making the task fully dynamic, and necessitating the re-learning of the value function via planning at each step. Therefore in some sense, while our task is dynamic it may not be 'fully dynamic' in all environments (which is further evidenced by the success of blind habit in low and high entropy environments). Part of what would make navigation more complex is how dynamic the environment is. Although we have not provided a rigorous account of this explanation, the success of our hybrid decision maker suggests that this may be the underlying cause. Therefore, we have tried to address complexity in navigating a space under a dynamic threat through the quantification of the spatial clustering of eigencentality.

However, in our explanation of spatial complexity at page 7, line 174 we have explicitly acknowledged that network complexity is based on a visual graph, and thus quantifies visual complexity and approximates spatial complexity.

Comment 3.10 I was also curious what the eigenvector centrality correlation with number of likely successpaths is for a given location?

Response 3.10. The reviewer asks a very interesting question that is difficult to answer. One major difficulty is that the set of possible success paths between two cells could essentially be infinite. To answer this question rigorously we would need to answer a corollary question to address how the value function changes with respect to the dynamic predator in these different environment. As mentioned in Response 3.8, our intuition is that in low and high entropy environments, despite the task being dynamic, the state-value function does not significantly change due to the dynamic threat. This in turn creates these stereotypical paths that we see in both of these types of environments, while also allowing blind habit to succeed (although high entropy environments are rather trivial since the environment is so constricted). We are currently working on building a more theoretical and rigorous relationship between number of viable futures and eigencentality.

Nevertheless, we will try to answer this question in a more roundabout way. We will approximate the number of success paths based on the number of times each action was chosen from each cell when an episode resulted in survival. This is essentially no different than the success path heatmaps that we generate. Our aim here is to generate a probability distribution over action choices, based on the frequency of each action choice for a given cell. For example, in low entropy environments, because of the stereotypy of success paths, there is one action that is chosen at a higher frequency, and thus will have a greater probability of being chosen (e.g. 'East' for cells along the bottom right wall). Cases where we see diffuse success paths (i.e. mid-entropy environments)—which arise as a result of immediate responses to the changing predator location—will likely feature an almost uniform probability distribution over allowable action choices.

In order to understand the correlation between the distribution of action choices and eigencentality, we clustered the environment eigencentalities into 4 distinct clusters. At the one end, these clusters identified regions of very low eigencentraity (e.g. walls in empty environments), and on the other end they identified regions that have very high eigencentality (e.g. the center in empty environments). This process was only done for low and mid entropy environments due to the triviality of high entropy environments outlined above. We then calculated the correlation between the mean variance of action distributions for a given cluster with the eigencentality value at the cluster center. Here, the mean variance of an action distribution is higher if it is skewed toward one action (e.g. compare the variances two action distribution arrays: $Var([100, 3, 0, 0]) \approx 1839$, and $Var([30, 21, 31, 21]) \approx 23$ since in both cases the mean is $E([30, 21, 31, 21]) = 25.75$). Therefore, action distributions which only have one or two actions chosen at higher frequency will have larger variances when compared to action distributions that are considerably more uniform across allowable actions.

One limitation of this approach is that in most cases we do not uniformly sample each cell. In low entropy environments, high eigencentality regions are almost never visited. We have only calculated action distributions if there was visitation to that particular cell. In doing these calculations, we find that across all environments with entropies that lie between 0.0–0.6, the mean correlation between eigencentality and variance of action distribution is -0.66. This implies that as eigencentality increases the probability distribution of action choices becomes broader, suggesting that there may be an increase in the number of viable futures from that position for which planning is required to account for their differences in value. Therefore, the action choice in high eigencentality will depend on the configuration of the environment, where the prey is with respect to the predator and the occlusions.

Comment 3.11 There is a typo in the following: “734 the prey observed the predator during this sweep than it knew the predator initial location, which” Then not than and predators not predator

Response 3.11. This has been fixed.

Response to Reviewer 4

Comment 4.1. The paper “The shift from life in water to life on land..” is an extremely exciting one, a major step forward in understanding brain-behavior evolution from an integrated computational and ecological perspective. It combines both empirical and modeling work in a novel way—while the main information presented derives from modeling of predator-prey interactions in a variety of environments, an extraordinary amount of integration of basic empirical information, from a wide range of fields, is presented a prerequisite to this integration. These subjects presented include the comparative organization of map formation and habit formation in the brain, optical analysis of animal/environment systems, particularly resolution properties, fractal analysis of various environmental topographies, energetics of respiration and locomotion in land versus water, kinetics of animal pursuit and escape, and more.

This paper raises a great number of questions, the raising of which should not be taken as criticism, as it is a major strength of this paper that it opens new research fields. A good paper should not be penalized for doing so. I will list a number of them, assuming the authors can decide what kind of question each is: weaknesses that might be addressed, misunderstandings arising from problems in exposition, desirable questions that they intended to evoke; future studies, and so on.

There is one feature of exposition that I think might be improved—the author’s narrative structure is story-like, raising the immediate information about brain structure and environment relevant to their modeling work, and proceeding through their modeling experiments in a logical sequence. As they proceed, however, the reader encounters one major evolutionary question after another with a proposed answer in part or whole, as well as applications of this work for interpreting laboratory experiments. (as an aside, the one large question they do mention in the abstract, the basis of the limited ability of animals to plan, is hardly addressed at all). I think some of this might be made explicit in the abstract or introduction or the significance of these experiments could be missed. For example, what is the likely reason for the increase in cognitive ability from water to land? How is homeothermy related? To what extent do sensory specializations themselves drive the computational structure of decision making? Is there a predator/prey arms race, and on what features? How does memory capacity inform decision-making? How might we understand common laboratory measures like open field “anxiety”, freezing and so forth from this ecological context?

Response 4.1. We thank Reviewer 4 for pointing this out and we have made changes to the introduction and abstract, and added “core cognitive faculty” to title, to make some of these points more prominent: 1) we’ve specified the likely cause of the increase in cognitive ability from water to land—a combination of an increase in visual

range and consequent profusion in the number of viable futures due to a “gappy” habitat revealing and hiding dynamic adversaries at long distances; 2) We have moved forward the homeothermy and computational complexity point; 3) We have amplified the anxiety point by referring to “regional openness and its psychological correlates” within the abstract and elsewhere in the paper.

Comment 4.2. This reader would also have appreciated somewhat better delineation of when the authors are choosing a particular stance in research areas that are controversial, when they’re simply citing established work (like the fractal geometry of the environment?) and when they’re making their best guess in areas that haven’t been much studied. Overall, a bit more meta-comments both at the intention and the data aspects of the project.

Response 4.2. We have done this in several places to address this concern. In the introduction, we now both cite the claim that seascapes are less complex than landscapes from a review we have found, as well as refer to the fact that we compile previously published data to establish this fact; we make it clear that it is our hypothesis that vision is a key sensory modality for planning (penultimate paragraph of Discussion); in the final paragraph we make it clear that it is our perspective that this was key to the evolution of planning. In a variety of other locations we have endeavored to make the Reviewer’s requested delineation clearer.

Comment 4.3. Lines 32-35: Is a direct parallel between lateral striatum and hippocampus/pfc in birds and mammals with the indirect and direct pathways in lamprey striatum intended? Possibly better to say that there is an overall homology of basic reinforcement of motor decisions. Might you say why the focus of the paper is delineating these two components of the forebrain, and not other brain divisions?

Response 4.3. We have modified the intro to be more general, and removed the direct comparisons to the lamprey basal ganglia at page 2, line 38. Instead we say that there is a ‘conserved organizational structure of the basal ganglia’.

In the plan-based section of our introduction we wanted to overview current research. Rodent literature suggests that the synchrony between the hippocampus and the prefrontal cortex occurs during planning, and is important for temporally extended tasks. Despite the difference in brain architecture between birds and mammals (nucleated rather than laminated), we wanted to emphasize that current research suggests that computations akin to those performed by the mammalian cortex also occur in birds. For example, it has been shown that lesions to the nidopallium caudolaterale (NCL) results in a tendency to perservate on previously rewarded stimuli (revert back to habit), similar to what is seen when lesions to the mammalian PFC are performed. In doing a more parallel comparison between mammalian PFC

and avian NCL we wanted to emphasize that a lot functional findings within the better known mammalian research extends to avian decision making. We have tried to emphasize this point at page 2, line 56.

Comment 4.4. Lines 36: “nonlocal spatial representations”? Maybe just say prediction or “imagination”?

Response 4.4. The literature has not reached consensus that nonlocal spatial representations are indicative of prediction or imagination, although some studies interpret it in this way. However, we agree that this connection does exist and helps the understanding of the reader, so we have added “sometimes interpreted as prediction or imagination” to the end of this sentence.

Comment 4.5. Lines 43-44: It’s my understanding that many researchers would locate a lot of the action-sorting in the basal ganglia proper, back to thalamus, to PFC/motor commands, contextualized by hippocampus. I think the interpretation offered in the paper is reasonable, but this might be a place where selection of an interpretation might be signaled.

Response 4.5. We wanted to convey the theorized need for the interaction between PFC and the hippocampus for value encoding of imagined simulations. We have adjusted the wording to move away from ‘imagined action sequences’ to more general ‘imagined simulations’ to emphasize the role of general option sorting that is done through this interaction (page 2, line 53).

Comment 4.6. Lines 66-68: The distinction between “prey’s location” in the egocentric (e.g. left 30 degrees) and not hippocampal, versus allocentric (e.g.the meadow I hunted yesterday) should be made somewhere.

Response 4.6. Addressed by specifying the hippocampal contribution is to provide allocentric location.

Comment 4.7. Lines 74-80: Struggling a bit here to distinguish different classes of prediction. Overall, though, this is an unusually clear exposition of a very confusing literature.

Response 4.7. We have adjusted the language regarding these two classes of predictions to try to help in distinguishing between them at page 3, line 95.

Comment 4.8. Lines 87-89: in one of these animals at a time — I thought you meant just the prey for a while

Response 4.8. R4 is correct in her initial impression that we just meant the prey. We have made this clearer at page 4, line 104. Note that when the predator is out of

view, there is a mechanism for propagating an updated belief as to the likely location of the predator, noted at page 30, line 792 and page 33, line 881.

Comment 4.9. Lines 99: again, ego-allocentric. You'd been talking about the predator tracking the prey with uncertainty, and you didn't mark the change about which spatial context you were referring to.

Response 4.9. We have specified allocentric location here.

Comment 4.10. Circa 112: I'm having some trouble visualizing how this scales up from a single, simple escape response to a sequence of actions, but the next section of the paper made the kinds of outlines much more clear. You might telegraph the reader to restrain their anxiety.

Response 4.10. We have signaled that the text R4 is referring to is an abstract description that will be followed by concrete examples at page 5, line 120.

Comment 4.11. Line 229: also very nice. I quite liked your supplementary Figure 3 for visualizing the strategies — are you out of allowed figures?

Response 4.11. Currently we are significantly over length. Should you accept our resubmission, we will consult with you as to where this should be corrected, and we expect this is likely to also rule out R4's suggestion here.

Comment 4.12. Line 254: A lot more information about how you acquired the images for the fractal analysis would be nice, both here and in the methods. You don't even mention the terrestrial — perhaps it's because this is very well known?

Response 4.12. Since we have completely revised this portion of the study to use lacunarity rather than fractal dimension so that this part of the study has better data support, we expect R4 will no longer have this concern.

Comment 4.13. Lines 311-340: Discussion of strategies. I'm wondering whether the assumption that the layout of the map is completely available to the animals makes a whole lot of sense, especially for occluding "clutter". Is that perhaps part of what pushes behavior out of the planning approach?

Response 4.13. Concern about equipping our agents with a perfect map was raised by Reviewer 2. In order to address both Reviewer 4 & 2's concern about the perfect map assumption, we have added a model limitations section to our Supplementary Text that discusses the need for such a simplification. Our response there will I think address R4's concern as well:

Perfect map assumption. In the case of our assumption that the preda-

tor and prey have a perfect cognitive map (in the predator's case, excluding the location of the safety point that the prey is trying to reach), our modeling choice is driven by a desire to surgically isolate the relative properties of planning versus habit. First, the assumption that the predator has a perfect cognitive map (minus the prey's goal) is the most detrimental to the prey's survival; we think this is a good assumption to keep particularly as we are not able to simultaneously implement planning in the predator due to computational limitations. In the context of the current body of literature on model-based decision making, let's consider the following possibilities for the prey: 1) no cognitive model (no cognitive map), but the prey learns the model; 2) inaccurate cognitive model (map), updated with learning; 3) inaccurate cognitive model (map), not updated with learning; 4) perfect cognitive model (map), not updated with learning.

For 1) our study becomes much more difficult (in fact, we are not sure how we would proceed), since a large number of empirically poorly-justified modeling choices on learning rates and number of exposures to a given random environment would need to be made so that the planning algorithm has a model over which to plan. For 2) largely the same considerations apply, but now the planning algorithm can generate action decisions; however, those action decisions will be based on a value function that corresponds to this inaccurate cognitive model. The decisions may not be optimal. It is unclear, in this case, in what way to add noise or error to the cognitive model. Putting that problem aside, if the decisions fail to be optimal, in the context of this scenario (irreversibility since death occurs upon capture), there may not be much to learn; but inasmuch as learning can occur, the issues raised about the complexity of our study and poorly justified parameters enter in. For 3) decisions may again not be optimal; but now the failures of the cognitive model—again, the parameterization of which is uncertain—are not corrected over time. For 4) the attraction of this choice for our study is avoidance of uncertain parameterization of learning the model, and avoidance of uncertain parameterization of having an inaccurate model. We believe this enables us to surgically isolate the impact of planning in the context of a generous assumption of learning having previously been completed, and completed with perfect fidelity to the underlying structure (not counting the predator), where that structure is necessarily static (since learning cannot occur to accommodate a dynamic environment, and a dynamic environment without learning would quickly lead to an inaccurate model—case 3).

Comment 4.14. Line 359: on mammals —To me, the elephant in the room here (obstructing the egress) is that most early mammals are nocturnal. You should really discuss this some.

Response 4.14. An important point, indicated by the new Supplementary Figure 1, is that even in moonlit and starlit conditions terrestrial nocturnal vision does better than aquatic diurnal vision. We have added this point in the Discussion at page 16, line 406. This can help resolve strange patterns in the role of vision in nocturnal mammals, such as the need for visual cues in mazes that are learned via olfactory cues, as referenced in the Discussion.

Comment 4.15. Line 370: LI-IS inverse correlation. There's no evidence that developmental pattern is a "constraint" —minimally, it's just the mechanism by which a selected pattern is put in place. Neural thigmotaxis — best overall strategy.

Response 4.15. We have removed this sentence.

Comment 4.16. 394 – Nice!

Methods: I am not able to evaluate the mathematical detail, but I have no reason to distrust it.

Reviewer – Barbara Finlay, Cornell University

In closing, we once again want to thank all the reviewers for their detailed reviews, feedbacks, and insightful comments. We hope these responses meet their concerns adequately. Since addressing all of the reviewers' concerns has added to the text length, we would appreciate their feedback on any parts of the revision they feel could be streamlined.

M. Maclver

- [1] Andreas Hula, P Read Montague, and Peter Dayan. Monte Carlo planning method estimates planning horizons during interactive social exchange. *PLoS Comput. Biol.*, 11(6):e1004254, 2015.
- [2] David Silver and Joel Veness. Monte-Carlo planning in large POMDPs. In *Adv. Neur. In.*, pages 2164–2172, 2010.
- [3] Nathaniel D. Daw. Are we of two minds? *Nature Neuroscience*, 21(11):1497–1499, 2018. ISSN 1546-1726. doi: 10.1038/s41593-018-0258-2.
- [4] Roy E Plotnick, Robert H Gardner, and Robert V O’Neill. Lacunarity indices as measures of landscape texture. *Landsc. Ecol.*, 8(3):201–211, 1993.
- [5] Roy E Plotnick, Robert H Gardner, William W Hargrove, Karen Presteggaard, and Martin Perlmutter. Lacunarity analysis: a general technique for the analysis of spatial patterns. *Phys. Rev. E*, 53(5):5461, 1996.
- [6] C Broglio, I Martín-Monzón, FM Ocaña, A Gómez, E Durán, C Salas, and F Rodríguez. Hippocampal pallium and map-like memories through vertebrate evolution. *J. Behav. Brain Sci.*, 5(03):109, 2015.
- [7] Soizic Le Fur, Emmanuel Fara, Hassane Taïssou Mackaye, Patrick Vignaud, and Michel Brunet. The mammal assemblage of the hominid site TM266 (Late Miocene, Chad Basin): ecological structure and paleoenvironmental implications. *Naturwissenschaften*, 96(5):565–574, May 2009. ISSN 1432-1904. doi: 10.1007/s00114-008-0504-7.
- [8] M. A. Maclver, L. Schmitz, U. Mugan, T. D. Murphey, and C. D. Mobley. Massive increase in visual range preceded the origin of terrestrial vertebrates. *Proc. Natl. Acad. Sci. U.S.A.*, 114(12):E2375–E2384, 2017.
- [9] D. E. Nilsson, E. Warrant, and S. Johnsen. Computational visual ecology in the pelagic realm. *Phil. Trans. R. Soc. B*, 369(1636), 2014.
- [10] W. E. Stein, C. M. Berry, L. V. Hernick, and F. Mannolini. Surprisingly complex community discovered in the mid-Devonian fossil forest at Gilboa. *Nature*, 483(7387):78–81, Mar 2012.
- [11] Geerat J Vermeij. How the land became the locus of major evolutionary innovations. *Current Biology*, 27(20):3178–3182, 2017.
- [12] Takehisa Yamakita and Tadashi Miyashita. Landscape Mosaicism in the Ocean: Its Significance for Biodiversity Patterns in Benthic Organisms and Fish. In *Integrative Observations and Assessments*, Ecological Research Monographs, pages 131–148. Springer Japan, Tokyo, 2014. ISBN 978-4-431-54783-9. doi: 10.1007/978-4-431-54783-9_7.
- [13] Katya E. Kovalenko, Sidinei M. Thomaz, and Danielle M. Warfe. Habitat com-

- plexity: approaches and future directions. *Hydrobiologia*, 685(1):1–17, Apr 2012. ISSN 1573-5117. doi: 10.1007/s10750-011-0974-z.
- [14] Saachi Sadchatheeswaran, Coleen L. Moloney, George M. Branch, and Tamara B. Robinson. Using empirical and simulation approaches to quantify merits of rival measures of structural complexity in marine habitats. *Marine Environmental Research*, 149:157 – 169, 2019. ISSN 0141-1136.
- [15] Erica A. Newman, Maureen C. Kennedy, Donald A. Falk, and Donald McKenzie. Scaling and complexity in landscape ecology. *Frontiers in Ecology and Evolution*, 7:293, 2019.
- [16] Brian H Kaye. *A random walk through fractal dimensions*. John Wiley & Sons, 2008.
- [17] David P Williams. Fast unsupervised seafloor characterization in sonar imagery using lacunarity. *IEEE Trans. Geosci. Remote Sens.*, 53(11):6022–6034, 2015.
- [18] Jai C Sleeman, Gary A Kendrick, Guy S Boggs, and Bruce J Hegge. Measuring fragmentation of seagrass landscapes: which indices are most appropriate for detecting change? *Mar. Freshw. Res.*, 56(6):851–864, 2005.
- [19] Mark S Fonseca and Susan S Bell. Influence of physical setting on seagrass landscapes near Beaufort, North Carolina, USA. *Mar. Ecol. Prog. Ser.*, 171: 109–121, 1998.
- [20] Daehyun Kim, David Cairns, and Jesper Bartholdy. Spatial heterogeneity and domain of scale on the Skallingen salt marsh, Denmark. *Dan. J. Geogr.*, 109: 95–104, 01 2009. doi: 10.1080/00167223.2009.10649598.
- [21] Sam Purkis, Giulio Casini, Dave Hunt, and Arnout Colpaert. Morphometric patterns in Modern carbonate platforms can be applied to the ancient rock record: Similarities between Modern Alacranes Reef and Upper Palaeozoic platforms of the Barents Sea. *Sediment. Geol.*, 321:49–69, 2015.
- [22] Mary K Donovan. *A Synthesis of Coral Reef Community Structure in Hawaii and the Caribbean*. PhD thesis, 2017.
- [23] James Richard Anderson. *A land use and land cover classification system for use with remote sensor data*, volume 964. US Government Printing Office, 1976.
- [24] JN Perry, AM Liebhold, MS Rosenberg, J Dungan, M Miriti, A Jakomulska, and S Citron-Pousty. Illustrations and guidelines for selecting statistical methods for quantifying spatial pattern in ecological data. *Ecography*, 25(5):578–600, 2002.
- [25] Merav Seifan and Ronen Kadmon. Indirect effects of cattle grazing on shrub spatial pattern in a mediterranean scrub community. *Basic Appl. Ecol.*, 7(6):

496–506, 2006.

- [26] Sebastian Hoehstetter, Ulrich Walz, and Nguyen Xuan Thinh. Adapting lacunarity techniques for gradient-based analyses of landscape surfaces. *Ecol. Complex.*, 8(3):229–238, 2011.
- [27] Yadvinder Malhi and Rosa María Román-Cuesta. Analysis of lacunarity and scales of spatial homogeneity in IKONOS images of Amazonian tropical forest canopies. *Remote Sens. Environ.*, 112(5):2074–2087, 2008.
- [28] Gordon W Frazer, Michael A Wulder, and K Olaf Niemann. Simulation and quantification of the fine-scale spatial pattern and heterogeneity of forest canopy structure: A lacunarity-based method designed for analysis of continuous canopy heights. *Forest Ecol. Manag.*, 214(1-3):65–90, 2005.
- [29] Xiuzhen Li, Hong S He, Xugao Wang, Rencang Bu, Yuanman Hu, and Yu Chang. Evaluating the effectiveness of neutral landscape models to represent a real landscape. *Landsc. Urban Plan.*, 69(1):137–148, 2004.
- [30] Žiga Fišer, Simona Prevorčnik, Nina Lozej, and Peter Trontelj. No need to hide in caves: shelter-seeking behavior of surface and cave ecomorphs of *Asellus aquaticus* (Isopoda: Crustacea). *Zoology*, 134:58 – 65, 2019. ISSN 0944-2006. doi: <https://doi.org/10.1016/j.zool.2019.03.001>.
- [31] C Ware, Jennifer Dijkstra, Kristen Mello, A Stevens, B O’Brien, and W Ikedo. A novel three dimensional analysis of functional-architecture that describes the properties of macroalgae as refuge. *Marine Ecology Progress Series*, 608: 93–103, 01 2019. doi: 10.3354/meps12800.
- [32] Mehdi Keramati, Peter Smittenaar, Raymond J. Dolan, and Peter Dayan. Adaptive integration of habits into depth-limited planning defines a habitual-goal-directed spectrum. *Proc. Natl. Acad. Sci. U.S.A*, 113(45):12868–12873, 2016. doi: 10.1073/pnas.1609094113.
- [33] Eleanor M. Caves, Nicholas C. Brandley, and Sönke Johnsen. Visual acuity and the evolution of signals. *Trends Ecol. Evol.*, 33(5):358–372, 2019/03/18 2018. doi: 10.1016/j.tree.2018.03.001.
- [34] T. W. Cronin. The visual ecology of predator-prey interactions. In Pedro Barbosa and Ignacio Castellanos, editors, *Ecology of Predator-Prey Interactions*, chapter 6. Oxford University Press, 2014.
- [35] M. E. Nelson and M. A. Maclver. Sensory acquisition in active sensing systems. *J. Comp. Physiol. A*, 192(6):573–586, 2006.
- [36] Brigitte Müller, Steven M Goodman, and Leo Peichl. Cone photoreceptor diversity in the retinas of fruit bats (megachiroptera). *Brain, behavior and evolution.*,

70(2):90–104, 2007. ISSN 0006-8977.

- [37] Efrén Álvarez-Salvado, Angela M Licata, Erin G Connor, Margaret K McHugh, Benjamin MN King, Nicholas Stavropoulos, Jonathan D Victor, John P Crimaldi, and Katherine I Nagel. Elementary sensory-motor transformations underlying olfactory navigation in walking fruit-flies. *Elife*, 7:e37815, 2018.
- [38] Adam J Calhoun, Sreekanth H Chalasani, and Tatyana O Sharpee. Maximally informative foraging by *caenorhabditis elegans*. *Elife*, 3:e04220, 2014.
- [39] Jean-Baptiste Masson. Olfactory searches with limited space perception. *Proc. Natl. Acad. Sci. U.S.A.*, 110(28):11261–11266, July 2013. ISSN 0027-8424, 1091-6490. doi: 10.1073/pnas.1221091110.
- [40] L. M. Miller, Y. Silverman, M. A. Maclver, and T. D. Murphey. Ergodic exploration of distributed information. *IEEE Transactions on Robotics*, 32(1):36–52, Feb 2016. ISSN 1552-3098. doi: 10.1109/TRO.2015.2500441.
- [41] Chen Chen, T. Murphey, and M.A. Maclver. Sense organ control in moths to moles is a gamble on information through motion. *bioRxiv*, 2019.

Reviewers' Comments:

Reviewer #2:

Remarks to the Author:

The authors have made considerable effort to respond to and make changes to address referee comments. I very much appreciate this. However, the major challenge with this work remains: the results of the analysis and simulations are simply not sufficient to support the main claims made in the Title, Abstract, and at the end of the Discussion. To be precise, the simulations that constitute the core results of this paper do not, in any general sense, show that a transition to land advantaged the cognitive faculty of planning, or led to the evolution of the capacity for planning. What the results show is that under a detailed, and specific set of assumptions used in the formulation of action selection models and simulated environment, planning-based action selection outperforms habit-based action selection at intermediate habitat complexity when a prey is escaping from a predator using only vision. To turn this into a very general statement about the evolution of the capacity for planning requires a major logical leap that I think many, including myself, will be unwilling to make. As I emphasized in my previous comments, over-selling the results in this way seems to me counterproductive; it undermines the value of the more immediate results the authors present, which despite their limitations are still interesting. The scope of the results is much more accurately summarized in line 63-64, where the authors state: "Here, we test the hypothesis that in non-habitizable scenarios, plan-based action selection is advantaged in proportion to visual range and environmental complexity." Lines 65-74 follow this statement, and in my opinion, give a more reasonable interpretation of the results. As the authors rightly point out in their response to the previous round of comments, every analysis proceeds from the starting point of assumptions and axioms. The best such analyses show that a specific conclusion holds under very general assumptions. Such results are powerful because they reveal deep underlying relationships that hold regardless of the fine details. The analysis the authors present in their manuscript is something different; they claim a very general conclusion about the evolution of cognitive ability, but they draw these conclusions from simulation of a very specific scenario. In this case, I suspect many of the fine details of the assumptions do matter (e.g. prey plan, predators don't; simulations take place in an arena with a border; environment and plans occur in discrete space, spatial environment is fully known, etc.), and although the authors have made laudable efforts to justify some of these (e.g. structural complexity of different landscapes, visual range, etc.), other assumptions are unjustified and perhaps even untestable. The design and analysis of the simulations are well-done. It is just that the major conclusions simply do not follow from the analysis. If the claims of the paper were dialed back to be more in line with the immediate results of the simulations, I think the paper would be much improved. But I am not sure it would be suitable for a general interest journal such as Nature Communications.

Reviewer #3:

Remarks to the Author:

I am happy with all the answers from the authors and congratulate them on a very nice paper.

Reviewer #4:

Remarks to the Author:

I believe the authors have done a thorough, and very impressive job of replying to all of the four reviewers. Particularly, I think that the tactical choices in modeling in the absence of unlimited computing power are now very well explained, and the new method of quantifying of environments, "lacunarity", is now appropriately backgrounded in the paper, allowing the main features of the model

to be underlined. The paper is a very much easier read as a result of successive focusing on model features in different environments, with fewer digressions. I would also underline my first evaluation of this paper as exceptionally interesting and groundbreaking, raising a large number of new research questions and approaches. Models need not be post-hoc summaries of masses of empirical observations – in this case, where the main parameters of interest can be laid out so clearly, they can make future empirical data-gathering much more intelligently structured.

That the discussion now comes around to deliver more detail on what a “core cognitive faculty” might be is good, but I also applaud their attempts to address variability of that core faculty. My sole suggestion is that they might look at the Discussion with an eye to stepping back from the jargon necessary to describe their model in prior sections. For example, “This allows habit-based strategies to succeed during a non-habitizable task.” is an odd importation of language originally used to describe basal ganglia and frontal-cortical computational models, here to describe evolved or innate behaviors. It actually obscuring an important new research possibility! What they’ve described is a way to use models of prey escape to generate a potential catalogue of evolutionarily-fixed strategies, even suggesting how they might be neurally implemented, for fixed strategies that cannot be learned in an animal’s lifetime. “Eigencentality” (variations of) might similarly restated about the choice environments confronting the animal, and what might be recognized.

BL Finlay